# Inhibitors supercharge kinase turnover through native proteolytic circuits

Natalie S. Scholes[1], Martino Bertoni[2], Arnau Comajuncosa-Creus[2], Katharina Kladnik[1], Xuefei Guo[3], Fabian Frommelt[1], Matthias Hinterndorfer[1,4], Hlib Razumkov[5,6], Polina Prokofeva[7], Martin P. Schwalm[8], Florian Born[8], Sandra Roehm[8], Hana Imrichova[1], Brianda L. Santini[1], Eleonora Barone[1], Caroline Schätz[1], Miquel Muñoz i Ordoño[1], Severin Lechner[1], Andrea Rukavina[1], Iciar Serrano[1], Miriam Abele[1], Anna Koren[1], Stefan Kubicek[1], Stefan Knapp[8], Nathanael S. Gray[6,9], Giulio Superti-Furga[1,10], Bernhard Kuster[7,11,12], Yigong Shi[13], Patrick Aloy[2,14] & Georg E. Winter[1,4 ✉]

Targeted protein degradation is a pharmacological strategy that relies on small molecules such as proteolysis-targeting chimeras (PROTACs) or molecular glues, which induce proximity between a target protein and an E3 ubiquitin ligase to prompt target ubiquitination and proteasomal degradation[1]. Sporadic reports indicated that ligands designed to inhibit a target can also induce its destabilization[2–4]. Among others, this has repeatedly been observed for kinase inhibitors[5–7]. However, we lack an understanding of the frequency, generalizability and mechanistic underpinnings of these phenomena. Here, to address this knowledge gap, we generated dynamic abundance profiles of 98 kinases after cellular perturbations with 1,570 kinase inhibitors, revealing 160 selective instances of inhibitor-induced kinase destabilization. Kinases prone to degradation are frequently annotated as HSP90 clients, therefore affirming chaperone deprivation as an important route of destabilization. However, detailed investigation of inhibitor-induced degradation of LYN, BLK and RIPK2 revealed a differentiated, common mechanistic logic whereby inhibitors function by inducing a kinase state that is more efficiently cleared by endogenous degradation mechanisms. Mechanistically, effects can manifest by ligand-induced changes in cellular activity, localization or higher-order assemblies, which may be triggered by direct target engagement or network effects. Collectively, our data suggest that inhibitor-induced kinase degradation is a common event and positions supercharging of endogenous degradation circuits as an alternative to classical proximity-inducing degraders.

In addition to proximity-inducing modalities, sporadic accounts of inhibitor-induced target degradation have been reported, for example, for inhibitors of gene-regulatory proteins such as BCL6 (refs. 2,3) or EZH2 (ref. 4) and, most frequently, for kinase inhibitors[5–7]. Many kinases require chaperones such as HSP90 for folding and maintained stability[8]. After kinase binding, some inhibitors, including clinical agents, have been shown to disrupt kinase–HSP90 interactions, which in turn results in kinase destabilization[9]. This process of chaperone deprivation is well established and is exemplified by the degradation of HER2 by neratinib[10] and the degradation of LMTK3 by C28 (ref. 11). Given that hyperactive mutant kinases are frequently

more reliant on HSP90, chaperone deprivation has also been associated with preferential degradation of mutant over wild-type (WT) kinases, such as the destabilization of EGFR(G719S) by erlotinib[12]. However, detailed studies, for example, of the mechanism of mutant selective PI3Kα degradation by taselisib and inavolisib[13,14], revealed that mechanisms can go beyond the widely accepted framework of chaperone deprivation.

Owing to the high conservation of the ATP-binding site in the kinome, many orthosteric inhibitors bind unselectively. Promiscuous inhibitors can act as selective destabilizers, as exemplified by sorafenib, which binds to many kinases[15] but destabilizes only BRAF(V600E) or

[1]CeMM, Research Center for Molecular Medicine of the Austrian Academy of Sciences, Vienna, Austria. [2]Institute for Research in Biomedicine (IRB Barcelona), Barcelona, Spain. [3]Beijing Frontier Research Center for Biological Structure, Tsinghua-Peking Joint Center for Life Sciences, School of Life Sciences, Tsinghua University, Beijing, China. [4]AITHYRA Research Institute for Biomedical Artificial Intelligence of the Austrian Academy of Sciences, Vienna, Austria. [5]Department of Chemistry, Stanford School of Humanities and Sciences, Stanford University, Stanford, CA, USA. [6]Department of Chemical and Systems Biology, ChEM-H, Stanford School of Medicine, Stanford University, Stanford, CA, USA. [7]Chair of Proteomics and Bioanalytics, Technical University of Munich, Freising, Germany. [8]Institute of Pharmaceutical Chemistry, Goethe University, Frankfurt am Main, Germany. [9]Stanford Cancer Institute, Stanford School of Medicine, Stanford University, Stanford, CA, USA. [10]Center for Physiology and Pharmacology, Medical University of Vienna, Vienna, Austria. [11]German Cancer Consortium (DKTK), partner site Munich and German Cancer Research Center (DKFZ), Heidelberg, Germany. [12]Bavarian Biomolecular Mass Spectrometry Center (BayBioMS), Technical University of Munich, Freising, Germany. [13]School of Life Sciences, Westlake University, Hangzhou, China. [14]Institució Catalana de Recerca I Estudis Avançats (ICREA), Barcelona, Spain. ✉e-mail: gwinter@aithyra.at

RET(M918T)[12,16]. Importantly, this polypharmacology brings about an opportunity of 'network drugging', whereby an inhibitor induces kinase degradation without direct engagement. Mechanistically, this can occur through modulation of an upstream kinase that activates a phosphodegron, leading to degradation of a downstream kinase[17]. Network effects can also be involved with directly acting inhibitors. For example, mutant PI3Kα degradation by inavolisib is dependent on hyperactive HER2 signalling[14].

The identification of monovalent kinase degraders has largely been serendipitous. We therefore lack systematic insights into their pervasiveness and mechanistic principles beyond chaperone deprivation. Owing to the importance of kinase inhibitors in modern medicine (80 FDA approved drugs as of January 2024, another 180 in clinical trials[18]), efforts to quantify and mechanistically dissect inhibitor-induced kinase degradation could identify therapeutic opportunities, explain adverse effects or outline principles of degrader design beyond proximity induction.

To systematically identify inhibitor-induced kinase destabilization, here we map the dynamic abundance profiles of 98 kinases after cellular perturbations with 1,570 kinase inhibitors. Our efforts cover 88 canonical (WT) and 10 mutant kinases. In total we identify 232 compounds that downregulate protein levels of at least one kinase and 66 kinases that are affected by at least one compound. We find that the predisposition of mutant kinases quantitatively and qualitatively differs from their WT counterparts. Even though destabilized kinases are enriched for HSP90 clients, many instances cannot be explained by chaperone deprivation. Notably, we encounter that the propensity of a kinase to be destabilized by an inhibitor is not correlated with its degradability by PROTACs, differentiating inhibitor-induced degradation from degradation by proximity induction[19].

Follow-up of three previously undescribed kinase degraders suggests an underlying mechanistic principle in which inhibitors accelerate endogenous degradation circuits that preferentially recognize a particular kinase state, a concept that we refer to as supercharging. By inducing these states, inhibitors destabilize the cellular pool of a kinase. We describe several mechanisms that lead to kinase degradation. For example, inhibitors can induce degradation-prone kinase states by modulating kinase activity (exemplified by LYN), perturb intracellular kinase localization (BLK) and induce higher-order kinase assemblies (RIPK2). Collectively, our findings highlight a unifying framework of inhibitor-induced target degradation.

## Charting a monovalent degrader map

To assess how inhibitors affect kinase levels, we opted for a scalable luminescent reporter setup using a lentiviral expression system in which 98 kinase open reading frames (88 canonical, 10 mutants) are expressed as nanoluciferase (Nluc) fusions in K562 cells (Fig. 1a, Extended Data Table 1 and Supplementary Data 1). As this setup informs on target abundance rather than stability, control measures were put into place, enabling us to segregate temporal inhibitor effects from global perturbations of transcription or translation. Moreover, we profiled control cell lines expressing long- and short-lived non-kinase control target (GFP–Nluc and destabilized GFP (dGFP)–Nluc, respectively; Methods). This panel of a total of 100 cell lines was dynamically assayed against an annotated library of 1,570 kinase inhibitors at regular time intervals (2, 6, 10, 14 and 18 h). Compounds were selected for minimal assay interference, excluding Nluc quenchers and compounds that showed cytotoxicity during the assayed time window (Methods and Supplementary Data 1). Scoring of the resulting dataset was performed using a multitiered scheme (Fig. 1a and Methods). This resulted in a total of 232 compounds that score and elicit destabilization across 66 of the tested kinases, including 7 mutants (Fig. 1a,b, Extended Data Table 1 and Supplementary Data 1). Among all of the hits (Supplementary Fig. 9), we identified 160 unique kinase–compound pairs, which are denoted

as selective (Fig. 1b). Reporter half-life was only weakly correlated with the frequency of scoring (Spearman's rank correlation = −0.381, $P$ = 0.0034373), supporting that the implemented controls successfully filtered out most global perturbations interfering with transcription or translation (Extended Data Fig. 1a).

Across all kinases, HER2, ABL1 and the mutant kinase RAF1(S257L) emerged as the most frequently degraded kinases. No specific family was enriched among kinases prone to destabilization (Extended Data Fig. 1b). When comparing our data to a recently published survey that assessed PROTAC-induced global kinase degradability[19], no global correlation was observed, suggesting that inhibitor-induced degradation mechanistically differs from degradation based on proximity induction (Extended Data Fig. 1c). By contrast, when assessing the HSP90 status of each kinase[8], both strong and weak clients had a markedly higher prevalence of being destabilized compared to non or not-defined clients (Fig. 1c). This points to an outsized contribution of chaperone deprivation to the observed degradation events, even though we cannot exclude that individual client kinases are degraded in a chaperone-independent manner. Given the established strong correlation of HSP90 clients, these observations also extend to the HSP90 co-chaperone CDC37 within the kinome[8]. Among the annotated HSP90 clients was also the frequently destabilized HER2. Closer examination of the HER2-degrading compounds in our survey revealed that we identified inhibitors such as AV-412, afatinib or neratinib, which have previously been implicated as HER2 degraders that function through chaperone deprivation[10,20–22]. Moreover, our data revealed destabilizing inhibitors, such as WZ4002 or dacomitinib. Notably, many of the identified HER2 destabilizers feature covalent warheads. Consistent with a direct effect, introducing a C805S mutation[23] in HER2 prevented inhibitor-induced degradation (Extended Data Fig. 1d,e). Further supporting an on-target effect, the identified degraders formed a definite cluster when mapped on their target space[24] (Fig. 1d).

With HER2 degraders exemplifying directly acting degraders, we next set out to identify degradation events driven by network modulation. Mapping the experimentally identified ABL1 destabilizers on their target space, we observed a dispersed distribution, yet we identified one cluster of hit compounds which coincided with dual PI3Kα and mTOR inhibitors (Extended Data Fig. 1f). Although we cannot exclude that the short half-life (1.5 h) of ABL1 results in this prevalence of scoring, we note that the compounds failed to score in the control dGFP (half-life, 1.1 h) cell line (Extended Data Fig. 1g and Supplementary Fig. 5), suggesting an effect beyond low baseline stability. As the screen was conducted in the BCR–ABL-driven leukaemic cell line K562, many of the most potent ABL1 inhibitors were eliminated from the screening library owing to their acute cytotoxicity. This probably contributed to an under-representation of directly acting ABL1 degraders. Akin to HER2 and ABL1, we analysed all remaining kinases (Extended Data Fig. 2). However, clustering of the hits on a per kinase basis failed to reveal generalizable trends, probably due to the limited number of identified degraders for most kinases and limited availability of comprehensive binding data for the assayed compounds. Likewise, binding modes of inhibitors did not significantly differ between hit and non-hit compounds (type I enrichment odds ratio = 0.74, $P$ > 0.05; type II enrichment odds ratio = 1.3, $P$ > 0.05).

Focusing on mutant kinases, we initially identified a decrease in protein half-life for activating over non-activating mutations (Fig. 1e), suggestive of an activity–stability trade-off[25]. Overall, we find distinctive degradation patterns with little overlap comparing WT and mutants (Jaccard distances (JD) ≥ 0.8; Fig. 1f). Although in some cases the degradation frequency corresponds to the reduced protein stability, we identify most hits for EGFR(G719S), which is more stable than the strongly transformative mutants. These observations imply that, mechanistically, the enhanced degradability of mutant kinases goes beyond the reduced half-life and could be rooted in altered signalling

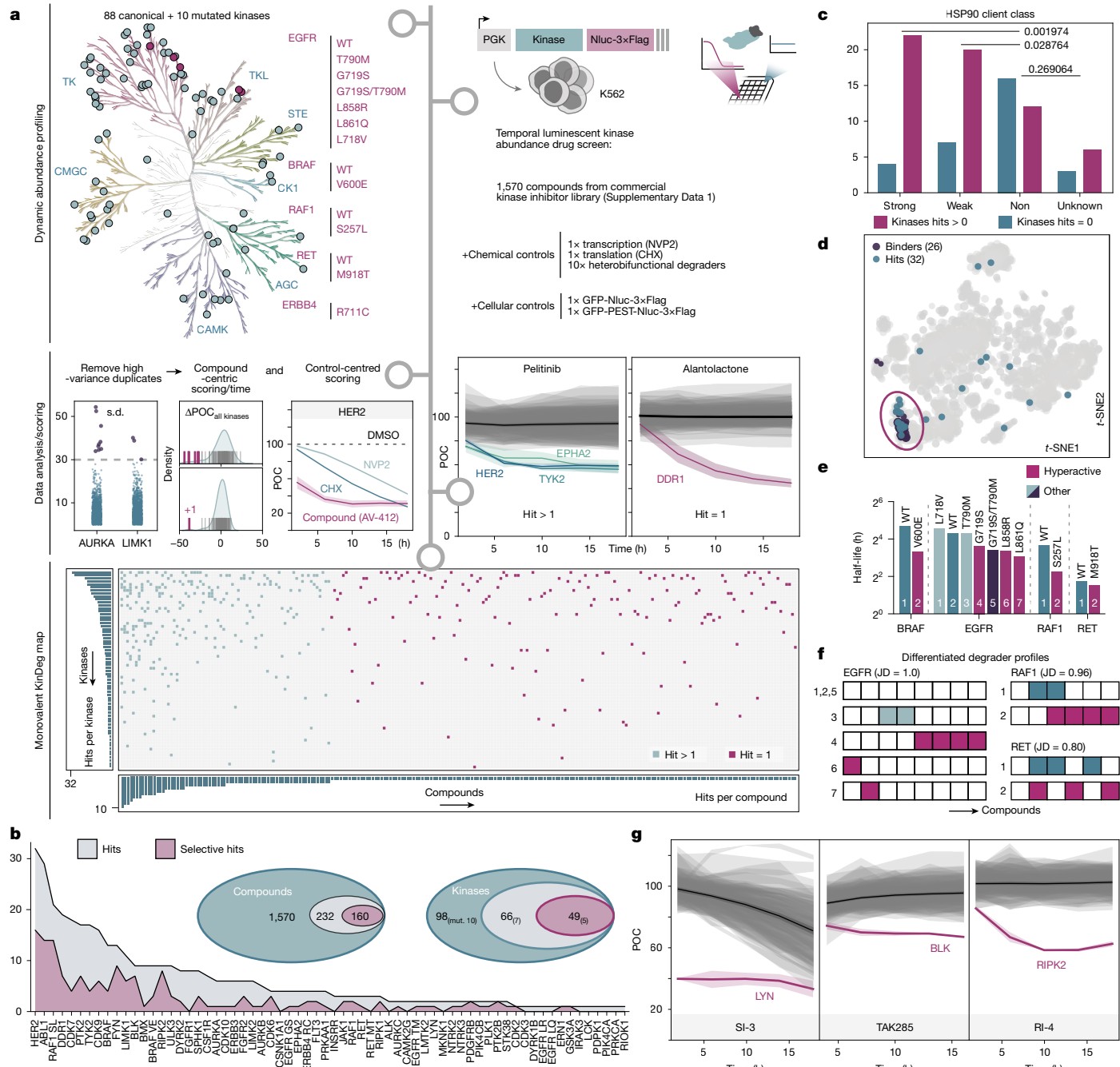

**Fig. 1 | Kinase degradation (KinDeg) map across 1,570 monovalent kinase-targeting compounds. a**, KinDeg map. Top left, breakdown of analysed kinases, including 88 canonical and 10 mutated kinases (highlighted in pink). Top right, the drug screen setup, including a breakdown of controls. The kinase tree illustration was reproduced courtesy of Cell Signaling Technology. Middle, data processing and hit calling to classify whether a compound perturbs kinase abundance. Example data are depicted for a small molecule with more than one downregulated kinase (hit > 1; pelitinib) and for a selectively perturbed kinase (hit = 1; alantolactone). POC, per cent over control. Bottom, the resulting binary KinDeg map sorted according to the observed degradation frequencies across kinases and compounds, including adjacent histograms of the summed scores in both dimensions. **b**, Breakdown of hit scores across the kinases and compounds (subscripts indicate mutated kinases). **c**, Stratified kinases for scoring at least

once (downregulated) or non-scoring kinases with respect to their HSP90 client status as defined previously[8]. Statistical analysis was performed using pairwise two-sided Fisher's exact tests. **d**, t-Distributed stochastic neighbour embedding (t-SNE) plot of the compound target landscape (Methods) across the used drug screening library with annotated HER2 binders (violet) and HER2 destabilizing screening hits (dark green) forming a co-cluster. **e**, Comparisons of experimentally derived half-lives across mutant and canonical kinase pairs. **f**, Representative hit comparison and JD calculations across matched canonical and mutant EGFR, RAF and RET data. Number and colour annotations are as defined in **e**. The full dataset is provided in Supplementary Data 1. **g**, Temporal screening trajectories of the top three hits selected for detailed mechanism of action elucidation. Data are the mean ± confidence intervals of m = 2 technical replicates.

and interactomes. Finally, orthogonal to destabilization, we identified multiple inhibitor-induced stabilization events covering 64 of our tested kinases, including also 5 mutant kinases as well as ABL1 and CDK2 (Supplementary Data 1 and Extended Data Fig. 3a–c). ABL1, for

example, was stabilized by multiple binders targeting the allosteric myristic-acid-binding site (Extended Data Fig. 3d,e).

After determining the frequency and general features of inhibitor-induced degradation, we next set out to dissect the mechanism of

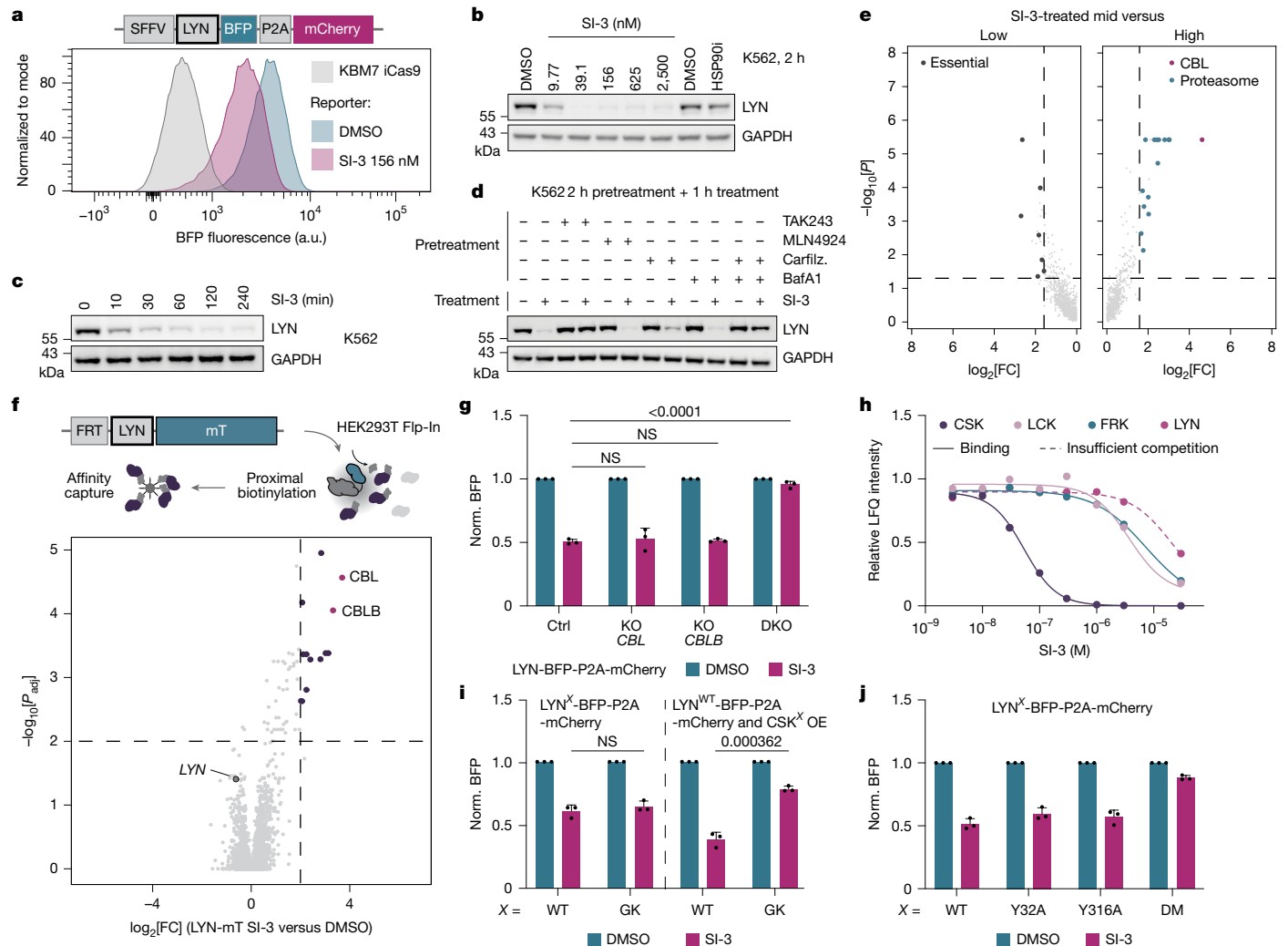

**Fig. 2 | LYN is rapidly degraded by SI-3 through a canonical activity–stability switch. a**, Flow cytometry stability reporter data of SI-3-induced LYN degradation (8 h). **b**, Dose-dependent degradation of endogenous LYN by SI-3 (half-maximum degradation concentration ($DC_{50}$) < 9 nM, maximal degradation ($D_{max}$) = 96%, quantified in Supplementary Fig. 6b). $n$ = 3. **c**, Time-dependent, SI-3-induced endogenous LYN degradation (quantified in Supplementary Fig. 6c). $n$ = 3. **d**, Chemical rescue. $n$ = 3. Carfilz., carfilzomib. **e**, The results of the ubiquitin–proteasome system (UPS)-focused FACS-based CRISPR–Cas9 screen. Essential genes (Methods) or CBL and proteasome subunits are highlighted in cases in which $P$ < 0.05 (one-sided MAGeCK) and $\log_2$-transformed fold-change ($\log_2$[FC]) > 1.585. $n$ = 2. **f**, Differential BioID results after SI-3 treatment. Enriched prey is highlighted for a $\log_2$[FC] > 2 and $-\log_{10}$[$P_{adj}$] > 2 (two-sided, analysis of variance (ANOVA), with $P$ value adjustment using the Benjamini–Hochberg method). $n$ = 3. **g–j**, Normalized BFP was calculated as the ratio to mCherry and normalized (norm.) to the respective genetic perturbations (Methods). **g**, Flow cytometry assay of the LYN stability reporter after genetic perturbation of a control gene (sgAAVS1), *CBL* or *CBLB*, or both E3s (*CBL* and *CBLB* dual KO, DKO). Statistical analysis was performed using two-way ANOVA with Sidak correction. NS, not significant ($P$ > 0.9999). $n$ = 3. **h**, The results of chemoproteomics analysis for SI-3, depicting selected hit kinases and LYN (insufficient competition). All other hits are shown in Supplementary Fig. 6g–i. $K_D$: 34.60 nM (CSK), 2.987 μM (LCK) and 11.43 μM (FRK). LFQ, label-free quantification. **i**, Flow cytometry analysis of SI-3-mediated destabilization of LYN WT or gatekeeper (GK) stability reporter, or WT LYN in the presence of overexpressed CSK WT or GK. OE, overexpression. Statistical analysis was performed using unpaired $t$-tests; NS, $P$ = 0.376936. $n$ = 3. **j**, Flow cytometry measurement of SI-3-mediated destabilization of LYN WT, LYN(Y32A), LYN(Y316A) or LYN(Y32A/Y316A) (double mutant, DM). $n$ = 3. All flow cytometry was performed using KBM7-iCas9 cells. SI-3 was administered at 156 nM unless specified otherwise. Data are mean ± s.d. $n$ represents biological replicates.

action of three selective inhibitor–kinase pairs with different HSP90 client status[8]: LYN (strong client, degraded by SRC inhibitor 3 (ref. 26), hereafter SI-3), BLK (weak client, degraded by TAK285 (ref. 27)) and RIPK2 (non-client, degraded by RIPK-IN-4 (ref. 28), hereafter RI-4) (Fig. 1g and Extended Data Fig. 1h).

## Rapid destabilization of LYN by SI-3

A notable effect in our dataset was the degradation of the SRC-family kinase LYN by the compound SI-3 with near-complete protein ablation after 2 h (Fig. 1g). Using a previously described fluorescence protein stability reporter[29], we validated that SI-3 affects LYN stability (Fig. 2a).

The effect of SI-3 was also confirmed through immunoblotting for endogenous LYN, revealing that degradation occurs in the nanomolar range (Fig. 2b) and already minutes after ligand exposure (Fig. 2c). Out of a total of 260 quantified kinases, LYN was the most significantly destabilized kinase in a quantitative expression proteomics experiment (Extended Data Fig. 4a,b). Consistent with the annotation as a SRC-family inhibitor and previously published kinome selectivity profiling[26], we established that SI-3 inhibits recombinant LYN protein with a half-maximum inhibitory concentration ($IC_{50}$) of 56.7 nM (Extended Data Fig. 4c). Notably, even though 25 out of the 1,570 profiled inhibitors are annotated to bind to LYN, SI-3 was the only inhibitor that prompted LYN degradation (Extended Data Fig. 4d). Additional manual validation

of LYN degradation on the endogenous protein levels across a selected set of LYN binders equally could not identify any further degraders (Extended Data Fig. 4e).

Although SI-3-induced LYN degradation was ubiquitin dependent (TAK243), it was not sensitive to inhibition of neddylation (MLN4924), thereby excluding dependency on a cullin-RING E3 ubiquitin ligase (CRL) (Fig. 2d). Moreover, degradation was not rescued by pharmacological inhibition of the proteasome (carfilzomib) or lysosomal acidification (bafilomycin A1, BafA1). Only simultaneous inhibition of both degradation pathways prevented LYN degradation. Albeit annotated as a strong HSP90 client[8], pharmacological inhibition of HSP90 did not induce LYN degradation both on the endogenous and reporter setup, therefore excluding chaperone deprivation as the underlying mechanism (Fig. 2b and Extended Data Fig. 4f).

To decipher the mechanism of SI-3-induced LYN degradation, we opted for a two-pronged discovery approach to (1) identify which genes are required to induce LYN degradation after SI-3 treatment and (2) chart how SI-3 treatment changed the interactome of LYN. To map genes required for SI-3-dependent LYN degradation, we performed a fluorescence-activated cell sorting (FACS)-based CRISPR–Cas9 screen in KBM7 cells with an inducible Cas9 (iCas9) allele[29] and the LYN stability reporter. This revealed the E3 ligase *CBL* as the strongest enriched gene (Fig. 2e). CBL had previously been associated with the turnover of multiple SRC-family kinases[30,31], including LYN[32,33]. In support of a physiologically relevant interaction, we identified *CBL* as the strongest hit not only in SI-3 treated cells, but also in vehicle-treated (DMSO) conditions (Extended Data Fig. 4g). To map how SI-3 altered LYN interactions, we used a BioID[34,35] setup in which LYN was expressed as fusion to the miniTurbo (mT) biotin ligase, enabling the identification of proteins that are recruited to LYN after cellular SI-3 treatment by mass spectrometry (MS). Consistent with the CRISPR–Cas9 screen, this led to the identification of CBL as the strongest recruited effector (Fig. 2f and Extended Data Fig. 4h). Moreover, we found the closely related ligase CBLB as a strongly enriched LYN interactor in a SI-3-dependent manner, suggestive of a potential functional redundancy. Indeed, population-level knockout (KO) of *CBL* was insufficient to rescue SI-3-induced LYN degradation (Fig. 2g), but substantially increased the baseline levels (Extended Data Fig. 4i), therefore explaining why it scored as a hit in the CRISPR–Cas9 screen. Likewise, single KO of *CBLB* showed no significant rescue. Only combined genetic disruption of both effectors fully rescued degradation in the stability reporter and endogenously (Fig. 2g and Extended Data Fig. 4j). Having identified this redundancy, we next addressed whether the two E3s would separately be responsible for licensing LYN's proteasomal or lysosomal degradation. Combining single ligase KO with pharmacological inhibition of either degradation pathway revealed that CBLB-mediated degradation preferentially co-opts the proteasomal machinery, while degradation mediated by CBL relies on both proteasomal and lysosomal degradation (Extended Data Fig. 5a). Consistent with the ability to engage both cellular degradation routes, SI-3 induced LYN ubiquitination involved Lys48- and Lys63-linked ubiquitin chains (Extended Data Fig. 5b). Similar to other SRC-family members, LYN is N-terminally anchored to the membrane through lipid modifications. To assess whether membrane association is required for degradation, we genetically and chemically disrupted LYN's membrane association. While degradation is independent of membrane anchorage, we noted that cytosolic LYN is exclusively degraded by SI-3 through the proteasomal route (Extended Data Fig. 5c–e).

Physiological activation of LYN has been associated with the degradation of LYN through CBL and CBLB[32]. Counterintuitively, this connects LYN activation to the degradation mechanism for the inhibitor SI-3, prompting us to more closely investigate the cellular target spectrum of SI-3. Chemoproteomic profiling revealed 11 high-confidence targets, including 8 kinases (Fig. 2h and Supplementary Fig. 6g–i). We identified CSK ($K_D = 34.60$ nM) as the most potently engaged kinase

target, confirming previously reported recombinant assay data of SI-3 ($IC_{50} = 4$ nM CSK)[26]. In contrast to our in vitro data (Extended Data Fig. 4c), LYN was engaged only at much higher concentrations in the cellular context (Fig. 2h). Orthogonally, we validated SI-3 target engagement with CSK using a NanoBRET displacement assay (half-maximum effective concentration ($EC_{50}$) = 15 nM; Extended Data Fig. 5f). On the basis of the known role of CSK as a negative regulator of SRC kinases, including LYN, we therefore hypothesized that SI-3, despite directly binding to LYN, might induce indirect LYN degradation through its preferential inhibition of CSK. Similar observations have been made with a chemical genetics setup[33]. To validate this hypothesis, we genetically modified either LYN or CSK to impair drug binding of SI-3 and assessed the consequences for SI-3-induced LYN degradation. Mutating the gatekeeper residue of LYN (T319I)[36] did not alter the degradation capacity. However, overexpression of the CSK gatekeeper mutant (T266M)[37] rescued degradation markedly (Fig. 2i). This suggested that inhibition of CSK is the dominant driver of SI-3-mediated LYN destabilization and that SI-3 functions by perturbing the intrinsic regulatory network. Consequently, blocking CSK-induced LYN activation through pharmacological inhibition of LYN rescued the effect of SI-3-induced LYN degradation on endogenous protein levels (Extended Data Fig. 5g), phosphorylation of CSK's target site LYN Tyr508 was reduced after SI-3 treatment (Extended Data Fig. 5h,i) and mutating Tyr508 to alanine rendered LYN resistant to SI-3 degradation (Extended Data Fig. 5j).

Previous research identified Tyr32 as a phosphodegron that is important for CBL recognition[33]. However, LYN(Y32A) was not resistant to SI-3-induced degradation, pointing to an additional, redundant phosphodegron (Fig. 2j). Indeed, while degradation of WT LYN was dependent on both CBL and CBLB, we found that degradation of LYN(Y32A) solely depended on CBL (Extended Data Fig. 5k), suggesting that this E3 recognizes the elusive phosphodegron. Turning back to our BioID dataset, we identified increased phosphorylation of Tyr316 in LYN after SI-3 treatment (Extended Data Fig. 5l). Supporting a functional role of Tyr316, we found that the double-mutant LYN(Y32A/Y316A) was almost completely inert to SI-3-induced degradation (Fig. 2j). The degradation of the single LYN(Y316A) mutant was comparable to the degradation of WT LYN, suggesting a redundancy of both phosphodegrons in SI-3-induced LYN degradation. Contrary to LYN(Y32A), LYN(Y316A) retained dependency on both E3 ligases CBL and CBLB for SI-3 induced degradation (Extended Data Fig. 5m).

In summary, our data identified SI-3 as uniquely differentiated inhibitor that is sufficiently selective for CSK over LYN to exploit an endogenous activity–stability switch that ensures immediate and near-complete LYN degradation after its activation.

## γ-Secretase governs BLK degradation

From several inhibitors that cause downregulation of BLK in our assay, we focused on the selective hit TAK285 (Fig. 1g). First, we validated that TAK285 affects BLK stability (Extended Data Fig. 6a,b). TAK285-induced BLK degradation was ubiquitin and proteasome dependent, but independent of neddylation or lysosomal degradation (Extended Data Fig. 6c). TAK285 equally affected endogenous BLK levels without downregulating any other kinase (Fig. 3a). BLK is a weak HSP90 client[8] and showed sensitivity towards chaperone deprivation by HSP90 inhibition at comparable kinetics (Extended Data Fig. 6b,c). We therefore hypothesized that TAK285 functions through the HSP90 regulatory axis.

To map the underpinning genetic determinants, we performed a FACS-based, genome-wide CRISPR–Cas9 screen (Fig. 3b and Extended Data Fig. 6d). Notably, we identified all four members of the γ-secretase complex (APH1A, NCSTN, PSEN1 and PSENEN) as the most strongly enriched hits, suggesting a functional link to BLK degradation by TAK285. We validated involvement of the γ-secretase by pharmacological inhibition through DAPT (Fig. 3c and Extended Data Fig. 6e,f), as well

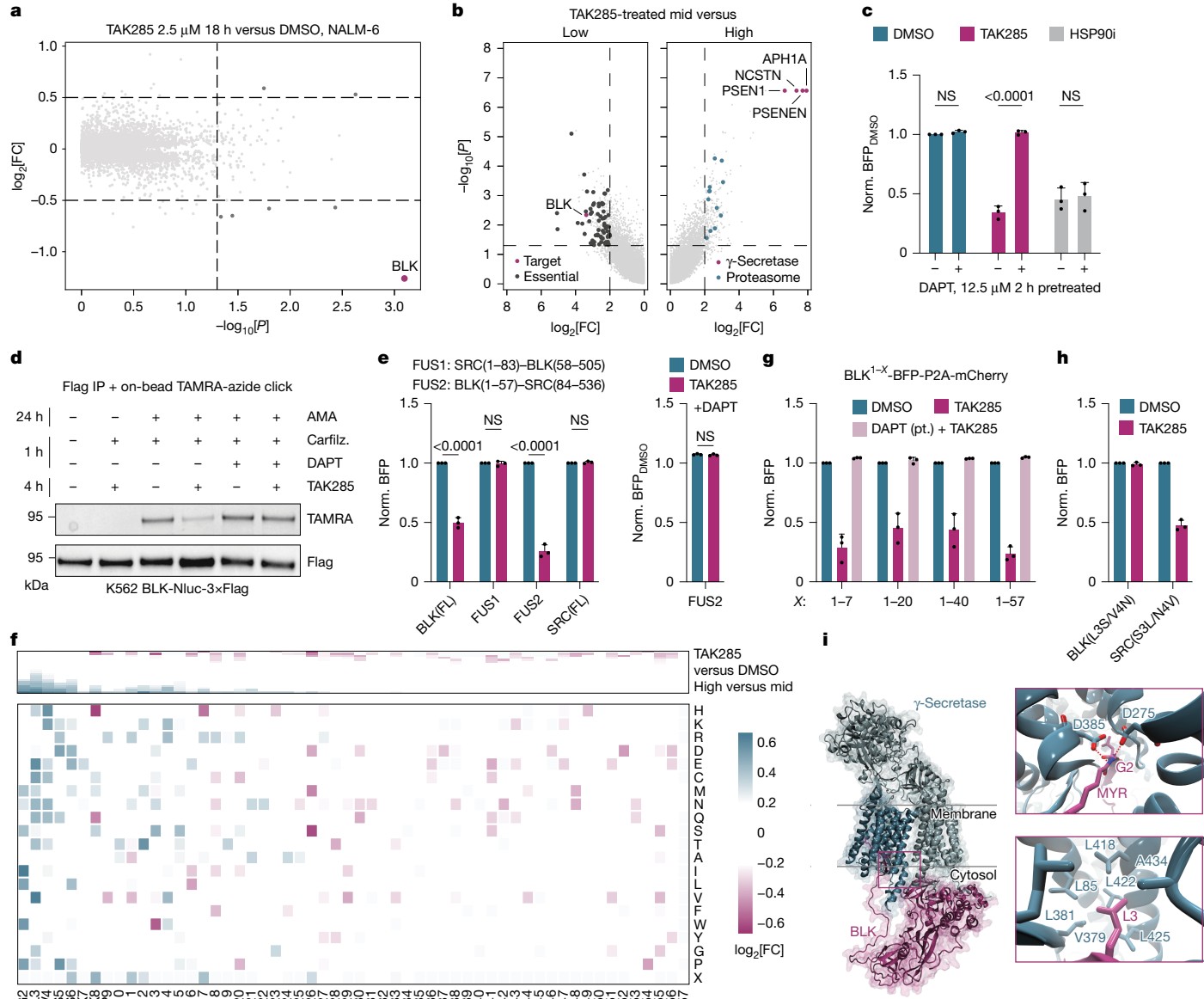

**Fig. 3 | BLK is degraded by TAK285 in a γ-secretase-dependent manner.**
**a**, Expression proteomics of TAK285-treated NALM-6 cells. Kinases are highlighted where $P < 0.05$ (one-way ANOVA) and $|\log_2[FC]| > 0.5$. $n = 3$.
**b**, Genome-wide FACS-based CRISPR–Cas9 screen. Essential genes, BLK, γ-secretase and proteasome subunits are highlighted in cases in which $P < 0.05$ (one-sided MAGeCK) and a $\log_2[FC] > 2$. $n = 2$. **c**, Flow cytometry assay for BLK stability reporter cells pretreated with DAPT followed by DMSO, TAK285 or HSP90i (6 h; NS, $P = 0.9978$ (DMSO), 0.9966 (HSP90i)). $n = 3$.
**d**, Immunoprecipitation in BLK-Nluc-3×Flag K562 cells after pretreatment with alkynyl myristic acid (AMA, 100 µM), followed by 1 h carfilzomib (1 µM) and/or DAPT plus DMSO or TAK285. In-gel fluorescence (top, TAMRA) and Flag immunoblotting (bottom) analysis of the immunoprecipitated fractions.
**e**, Stability reporter data for the indicated constructs treated with DMSO or TAK285 (left) (6 h; NS, $P \geq 0.9999$ (FUS1), 0.9969 (SRC full length (FL))). $n = 3$.

Right, FUS2 stability reporter data after pretreatment with DAPT (2 h) followed by DMSO or TAK285 (6 h; NS, $P = 0.7333$) $n = 3$. **f**, DMS data for the TAK285-treated (6 h) BLK stability reporter panel, displayed as the DMSO-normalized $\log_2[FC]$ of sorted fractions. $n = 3$. **g**, Flow cytometry analysis of unique domain stability reporter fusions treated (6 h) with DMSO or TAK285. $n = 3$. pt., pretreatment. **h**, Flow cytometry analysis as in **g** for the indicated stability reporters. **i**, AlphaFold3-derived model of the BLK γ-secretase complex (Methods). Critical H-bonds are shown at the top right and the positioning of L3 is shown at the bottom right. Normalization of flow cytometry data was performed against the respective genotype/pretreatment unless specified otherwise. Data are mean ± s.d. $n$ represents biological replicates. For **c** and **e**, statistical analysis was performed using two-way ANOVA with Tukey's test for multiple comparisons. For **a**–**h**, inhibitor concentrations were as follows: TAK285 (2.5 µM), HSP90i (10 µM) and DAPT (12.5 µM).

as by genetic ablation (Extended Data Fig. 6g). In contrast to TAK285, BLK degradation induced through an HSP90 inhibitor (HSP90i) was independent of γ-secretase function. Thus, TAK285-induced BLK degradation is functionally differentiated from chaperone deprivation. Notably, γ-secretase subunits also scored as hits in steady-state (vehicle-only) conditions of the CRISPR–Cas9 screen (Extended Data Fig. 6d), which we orthogonally confirmed by genetic ablation of *PSENEN* (Extended Data Fig. 6h). This implied a role of γ-secretase

in native BLK turnover. Supporting this, we identified an interaction between BLK and NCSTN when performing proximity labelling-based proteomics (Extended Data Fig. 6i). Mining of previously reported immunoprecipitation–MS data similarly revealed baseline interactions between NCSTN and BLK[38]. We orthogonally confirmed the interaction of BLK and γ-secretase in vitro (Extended Data Fig. 6j; $K_D$ 2.53 µM). Contrary to a traditional molecular-glue-like mechanism, this affinity is not enhanced by TAK285 (Extended Data Fig. 6j; $K_D$ 3.49 µM). Collectively,

the data support a physical interaction between BLK and the γ-secretase that is functionalized after TAK285 treatment.

TAK285 only partially inhibited BLK in vitro (around 37%, $IC_{50} > 30 \mu M$; Supplementary Fig. 7g) and competition experiments suggested a mechanism that is independent of orthosteric BLK binding (Extended Data Fig. 6k). Moreover, TAK285 did not bind to γ-secretase, suggestive of a network effect (Extended Data Fig. 6l). To identify the potential intermediate kinase targeted by TAK285, we performed two orthogonal, dose-ranging chemoproteomics experiments. Kinobead profiling[39] revealed three binding partners: the established TAK285 target EGFR, the kinase MAP2K5 (MEK5) and the non-kinase protein ERCC2 (Supplementary Data 6). BLK was not identified as direct interactor (Extended Data Fig. 6m), supporting biochemical data. Orthogonal to the Kinobead profiling, we performed direct target enrichment through a tethered TAK285 analogue (Supplementary Data 7), which revealed MAP2K5 as the only overlapping target. However, KO of *MAP2K5* did not alter TAK285-mediated BLK degradation (Extended Data Fig. 6n). Collectively, these data suggested that TAK285-induced BLK degradation is independent of orthosteric BLK binding or direct γ-secretase engagement, and does not further augment the affinity between BLK and γ-secretase but instead depends on a network effect driven by a yet elusive target.

## TAK285 alters the localization of BLK

To further decipher this mechanism, we focused on understanding the contribution of the γ-secretase, an intramembrane protease that is best known for its involvement in Notch-1 and APP. Thus far, all reported protein targets are type I transmembrane proteins. However, recent evidence revealed that γ-secretase can cleave lipid-anchors, specifically on membrane-tethered small molecules[40]. We therefore surmised that TAK285 could induce γ-secretase cleavage of the myristoylation anchor of BLK, thereby releasing BLK into the cytosol and unveiling an unstable N terminus that is rapidly turned over. Indeed, we observed membrane dissociation of BLK after TAK285 treatment (Extended Data Fig. 7a,b). Moreover, orthogonal perturbation of the membrane association of BLK through pharmacological inhibition of *N*-myristoyltransferases 1/2 (IMP-1088) also destabilized BLK. Consistent with our hypothesis, cytosolic BLK could not be further destabilized by TAK285 (Extended Data Fig. 7c). Leveraging a TAMRA click reaction with alkynyl myristic acid, we could next validate that TAK285 treatment induces a γ-secretase-dependent loss of myristoylation on BLK (Fig. 3d). Collectively, these data support a mechanism whereby TAK285 treatment decreases BLK myristoylation to cause a membrane-to-cytosol transition of BLK where it is rapidly turned over.

Myristoylation of BLK and other SRC kinases occurs at the N-terminal unique domain[41]. Consistent with a critical role of BLK's N-terminal domain in determining the specificity of the TAK285-induced degradation, we found that a domain swap to the N-terminal domain of SRC retained membrane association (Extended Data Fig. 7d) but disabled TAK285-mediated degradation (Fig. 3e). Inversely, fusing BLK's unique domain to SRC was sufficient to enable TAK285-mediated and γ-secretase-dependent degradation (Fig. 3e). We therefore turned our attention towards deciphering the role of the N-terminal unique domain of BLK. Mutating all residues that could directly be modified through upstream phosphorylation networks initially led us to identify that BLK(S6A) strongly abrogated inhibitor-induced degradation (Extended Data Fig. 7e). The conserved Ser6 residue had previously been associated with regulating myristoylation and membrane association of the related SRC-family kinase LCK[42]. Accordingly, BLK(S6A) lost membrane association, appeared predominantly cytoplasmatic and had a lower baseline stability (Extended Data Fig. 7d,f). Thus, the BLK(S6A) mutant mirrors the effects of pharmacological myristoylation inhibition on BLK WT.

To gain an unbiased per-residue-resolved map of the unique domain, we next opted for a deep mutational scanning (DMS), mutating every residue of the unique domain (residues 2–57) to each other possible amino acid. Cells were transduced with stability reporter variant libraries, drug treated and sorted based on BFP expression (Fig. 3f and Extended Data Fig. 7g). IMP-1088 dependent destabilization was mainly abrogated by mutations on G2, the position that is myristoylated in cellulo. TAK285 showed clear dependencies on residues Gly2, Leu3, Val4 and Ser6. Focusing on the TAK285-specific positions, we could identify variants (L3A, L3G, L3V, V4S) that disrupted TAK285-mediated degradation (Extended Data Fig. 7h). Notably, these variants remained N-terminally myristoylated, anchored to the membrane and therefore sensitive to IMP-1088 treatment (Extended Data Fig. 7h–k). Systematically shortening unique domain fusions to BFP highlighted the first seven amino acids as minimal motif required for TAK285-mediated degradation (Fig. 3g). Moreover, mutating solely residues Ser3 and Asn4 in SRC to the corresponding residues of BLK rendered SRC degradable by TAK285 in a γ-secretase-dependent manner (Fig. 3h and Extended Data Fig. 7l). Vice versa, the opposite mutations in BLK convey resistance to TAK285 degradation. In summary, while G2 mutations globally alter myristoylation status and shift BLK into the cytosol, mutations on Leu3 and Val4 more specifically impair TAK285-mediated delocalization and degradation.

Enabled by this residue-level, functional understanding, we predicted the complex of BLK and γ-secretase using AlphaFold3 (ref. 43) (Fig. 3i). Notably, the most confident model directly placed the N-terminal domain within the active site of γ-secretase. Specifically, across molecular dynamics simulations, we consistently observe that the peptide bond between the myristic acid and BLK Gly2 is coordinated by two hydrogen bonds between the two catalytic residues Asp385 and Asp275 of the γ-secretase, therefore providing a structural rationale for the experimentally observed loss in BLK myristoylation and the ensuing cytoplasmic localization (Fig. 3d and Extended Data Fig. 7a,b). Furthermore, we noted that the functionally relevant Leu3 is placed in a thermodynamically favourable hydrophobic pocket just neighbouring the catalytic site (Fig. 3i). This provides a rationale for why identified Leu3 mutations prevented TAK285-induced degradation and provides an explanation as to why SRC WT is refractory to TAK285 treatment unless mutated in positions 3 and 4 (Fig. 3e,h) to mimic BLK's minimal degradation sequence (Fig. 3g).

In conclusion, we found that TAK285 induces a γ-secretase-dependent dissociation of the membrane-associated BLK into the cytoplasm where BLK is intrinsically unstable. TAK285-induced destabilization is encoded by its unique N-terminal domain and critically mediated by its myristoylation status.

## Lysosomal degradation of RIPK2 by RI-4

Finally, we turned our attention to RIPK2, a cytoplasmic kinase that is involved in the clearance of bacterial pathogens by linking activation of the pattern recognition receptors NOD1/NOD2 to intracellular signalling[44]. Degradation of RIPK2, but not inhibition of its kinase activity suppresses the NOD2 signalling pathway, establishing the motivation to identify monovalent and bivalent RIPK2 degraders[45,46]. Indeed, kinome abundance trajectories revealed nine inhibitors potentially destabilizing RIPK2. Among those inhibitors, RI-4 (Fig. 1g) prompted the most selective and potent degradation response (Extended Data Fig. 8a) and was therefore selected for further mechanistic workup. In accordance with RI-4's role as a RIPK2 inhibitor[28], recombinant binding assays confirmed RIPK2 engagement, suggesting a directly induced degradation event (Extended Data Fig. 8b). We validated that RI-4 functions at the level of protein stability (Fig. 4a,b) and confirmed degradation of endogenous RIPK2 (Fig. 4c). Quantitative expression proteomics established RI-4's degradation selectivity (Extended Data Fig. 8c).

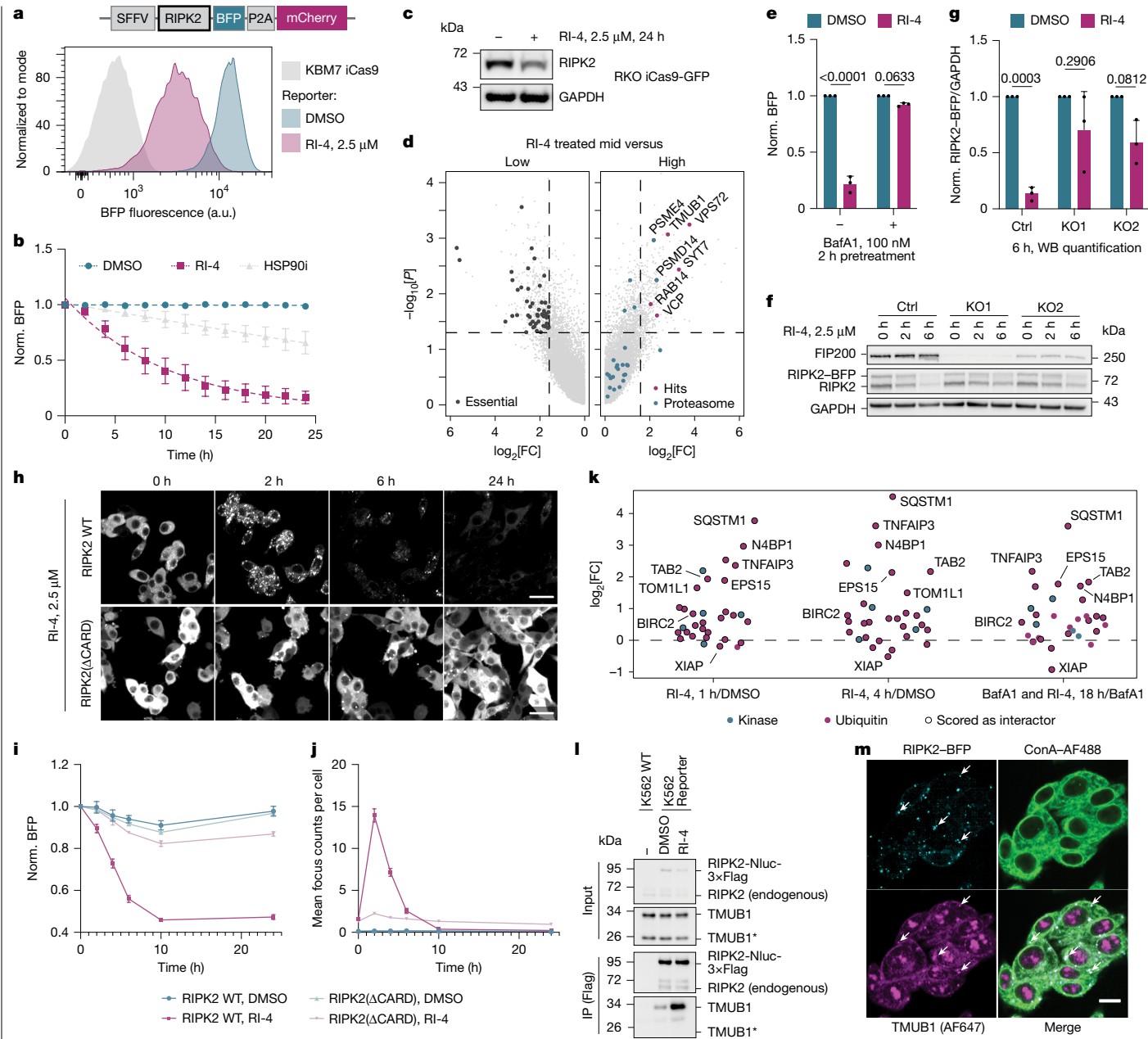

**Fig. 4 | RI-4 destabilizes RIPK2 through TMUB1-facilitated multimerization and macroautophagy. a**, Flow-based stability reporter for RI-4-induced (2.5 μM, 18 h) RIPK2 degradation. **b**, As in **a**, but after time-ranging DMSO, 2.5 μM RI-4 or 10 μM HSP90i treatment. $n = 3$. **c**, Endogenous RIPK2 destabilization in RKO-iCas9 cells (2.5 μM, 24 h). $n = 3$. **d**, Genome-wide FACS based CRISPR–Cas9 screen. Essential genes or hits are highlighted in cases in which $P < 0.05$ (one-sided MAGeCK) and $\log_2[FC] > 1.585$. Proteasome subunits are highlighted irrespective of cut-offs. $n = 2$. **e**, Flow-based RIPK2 stability reporter data after BafA1 pretreatment (100 nM, 2 h) followed by DMSO or RI-4 (2.5 μM, 18 h). $n = 3$. **f**, Immunoblot of RIPK2 stability reporter (RKO-iCas9) expressing a control sgRNA (Ctrl, sgAAVS1) or sgRNAs targeting *FIP200* (KO1/KO2). $n = 3$. **g**, Western blot (WB) quantification of **f**. **h**, Microscopy images (BFP) of RIPK2(FL) or RIPK2(ΔCARD) stability reporters after RI-4 treatment (2.5 μM) in RKO-iCas9 cells. $n = 3$, $m = 2$. Scale bar, 25 μm. Brightness was adjusted per genotype

(equally adjusted images are shown in Supplementary Fig. 8b and DMSO controls are shown in Extended Data Fig. 8h). **i,j**, Quantification of RIPK2 stability (**i**) and the mean number of RIPK2 foci per cell (**j**) as shown in **h**. $n = 3$ and $m = 2$. Data are normalized to $t = 0$ h per condition. **k**, The $\log_2[FC]$ of interactors for 1 h, 4 h RI-4 treatment versus DMSO, or 18 h of RI-4 and BafA1 co-treatment versus BafA1, filtered for GO terms associated with ubiquitin or kinase function. **l**, Flag co-immunoprecipitation in K562 RIPK2-Nluc-3×Flag cells after 2 h DMSO or RI-4 (2.5 μM) treatment (asterisk denotes short isoform). $n = 3$. **m**, Representative immunofluorescence of concanavalin A1 (ConA; green) and TMUB1 (magenta) in RIPK2 stability reporter RKO-iCas9 cells after 2 h RI-4 treatment. $n = 3$. Scale bar, 12.5 μm. For **e** and **g**, statistical analysis was performed using two-way ANOVA with Tukey's test for multiple comparisons. All data are mean ± s.d. $n$ represents biological replicates and $m$ represents technical replicates.

To reveal cellular effectors that are required for RIPK2 degradation, we again ran a CRISPR–Cas9 screen, which revealed enrichment of hits involved in lysosomal degradation (GO Biological Process 2025: endosomal vesicle fusion (GO: 0034058, $P = 0.003409$); phagolysosome assembly (GO: 0001845, $P = 0.003959$)) (Fig. 4d). Moreover,

we identified the ubiquitin-like (UBL) domain containing TMUB1 as strongly enriched[47]. Consistent with the screening data, pharmacological inhibition of lysosome acidification rescued induced RIPK2 degradation (Fig. 4e). Moreover, RIPK2 degradation was abrogated in cells deficient for the macroautophagy mediator FIP200 (ref. 48),

but not after KO of *PSMB5*, supporting that RIPK2 degradation by RI-4 depends on macroautophagy (Fig. 4f,g and Extended Data Fig. 8d,e).

Investigating RI-4-induced changes in RIPK2 abundance using microscopy, we observed that RIPK2 assembled into foci before degradation (Fig. 4h–j). These structures were reminiscent of RIPosomes—physiologically relevant, higher-order RIPK2 assemblies that form after endogenous activation[49]. We confirmed RIPK2 assembly formation by co-transducing the RIPK2-stability reporter (RIPK2-BFP-P2A-mCherry) and an orthogonal RIPK2–GFP reporter, highlighting co-localization after RI-4 treatment (Extended Data Fig. 8f,g). A key feature of physiological RIPK2 assembly is the essential role of RIPK2's CARD domain[50]. Likewise, RI-4-induced RIPK2 assemblies are CARD-domain dependent, therefore implying a functional resemblance (Fig. 4h–j and Extended Data Fig. 8h). Supporting observations made with RIPK2-stability reporters, clearance of RI-4-induced RIPK2 assemblies was rescued through BafA1 treatment, highlighting another similarity to RIPosomes[49] (Extended Data Fig. 8i–k). Moreover, clearance of RI-4-induced RIPosomes is ubiquitin dependent (Extended Data Fig. 8l,m).

To assess interactors of the RI-4-induced RIPosomes and to identify potentially involved E3 ligases, we conducted dynamic BioID profiling (Supplementary Data 5). GO enrichment highlighted ubiquitin and specifically Lys63-dependent ubiquitination in the process of RIPosome clearance (Extended Data Fig. 9a). Stratifying for terms containing ubiquitin or kinase-associated terms, we identified many previously established components associated with RIPosome clearance such as TNFAIP3, CYLD, N4BP1 and CCDC50 (Fig. 4k). Identification of SQSTM1 (p62) substantiated a direct involvement of the autophagy machinery. Moreover, we detected two E3 ligases that scored as interactors: cIAP1 (encoded by *BIRC2*) and XIAP. Both ligases contain an IAP domain and have been previously implicated in turnover and ubiquitin-dependent regulation of RIPK2. KO of both ligases impaired degradation (Extended Data Fig. 9b–d), highlighting their functional and potentially redundant role in RIPK2 degradation. RIPK2(I212D) has been shown to disrupt IAP domain binding[51]. Supporting the functional relevance of cIAP1 and XIAP, RIPK2(I212D) phenocopied the *BIRC2/XIAP* double KO, retaining RI-4's ability to induce multimer formation while RIPosome clearance was impaired (Extended Data Fig. 9e–g).

As we could not detect peptides of TMUB1 in the BioID dataset, we separately set out to dissect its contribution. TMUB1 depletion revealed a marked delay in the assembly formation, a reduction in the total number of observed foci and, consequently, delayed degradation (Extended Data Fig. 9h–j). Immunoprecipitation further highlighted a drug-induced interaction between RIPK2 and TMUB1 (Fig. 4l and Extended Data Fig. 9k), which we confirmed by immunofluorescence staining (Fig. 4m and Extended Data Fig. 9l,m). This implies that TMUB1 is an early-acting facilitator. Taken together, our data support a model in which RI-4 induces higher-order assemblies of RIPK2 through involvement of the UBL-domain-containing protein TMUB1. These assemblies mimic multimers that are formed in response to physiological stimuli by pathogens. Pathogen-induced and inhibitor-induced assemblies are subsequently turned over through macroautophagy.

## Discussion

Analysis of dynamic abundance profiles of 98 kinases after cellular exposure to 1,570 annotated kinase inhibitors revealed that inhibitor-induced kinase degradation is a frequent phenomenon. Known HSP90 clients are enriched among the degraded kinases, suggesting chaperone deprivation as a widespread mechanism of inhibitor-induced degradation. However, in-depth mechanistic investigation of three degraded kinases with graded HSP90 dependency revealed a differentiated, yet shared mechanism of action. In all cases, inhibitor-induced kinase degradation further elevated physiological turnover mechanisms by inducing kinase states that are primed for degradation. Mechanistically, different phenomena can manifest in these unstable states, including altered kinase activity (LYN), localization (BLK) or assembly states (RIPK2).

For the investigated inhibitors, we found that induction of these states can be triggered by direct target engagement, or through network drugging. This highlights that potent off-target degradation can result from on-target inhibition, emphasizing the relevance of unbiased profiling approaches. Indeed, systematic proteomics campaigns have revealed a breadth of proteome-wide effects[52,53]. One of our most unexpected findings is the drug-induced cleavage of the myristoylation anchor of BLK through the γ-secretase, resulting in rapid turnover of BLK in the cytosol. While the direct TAK285 target mediating this effect remains elusive, this outlines the feasibility of precise pharmacological manipulation of protein anchoring to the membrane to control protein function and stability. Disrupting membrane association of disease-relevant proteins is a longstanding challenge. However, to date, it has predominantly been attempted by enzymatic inhibition of promiscuous enzymes, such as farnesyltransferase inhibition, which suffered from limited efficacy due to compensatory lipidation and from toxicities[54,55]. Additional research will be required to further our understanding of the generalizability and mechanistic principles of γ-secretase-dependent target dissociation.

We did not detect a pronounced overlap between kinases that are prone to be destabilized by inhibitors and kinases that are primed for degradation through proximity-inducing modalities, such as PROTACs. Nevertheless, there is evidence that also proximity-inducing molecular glue degraders can be prospectively furnished to re-establish physiological degradation, as exemplified by molecular-glue degraders of mutant β-catenin[56]. Future research will be required to understand the scope of supercharging endogenous degradation events beyond kinases. In addition to BCL6 for which induced multimerization enables degradation through its native E3 ligase SIAH1, certain selective oestrogen receptor degraders destabilize the oestrogen receptor through the E3 ligase UBR5, which is responsible for physiological endoplasmic reticulum turnover after oestradiol exposure[57]. Moreover, a recent study reports directly acting degraders of IDO1 (ref. 58) that function by accelerating IDO1 degradation through the endogenous E3 CRL2[KLHDC3]. Collectively, these studies indicate supercharging of physiological degradation routes as a general mechanism of ligand-induced protein degradation that is complementary to proximity-inducing modalities such as PROTACs or molecular-glue degraders.

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

## Methods

### Cell lines and cell culture

KBM7 cells (obtained from T. Brummelkamp) and KBM7 iCas9 cells (a gift from J. Zuber) were grown in IMDM (Thermo Fisher Scientific) supplemented with 10% heat-inactivated FBS (Sigma-Aldrich) and 1% penicillin–streptomycin (Gibco). RKO iCas9-GFP and iCas9-BFP (gifted by J. Zuber), K562 (purchased from ATCC) and NALM-6 (obtained from A. Villunger) cells were cultured in RPMI 1640 (Thermo Fisher Scientific) supplemented with 10% FBS and 1% penicillin–streptomycin. HEK293T lentiviral packaging cells (obtained from Clontech), HEK293T (purchased from ATCC) and Flp-In T-REx 293 (obtained from Invitrogen) cells were cultured in DMEM (Thermo Fisher Scientific) supplemented with 10% FBS and 1% penicillin–streptomycin.

For competitive Kinobead pull-downs, Jurkat, MCF7, K562, COLO-205 and MV-4-11 cells were cultured in RPMI 1640 medium (Biochrom) supplemented with 10% (v/v) FBS (Biochrom). SK-N-BE(2) cells were grown in DMEM/Ham's F-12 (1:1) supplemented with 10% (v/v) FBS and OVCAR-8 cells were cultured in IMDM medium (Biochrom) supplemented with 10% (v/v) FBS.

Cell lines were cultured at 37 °C and 5% $CO_2$ in a humidified incubator and were regularly tested for mycoplasma contamination.

### Plasmids and cloning

All plasmid preparation, unless specified otherwise, was performed in stable competent *Escherichia coli* (NEB) or, in the case of destination vectors, in One Shot ccdB Survival 2 T1R Competent Cells (Invitrogen) according to the manufacturer's instructions.

The pLEX305-ccdB-Nluc-3×Flag luminescent reporter vector was generated as a destination vector starting from pLEX_305-ccdB-dTAG destination vector (Addgene, 91798) by restriction digest with AgeI and MluI and T4 DNA ligation (NEB) of a synthesized gene block (TWIST) containing the Nluc sequence and a C-terminal 3×Flag tag (5′-CGG GCAAAACCGGTGTCTTCACACTCGAAGATTTCGTTGGGGACTGGCGA CAGACAGCCGGCTACAACCTGGACCAAGTCCTTGAACAGGGAGGTG TGTCCAGTTTGTTTCAGAATCTCGGGGTGTCCGTAACTCCGATCCAA AGGATTGTCCTGAGCGGTGAAAATGGGCTGAAGATCGACATCCATGTC ATCATCCCGTATGAAGGTCTGAGCGGCGACCAAATGGGCCAGATCGA AAAAATTTTTAAGGTGGTGTACCCTGTGGATGATCATCACTTTAAGG TGATCCTGCACTATGGCACACTGGTAATCGACGGGGTTACGCCGAAC ATGATCGACTATTTCGGACGGCCGTATGAAGGCATCGCCGTGTTCGA CGGCAAAAAGATCACTGTAACAGGGACCCTGTGGAACGGCAACAAA ATTATCGACGAGCGCCTGATCAACCCCGACGGCTCCCTGCTGTTCC GAGTAACCATCAACGGAGTGACCGGCTGGCGGCTGTGCGAACGC ATTCTGGCGGACTACAAGGACCACGACGGTGACTACAAGGACCACG ACATCGACTACAAGGACGACGACGACAAGTAGTAAACGCGTTGACGA TGG-3′).

To generate a destabilized version of the vector, an identical gene block was synthesized with the addition of the PEST sequence (Promega) 5′-AATTCTCACGGCTTTCCGCCTGAGGTTGAAGAGCAAGC CGCCGGTACATTGCCTATGTCCTGCGCACAAGAAAGCGGTATGGACC GGCACCCAGCCGCTTGTGCTTCAGCTCGATCAACGTC-3′ upstream of the stop codon. pENTR223 Gateway entry vectors for the kinases were obtained from Hahn/Root Labs Human Kinases ORF Kit[59,60] (Addgene Kit, 1000000014) with the exception of FYN and MAPK4, which were purchased separately (BCCM, LMBP ORF81088-E05, LMBP ORF81100-B12). pRK5-HA-ubiquitin, pRK5-HA-ubiquitin_K48R and pRK5-HA-ubiquitin_K63R plasmids were provided by G. Versteeg. A pENTR221-GFP vector was generated by BP Gateway cloning (Invitrogen), starting from the PCR-amplified GFP sequence of pCAG-GFP[61] (gifted by C. Cepko, Addgene, 11150) and insertion into the empty pDONR221 (Invitrogen, 12536017). Final luminescent reporter vectors were generated by LR Gateway cloning according to the manufacturer's recommendations (Invitrogen, incubation was routinely run overnight at 25 °C before heat inactivation and transformation). The correct

insert size was assessed by analytical digest and in-frame cloning was verified by sequencing (Microsynth).

Single point mutations, with the exception of BLK S5A and S6A, were generated from the respective pENTR223 plasmids using either the Q5 site-directed mutagenesis kit (primers are shown in Supplementary Table 1, method, SDM, NEB) or by Q5 (NEB) PCR amplification (primers are shown in Supplementary Table 1, method, PCR), followed by 1 h of DpnI digest (NEB) and direct transformation into DH5α *E. coli* (NEB).

Stability vectors were generated by digesting the previously published plasmid backbone pRRL_SFFV_empty_BFP_P2A_mCherry[29] with SalI and BamHI, before insertion of the PCR-amplified kinase of interest using the NEBuilder HiFi DNA Assembly Master mix (NEB) according to the manufacturer's instructions. BLK G2I, G2L, L3A, L3G, L3V, V4S, S5A and S6A stability reporter plasmids were generated analogously using the corresponding mutated primer pairs. Vectors for domain-swap experiments, truncated versions of the BLK stability reporter (with the exception of 1–7) and the ABL1(C464W) stability reporter were generated similarly by amplifying the respective DNA sequences from each kinase and performing two- or three-part assemblies. For the aforementioned truncated BLK version 1–7, oligos (Supplementary Table 3) were annealed, phosphorylated and ligated into the corresponding vector. Primers were designed using the NEBuilder Assembly tool (the sequences are provided in Supplementary Table 2).

pRRL_SFFV_CSK* EF1a_iRFP670 and pRRL_SFFV_BLK*_GGGS_3×Flag EF1as_BFP were generated by restriction digest with SalI and XhoI (or BamHI (3×Flag)) of pRRL.SFFV.DACF16.EF1as.iRFP670[29] or pRRL_SFFV_ empty_GGGS_3×Flag EF1as_BFP and insertion of the PCR-amplified CSK/BLK fragments using the NEBuilder HiFi DNA Assembly Master Mix (NEB) (the sequences are provided in Supplementary Table 2).

The gateway vectors pcDNA5_FRT_ccdB_3×Flag_miniTurbo and pSTV6_ccdB_3×Flag_miniTurbo (provided by A. Gingras), as well as pRRL_EF1a_ccdB_emGFP_IRES_HygroR (gifted by G. Superti-Furga) formed the basis for the generation of the remaining kinase reporter vectors via LR gateway cloning (Invitrogen). For the LYN and RIPK2 BioID dataset pENTR223_LYN, pENTR223_RIPK2 or pENTR221_GFP were cloned into pcDNA5_FRT_ccdB_3×Flag_miniTurbo, and, for the BLK BioID dataset, pENTR223-BLK or pENTR223-APH1A (BCCM, LMBP ORF81047-H06) and pENTR221-GFP were cloned into pSTV6_ ccdB_3×Flag_miniTurbo.

sgRNAs were cloned into a single sgRNA vector pLenti-U6-IT-EF1a-Thy1.1-P2A-Neo or dual sgRNA vector pLentiDual-hU6-IT-mU6-IT-EF1a-Thy1.1-P2A-Neo (both gifts from J. Zuber) as previously reported[29]. The sgRNA sequences are described in Supplementary Table 4 and were designed using VBC score[62].

The saturated mutagenesis library for BLK was ordered directly from GenScript and cloned starting from the BLK stability reporter plasmid. No barcoding was applied.

All inserted DNA sequences were verified by Sanger sequencing (Microsynth).

### Cell line generation by lentiviral transduction

With exception of the generation of the LYN and RIPK2 BioID cell lines (see the 'Generation of cell lines through flp recombinase'), all cell lines were generated by transduction of lentivirus. For virus production, HEK293T lentiviral packaging cells were transfected at 70% confluence with the to-be-packaged plasmid in addition to the two packaging plasmids (pCMVR8.74 helper, pMD2.G envelope, both gifted by D. Trono (Addgene, 22036 and 12259)) using polyethylenimine (PEI MAX MW 40000, Polysciences). Viral supernatants were collected 60 h after transfection and cell debris was removed using a 0.45-μm poly-ethersulfone filter.

For transduction, 1 million cells (K562 for luminescent reporters, HEK293T, KBM7, KBM7 iCas9 or RKO iCas9-GFP/BFP for all other reporters) per 2 ml were transduced with 250 μl virus solution and 8 μg ml$^{-1}$ polybrene. If required, the virus volume was adjusted to achieve the

desired transduction efficiency. Then, 24 h after transduction, cells were expanded. For luminescent reporter cell lines, selection was performed with puromycin (1 µg ml$^{-1}$, Gibco) starting 48 h after cell recovery. Subsequently cell pools were subjected to quality control by means of immunoblot analysis using the C-terminal Flag epitope tag, as well as assessment of luminescence levels using NanoGlo Luciferase Assay System (Promega). For the latter, 10$^5$ cells were seeded for each reporter cell pool in 30 µl on a 384-well plate and the luminescence was measured on the Victor X3 2030 Multilabel Reader (Perkin Elmer).

The three cell lines generated for the BLK BioID experiment (performed in KBM7) were likewise selected with puromycin (1 µg ml$^{-1}$, Gibco). Generated cell pools were assessed after cell recovery for construct expression 24 h after doxycycline treatment (1 µg ml$^{-1}$, PanReac AppliChem). Both correct fusion size and biotinylating efficiency were tested by immunoblotting. The latter was conducted by an additional incubation of cells with 100 µM biotin for varying timeframes and blotting for the biotinylated proteome using an anti-biotin antibodies (see the 'Immunoblotting' section).

sgRNA-vector-containing cells were selected using G418/neomycin (1 mg ml$^{-1}$, Gibco) 72 h after transduction. Completion of selection or sgRNA transduction efficiency was assessed by staining with APC anti-mouse Thy1.1 antibody (1:400, 202526, BioLegend) in human TruStain FcX Fc receptor blocking solution (1:1,000, 422302, BioLegend) for 5 min at 4 °C, followed by two PBS washes and subsequent analysis by flow cytometry. Genetic KOs were generated by induction of the tightly inducible Cas9 cassette by doxycycline (0.4 µg ml$^{-1}$, PanReac AppliChem) for a timeframe of 48 h up to 1 week (the incubation times per sgRNA are provided in Supplementary Table 4) before analysis using immunoblotting, imaging or flow cytometry.

All fluorescent reporter cell lines were either used directly for flow cytometry or selected by FACS using the CytoFLEX SRT Benchtop Cell Sorter (CytExpert SRT-Software (v.1.1.0.10007). In the first round of sorting, pools of reporter-positive cells (Supplementary Fig. 2) were enriched. For selected cell lines, single cells were sorted, expanded and used for flow cytometry, FACS-based CRISPR–Cas9 screens or imaging experiments. Specifically, the main stability reporters generated in KBM7 iCas9 for LYN WT, BLK WT and RIPK2 WT were used as clonal cell lines. Moreover, the BLK(WT)–GFP reporters in RKO iCas9-BFP cells were also used as clones. The remainder of the stability reporters generated for imaging purposes in RKO iCas9-GFP cells were used as sorted pools. The KBM7 iCas9 LYN(Y32A) stability reporter was used as a sorted cell pool, whereas the suite of KBM7 iCas9 BLK mutant stability reporters were used without sorting and instead analysis was performed on the reporter-positive cell gate. For the latter, a matched unsorted BLK WT stability reporter was used as a control in the corresponding datasets. All genetic KOs were performed in a pooled format after G418 selection as detailed above.

### Generation of cell lines through flp recombinase
Flp-In T-REx 293 cells were transfected with 200 ng of LYN-mT, RIPK2-mT and GFP-mT plasmids and 2 µg pOG44 vector (Invitrogen, V600520) using Lipofectamine 2000 (Invitrogen) according to the manufacturer's instructions in a 6-well format. Then, 24 h after transfection, cells were expanded to a 10-cm dish and after an additional 24 h, cell selection was initiated with 200 µg ml$^{-1}$ hygromycin B (Roth) and maintained for at least four weeks. Subsequently, expression and biotinylation capacity (as described for the BioID cell lines generated by lentiviral transduction) was performed.

### Temporal luminescence drug screen
Compounds from the kinase inhibitor library including 10 PROTAC controls (see the 'Compounds' section) and respective transcription and translation compounds were dispensed through an Echo 550 system into white 1,536-well plates (PerkinElmer, 6004684) at the appropriate concentrations (0.5–10 µM; Supplementary Data 1; 10 µM for CHX and 1 µM for NVP-2). Plates were sealed and stored at −20 °C. On the day of the drug screen, the plates were equilibrated to room temperature. Next, 5 µl of 1:100 endurazine (Promega) in buffered RPMI (complete RPMI supplemented with 50 mM HEPES pH 7, Sigma-Aldrich, H0887) were prelaid into each well using a liquid dispenser (Thermo Fisher Scientific, Multidrop Combi). Then, 5 µl of cells at a density of 640,000 cells per ml in buffered RPMI were dispensed on top. Cells were transferred to an incubator (humidified chamber, 37 °C, 5% CO$_2$) and the luminescence signal was measured every 4 h from 2 h to 18 h after seeding using the EnVision plate reader (Revvity). Raw luminescence signals were subsequently normalized for intraplate effects. Finally, compound effects were quantified by calculating percentage of control (POC) based on averaged, outlier-corrected DMSO (100%) and positive control (CHIR-99021; 0%) wells, for each plate and timepoint individually (Supplementary Fig. 4a). Only compounds that passed an initial preselection step were used for the final drug screen. The prescreen was performed following the identical steps but only for the two control cell lines GFP-Nluc and dGFP-Nluc and at two concentrations (2.5 µM or 10 µM for 10 mM stock compounds and 0.5 µM and 2 µM for 2 mM stock concentrations). Compounds were eliminated if any of the normalized POC data were smaller than 48 or larger than 150 or if the relative change relative to the 2-h timepoint was bigger than 0.58. In cases in which only the higher concentrations fulfilled these criteria, the corresponding lower concentration was used. In total, 1,620 compounds including 10 PROTACs were used for the drug screen (Supplementary Data 1).

### Data analysis of luminescence drug screen
After the normalization of the initial drug-screening data, we performed additional data processing to obtain a binary active/inactive classification for each compound–kinase pair.

First, we implemented a filter to exclude compound–kinase pairs exhibiting high variability across replicates (s.d. > 30; Fig. 1) consistently across all five timepoints, resulting in the removal of 138 compound–kinase pairs. Overall, the proportion of pairs having 0 timepoints with a high s.d. was 99.67%. We further filtered out compounds that exhibited high reactivity against all kinases, considering them false positives due to their low initial 2-h timepoint (35 compounds with a median POC across kinases <70). We further excluded the three non-small molecules disitertide (TFA), Pep2m myristoylated (TFA) and pm26TGF-β1 (TFA) from our analysis. To ensure comparability of time series and to eliminate bias towards absolute POC values, we centred the compound–kinase series around 100 POC relative to the 2-h timepoint. This centring process was first applied across kinases and then across compounds.

We used the time series of CHX, NVP2 and DMSO controls to assess whether compound–kinase pairs significantly deviated from each control. For each kinase and timepoint, we independently calculated the normalized compound $z$ score. A compound was considered to significantly reduce the kinase readout if it exhibited a substantial reduction (2 sigma) compared with the null model of the controls. We used the same methodology to calculate $z$ scores of compounds concerning the distributions of all other compounds. We also calculated the $z$ scores normalizing against only the initial 2-h timepoint to capture significant changes relative to the initial conditions.

This process resulted in eight normalization schemes: against CHX, NVP2, DMSO and compounds, considering both timepoint-independent and initial-timepoint-dependent situations. Each normalization offered varying selectivity over the compound–kinase time series, and we then expressed the scores as the count of significantly decreased timepoints (2 sigma).

Finally, we conducted a parameter scan to define a query for selecting hit compounds by combining the scores and specifying the minimal number of significantly deviated timepoints for each normalization scheme and the overall total combined through 'or' operators.

We determined the normalization score thresholds for the query by minimizing the false-discovery rate (FDR). This was achieved using the 10 PROTAC controls and their respective kinase targets as a reference for true positives as well as a manually curated inclusion list. The query that reflected our constraints is as follows:

$$[(DMSO\_norm \geq 5)||(CHX\_norm \geq 2)||(CPD\_norm \geq 5)$$
$$||(CHX\_norm2h \geq 5)]||(TOT \geq 10)$$

meaning that all of the positive kinase–compound pairs have to globally score 10 or more, or having a normalized score above the determined threshold in at least one of the individual screens (Supplementary Fig. 4b). For the rare instance of a missing timepoint (mainly associated with the kinase reporters for CDK4, CDK7 and CDK9), the score was corrected by +1. One compound was excluded from further data analysis due to scoring in >10 instances. The final KinDeg scores are shown in Supplementary Data 1 (including the annotation of excluded compounds). The final hit kinase trajectories are shown in Supplementary Fig. 9.

The screening data were used to fit the half-lives of each kinase. This was performed by fitting the equation $100 \times e^{(-x \times tau)}$ in Python (v.3.7.6) and the package scipy (v.1.4.1) to each kinase's CHX screening trajectory. t-SNE plots were generated with sklearn and matplotlib (v.1.0.1 and v.3.5.3, respectively) from ChEMBL drug-binding data processed as described in the Chemical Checker (CC)[24] and compounds were characterized with CC global bioactivity signatures. Chaperone client status was mapped from a previous study[8] to the respective canonical kinases (Supplementary Data 1) and respective pairwise comparisons were calculated using a Fisher's exact test, applying the fisher_exact function from Python's scipy.stats module (v.0.12.2). The JD values between kinase hit profiles were calculated as 1 − Jaccard similarities (JS), where the JS is the size of the intersection divided by the size of the union of two compound (hit) sets. Kinome trees were depicted using http://www.kinhub.org/kinmap/index.html (ref. 63).

To assess whether the data were enriched for type I, II or allosteric inhibitors, we manually annotated our hit compounds for their respective binding mode (Supplementary Data 1) using the available literature data, structural properties of the inhibitors as well as structural data where available. We further used the data available in the PKIDB database[64,65] to annotate the remaining compounds.

Finally, for scoring of the stabilization events, we used a similar approach to the degraders. However, we focused on compounds exhibiting a substantial readout increase (2 sigma) relative to the controls' null models. Importantly, we excluded the CHX normalization scheme from this analysis. Following this query:

$$DMSO\_norm \geq 5||CPD\_norm \geq 5$$

We further excluded compounds that scored in the GFP-Nluc-3×Flag control cell line or that scored in >10 instances. With these boundary conditions, we identified 204 stabilization events across 64 kinases and 128 compounds. The associated data are provided in Supplementary Data 1.

## Immunoblotting

Cell pellets (1–2 million cells per treatment) were lysed in urea lysis buffer (8 M urea, 1% CHAPS, 50 mM Tris-HCL pH 8) for 30 min with shaking at 4 °C and 1,200 rpm. Next, the samples were cleared by centrifugation for 15 min (20,000g, 4 °C) and quantified using the Pierce BCA protein assay kit (Thermo Fisher Scientific) according to the manufacturer's instructions. Finally, the samples were diluted with Bolt LDS sample buffer (4×) (Invitrogen) supplemented with final concentration (f.c.) 10% β-mercaptoethanol (Sigma-Aldrich) and denatured for 10 min at 70 °C. Then, 20 μg per protein sample was separated on the Bolt 4–12% Bis-Tris Plus Gel (10–17 wells) (Invitrogen) using the

Colour Prestained Protein Standard, Broad Range (10–250 kDa, NEB) as a marker. After transfer to a nitrocellulose membrane, membranes were stained by Ponceau-S. Next, the membranes were blocked with 5% milk in TBS-T (30 min, room temperature) and then incubated with primary antibodies overnight at 4 °C in TBS-T. The next day, the membranes were washed three times with TBS-T followed by incubation for 1 h at room temperature with the respective secondary antibodies if required. Finally, the membranes were again washed three times before analysis on the Chemidoc system using Pierce ECL Western Blotting Substrate (Thermo Fisher Scientific). The following antibodies and dilutions were used: GAPDH (1:5,000, Santa Cruz Biotechnology, sc-365062), GAPDH (1:5,000; Santa Cruz Biotechnology, sc-47724), vinculin (1:500; Szabo Scandic, SACSC-25336), Flag (1:2,000; Cell Signaling Technology, 2368), LYN (1:1,000; Cell Signaling Technology, 2796), BLK (1:1,000; Cell Signaling Technology, 3262), RIPK2 (1:1,000; Cell Signaling Technology, 4142S), phosphorylated LYN (Tyr507) (1:1,000; Cell Signaling Technology, 2731), FIP200 (1:1,000; Cell Signaling Technology, 12436), CDK9 (1:1,000; Cell Signaling Technology, 2316), TMUB1 (1:1,000, Abcam, EPR14066), cCBL (1:1,000; Cell Signaling Technology, 2747), phosphorylated LYN (Tyr397) (1:1,000; Cell Signaling Technology, 70926), HRP-conjugated anti-biotin (1:1,000; Cell Signaling Technology, 7075), peroxidase-conjugated goat anti-rabbit IgG (1:10,000; Jackson ImmunoResearch 111-035-003), peroxidase-conjugated goat anti-mouse IgG (1:5,000; Jackson ImmunoResearch, JAC115035003). For quantifications the accompanying ChemiDoc ImageLab software (v2.4.0.03) was used, normalized to the respective loading control and plotted as fold changes with respect to each genotype's DMSO control or 0-h timepoint. The data were plotted as the mean from three independent biological replicates ± s.d. Replicates and uncropped images are shown in Supplementary Fig. 1.

## Compounds

Carfilzomib (Cay17554-5) and BafA1 (Cay11038) were purchased from Cayman, HSP90i (4-(4-(23-dihydro-14-benzodioxin-6-yl)-5-methyl-1H-pyrazol-3-yl)-6-ethylresorcinol), 385920) was obtained from Calbiochem. All other small-molecule inhibitors were sourced from MedChemExpress. These include the kinase inhibitor library (Supplementary Data 1; 1,996 compounds, HY-L009), MLN4924 (HY-70062), TAK-243 (HY-100487), TAK-285 (TAK285, HY-15196), Src inhibitor 3 (SI-3, HY-130254), RI-4 (HY-107978), AV-412 (HY-10346), neratinib (HY-32721), afatinib (HY-10261), WZ4002 (HY-12026), nintedanib (HY-50904), DAPT (HY-13027), alkynyl myristic acid (HY-140335), THAL-SNS-032 (dCDK9, HY-123937), NVP-2 (HY-12214A), dabrafenib (HY-14660), PLX 4720 (HY-51424), ibrutinib (HY-10997), ONO-4059 (HY-18951), R406 (HY-11108), dasatinib (HY-10181), asciminib (HY-104010), DPH (HY-12070) and GNF-2 (HY-11007). Cycloheximide (CHX) was purchased from Cell Signaling Technology (2112S).

All compounds were dissolved in DMSO (Sigma-Aldrich, D1435) as 1 mM, 10 mM, 20 mM, 25 mM or 100 mM stock solutions. Working dilutions were prepared as 1,000× or 2,000× stock solutions. The kinase inhibitor library was delivered as 2 mM or 10 mM stock solutions (Supplementary Data 1).

## Flow cytometry

Cells were treated with the compounds at the concentrations and timeframes indicated in the respective figure legends, and the fluorescent channels of interest were subsequently analysed on a LSR Fortessa (BD Biosciences) using the BD FACSDiva software (v.9.0). The data were analysed using FlowJo (v.10.6.2) as outlined in Supplementary Fig. 2 and the resulting mean BFP and mCherry values were exported for further processing. BFP/mCherry ratios were calculated after background subtraction (from matched WT cells) and normalized to either each pretreatment or genetic variant (referred to as normalized BFP in the figure legends) or normalized to a specific condition as indicated in the respective subscripts, for example, DMSO in Fig. 3c.

Decay functions were fitted using $Y = (Y_0 - \text{plateau}) \times e(-K \times X) +$ plateau and dose responses fitted using $Y = \text{bottom} + (\text{top} - \text{bottom})/(1 + (IC_{50}/X)n)$ where $n$ is the Hill slope using the in-built functions of GraphPad Prism (v.10.0.3) and nonlinear regression fitting. Matched mCherry flow histograms to Figs. 2a and 4a and Extended Data Fig. 6e are shown in Supplementary Figs. 6a, 7a and 8a, respectively.

## FACS-based CRISPR–Cas9 screen

The screens were performed as previously described[29]. First, cells were transduced at an multiplicity of infection (MOI) of 0.1–0.2 with lentivirus containing the respective sgRNA library, prepared as described in the 'Cell line generation by lentiviral transduction' section to achieve a 1,000× representation per sgRNA. For LYN, the previously published UPS-focused sgRNA library[66] (7,801 sgRNAs) and, for BLK and RIPK2, a genome-wide library[62,67] was used. Then, 72 h after transduction, the transduction rate was assessed by staining with APC anti-mouse Thy1.1 antibody (1:400, 202526, BioLegend) and human TruStain FcX Fc receptor blocking solution (1:1,000, 422302, BioLegend) for 5 min at 4 °C. Next, selection with G418 (1 mg ml$^{-1}$, Gibco) was initiated. Cells were maintained in G418-positive medium for at least 14 days, splitting cells every 48–72 h. For the screen, Cas9 expression was induced with doxycycline (0.4 µg ml$^{-1}$, PanReac AppliChem) and, after 72 h, cells were treated with DMSO or the respective inhibitors (SI-3, 156 nM, 8 h; TAK285, 6 h; RI-4, 2.5 µM, 18 h). Cells were centrifuged for 5 min at 500$g$ and stained with APC anti-mouse Thy1.1 antibody (1:400, 202526, BioLegend), Zombie NIR Fixable Viability Dye (1:1,000, BioLegend) and human TruStain FcX Fc receptor blocking solution (1:1,000, 422302, BioLegend) for 5 min at 4 °C. Subsequently cells were fixed with BD fixation buffer 4% (Thermo Fisher Scientific, Pierce) for 45 min at 4 °C followed by two washes with PBS and resuspension in FACS buffer (PBS, 5% FBS and 1 mM EDTA) for storage at 4 °C. All staining steps were performed in the dark. Cells were sorted within 48 h of fixation.

Sorting was performed on a BD FACS Aria Fusion (70-µm nozzle, BD Biosciences, BD FACSDiva software, v.8.0.2). First, cells were strained through a 35-µm nylon mesh. Next, cells were sorted for the 5% highest and lowest BFP-expressing cells as well as 30% of the mid-fraction (the gating strategy is shown in Supplementary Fig. 2). For each replicate and condition, cells corresponding to at least a 500-fold (genome-wide) or 1,000-fold (UPS-focused) library representation were sorted.

After sorting, the high, low and mid fractions were pooled per replicate and lysed overnight (14 h) at 55 °C with shaking at 1,200 rpm in lysis buffer (10 mM Tris-HCl, 150 mM NaCl, 10 mM EDTA, 0.1% SDS) supplemented with proteinase K (New England Biolabs). The next day, RNase was removed with DNase-free RNase (Thermo Fisher Scientific) for 2 h at 37 °C. The lysates were stored at −20 °C until further processing.

For DNA extraction, two rounds of phenol extraction (UltraPure Buffer-Saturated Phenol, Thermo Fisher Scientific, 15513039) using phase Lock Gel tubes (VWR, 7332477) followed by isopropanol precipitation overnight at −20 °C were performed. Next, the samples were barcoded using a two-step PCR protocol (AmpliTaq Gold polymerase, Invitrogen, 4311818). After each PCR step, amplicons were cleaned up with Mag-Bind TotalPure NGS beads (Omega Biotek) according to the the manufacturer's protocol for double-sided selection. Final NGS libraries were pooled at equimolar ratios and sequenced on the HiSeq 3000 or NovaSeq 6000 platform (Illumina).

The resulting reads were trimmed using fastx-toolkit (v.0.0.14) and subsequently aligned (Bowtie2, v.2.4.5) and quantified (featureCounts, v.2.0.1). The corresponding workflows are available at GitHub (https://github.com/ZuberLab/crispr-process-nf/tree/566 f6d46bbcc2a3f49f51bbc96b9820f408ec4a3 and https://github. com/ZuberLab/crispr-mageck-nf/tree/c75a90f670698bfa78bfd8be-786d6e5d6d4fc455). Gene-level enrichment was calculated by comparing each high or low population to the corresponding mid population using the median-normalized read counts. The resulting log$_2$[FC] and $P$ values as well as the number of scoring and total quantified sgRNAs

per gene are provided in Supplementary Data 3. Essential genes were retrieved from DepMap (23Q4)[68].

## FACS-based DMS

The screen was conducted similar to the CRISPR–Cas9 screen, first transducing cells at an MOI of 0.1–0.2 followed by FACS-based enrichment of double-positive cells. Cells were treated before the screen with DMSO, TAK285 (2.5 µM, 6 h) or IMP-1088 (1 µM 24 h). Cells were then fixed (see the 'FACS-based CRISPR–Cas9 screen' section) and sorted for 5% high or low or 30% mid BFP level cells. After DNA extraction, samples were barcoded by two-step PCR with customized primer sets. The samples were finally sequenced using the NovaSeq 6000 platform (Illumina) run in PE150.

For the analysis, we adapted our previously established pipeline[69]. In brief, the raw sequencing reads were converted to fastq format with samtools (v.1.17). Demultiplexing of paired-end reads was performed using cutadapt (v.4.4), matching read 1 5′ barcodes were provided in a separate FASTA file, with no trimming applied (-.action=none). Demultiplexed paired-end FASTQ files were converted to unaligned BAM format using Picard's FastqToSam tool (v.3.0.0) and trimmed using Trim Galore (v.0.6.6) in paired-end mode with Nextera adapter trimming enabled. Short reads were aligned to the BLK unique domain sequence and SAM files were generated using the mem algorithm from the bwa software package (v.0.7.17). The SAM file was converted to BAM format using samtools (v.1.15.1) and mutation calling was performed using the AnalyzeSaturationMutagenesis tool from GATK (v.4.1.8.1). Next, the relative frequencies of variants were calculated for each position and variants that were covered by less than 1 in 30,000 reads were excluded from further analysis. Read counts for each variant were then normalized to the total read counts of each sample and log$_2$[FC] values comparing high/low-to-mid fractions of each condition were calculated. $P$ values were adjusted for multiple testing using the Benjamini–Hochberg procedure to control the FDR. The resulting significant (adjusted $P < 0.05$) log$_2$[FC] low-to-mid and high-to-mid comparisons per condition are provided in Supplementary Data 4. For DMSO-normalized results, finally, log$_2$[FC] values were calculated with respect to the respective DMSO high-to-mid or low-to-mid log$_2$[FC]. Heat maps were generated using the pheatmap (v.1.0.12) package in R (v.4.1.0).

## Immunofluorescence staining

Cells were seeded in PhenoPlate 96-well microplates (Revvity) and subjected to drug treatment after 24 h of pre-attachment. After treatment, cells were fixed with BD Cytofix for 10 min at room temperature. After three PBS washes, cells were permeabilized with 0.2% sodium citrate, 0.1% Triton X-100 for 5 min at room temperature. Next, cells were washed three times with PBS after a 30 min block with BSA (0.024 g ml$^{-1}$) and incubated overnight at 4 °C with the respective antibodies diluted in blocking solution (1:500, TMUB1, Abcam, EPR14066). The next day, cells were washed three times with PBS followed by an incubation for 2 h at room temperature with secondary antibodies (1:500, Alexa-Fluor 647, Cell Signaling Technology, 4414) and 1:1,000 concanavalin A–Alexa Fluor 488 (Thermo Fisher Scientific, C11252). Finally, cells were washed three times and kept in 100 µl PBS. The samples were imaged within the next 24 h.

## High-content confocal imaging and data analysis

Cells were imaged using the PerkinElmer Opera Phenix automated microscope run on the Harmony software (v.4.9 or later) and using the pre-set filter settings for DAPI (BFP), AF-488 (GFP), AF-647 (TMUB1), mCherry and brightfield. Exposure was set to <400 ms per channel. BFP and GFP, as well as AF-647 and mCherry channels were separated during acquisition. Cells were seeded 24 h before imaging into 384-well or 96-well (CellCarrier Ultra, Revvity) plates to achieve a final cell density of 40–60%. Drugs were added immediately before imaging as indicated in the figure legends. All of the experiments were acquired with a

×40 air objective, with exception of the immunofluorescence data, which were acquired with a ×63 water objective.

Cells were segmented using cellpose[70] (0.6.5-foss-2020b) using either the mCherry (RIPK2) or GFP (BLK) channel and an adjusted diameter of 38, 50 or 80, respectively. Next, relevant features and fluorescence were extracted using custom-built cellprofiler pipelines (4.1.3-foss-2020b).

In all instances, ConvertImageToObjects (convert to boolean image (no), preserve original labels (yes)) was used to generate the primary objects. Next, for RIPK2, EnhanceOrSuppressFeatures was applied (Operation = Enhance, Type = Speckles, Size 6, Speed and accuracy = Fast) followed by IdentifyPrimaryObjects (diameter = 2-20, thresholding strategy = global, method = manual, threshold = 0.0016, smoothing scale 1.3488, method clumped objects&draw lines between clumped objects = Intensity, automatic smoothing and distance calculation enabled, holes filled in after both thresholding and declumping). RelateObjects was applied to assign the resulting speckles per cell object. Finally, MeasureObjectIntensity and MeasureObjectSizeShape were applied for measuring the respective parameters across the speckles and cell objects, before exporting the data to a database for further processing through self-written Python scripts. For the RIPK2 data associated with Fig. 4h–j and Extended Data Fig. 8h, due to the different absolute BFP fluorescence values of the constructs for RIPK2 WT and RIPK2(ΔCARD), two steps were added before EnhanceOrSuppressFeatures. Namely, ExpandOrShrinkObjects was applied to eliminate cell boundaries (Operation = shrink by a specified number of pixels, pixels = 4) followed by ImageMath, which was used to calculate the BFP to mCherry ratio. The thresholds in IdentifyPrimaryObjects were thus adapted to 0.2 instead of 0.0016. For RIPK2(I212D) and RIPK2 WT stability reporter data, an additional step of prefiltering cells with less than 0.01 mCherry signal was added before segmentation of the speckles. In the TAK243 dataset and the extended KO data for *XIAP* and *BIRC2*, the threshold for IdentifyPrimaryObjects was adjusted to 0.0024 and 0.002 respective to the total BFP signal per acquired dataset.

For the immunofluorescence staining and GFP co-localization experiment, a similar approach as above was conducted. After primary object identification, RIPK2 foci were again identified using EnhanceOrSuppressFeatures (Operation = Enhance, Type = Speckles, Size 6, Speed and accuracy = Fast) followed by IdentifyPrimaryObjects (diameter = 2-20, thresholding strategy = global, method = manual, threshold = 0.0015, smoothing scale 1.3488, method clumped objects&draw lines between clumped objects = Intensity, automatic smoothing and distance calculation enabled, holes filled in after both thresholding and declumping). For TMUB1, IdentifyPrimaryObjects was applied with a threshold of 0.035 and, for GFP, a threshold of 0.015 was used. Each speckle was first assigned to a corresponding cell using the RelateObjects function, followed by the RelateObjects function run on the foci per condition.

For the BLK–GFP cell clones, only the module MeasureObjectIntensity was applied after object classification. Corresponding data were exported to a spreadsheet for further processing.

In all instances, Python (v.3.7.6) was used to annotate the resulting data (condition, replicate) and normalize the data. Normalized data were then exported and depicted in GraphPad Prism (v.10.0.3). In all cases, data were averaged per biological replicate of the mean values per cell. The s.d. was correspondingly calculated across the biological replicates.

## NanoBRET

The assay was performed as described previously[71]. In brief, full-length CSK and LYN were obtained as plasmids cloned in frame with an N-terminal Nluc-fusion (gift from Promega). Plasmids were transfected into HEK293T cells using FuGENE HD (Promega, E2312), and proteins were allowed to express for 20 h. Serially diluted inhibitor and NanoBRET K4 Tracer (Promega, TracerDB: T000037) at the Tracer KD concentration taken from TracerDB[72] were pipetted into white 384-well plates (Greiner 781207) using an ECHO acoustic dispenser (Labcyte).

The transfected cells were added and reseeded at a density of $2 \times 10^5$ cells per ml after trypsinization and resuspending in Opti-MEM without Phenol Red (Life Technologies). The system was allowed to equilibrate for 3 h (37 °C, 5% CO$_2$) before the bioluminescence resonance energy transfer (BRET) measurements. To measure BRET, NanoBRET NanoGlo Substrate and extracellular Nluc Inhibitor (Promega, N2540) were added according to the manufacturer's protocol, and filtered luminescence was measured on the PHERAstar plate reader (BMG Labtech) equipped with a luminescence filter pair (450 nm BP filter (donor) and 610 nm LP filter (acceptor)). Competitive displacement data were then analysed using GraphPad Prism (v.10.0.3) software using a normalized three-parameter curve fit with the following equation: $Y = 100/(1 + 10(X - \log[IC_{50}]))$.

## Commercial recombinant binding/inhibitory assays

In vitro kinase inhibitory or kinase binding assays were performed using the SelectScreen platform (Thermo Fisher Scientific). TAK285 (BLK) and SI-3 (LYN) were screened using the Z′-LYTE assay, while RI-4 (RIPK2) was screened using the LanthaScreen Eu Kinase Binding Assay according to their respective assay availability. Threefold dilutions were performed starting from 30 μM and in presence of ATP, using its standard apparent $K_M$ per kinase.

## MST binding assay

Protein purification was performed as previously described[73]. Purified γ-secretase complex, modified γ-secretase complex (PS1 fused to GFP) and full-length BLK protein (fused to GFP) were diluted in buffer containing 25 mM HEPES pH 7.4, 150 mM NaCl and 0.1% (w/v) Digitonin.

TAK285 serial dilutions were mixed with purified γ-secretase (GFP-tagged). The mixture was loaded onto MO-K022 capillaries at room temperature. Microscale thermophoresis (MST) analyses were conducted on the Monolith NT.115 (NanoTemper) system with 20% LED power and 60% MST power. The MST data were analysed using MO.Affinity Analysis (v.2.3).

Characterization of the γ-secretase and BLK interaction was performed using GFP-tagged BLK (with or without 20 μM TAK285) as the target protein and addition of serially diluted untagged γ-secretase. Samples were measured as described above.

## Immunoprecipitation

Cell pellets (10 million cells) were lysed in 900 μl IP lysis buffer (50 mM Tris-HCL (pH 7.4), 150 mM sodium chloride, 0.1% Triton X-100, 1 mM EDTA and 5 mM magnesium chloride, 1× protease inhibitors) followed by lysate clearance, protein quantification and immunoprecipitation as described in the 'Immunoprecipitation, on-bead TAMRA click and in-gel fluorescence' section. After immunoprecipitation and sample washing, proteins were then directly eluted using 70 μl as final volume before analysis using immunoblotting. For blocking, 5% BSA in TBS-T was used instead of 5% milk in TBS-T and the phosphorylated LYN Tyr507 or phosphorylated LYN Tyr397 antibody was diluted 1:1,000 in TBS-T containing 3% BSA and 0.1% sodium azide.

## Immunoprecipitation, on-bead TAMRA click and in-gel fluorescence

Cell pellets (15 million cells per condition) were lysed in 900 μl NP40 lysis buffer (DPBS with 1.5 mM magnesium chloride, 1% NP40, 1× protease inhibitors, 1× benzonase) for 30 min on ice. The lysates were cleared by centrifugation (20 min, 4 °C, 20,000g) and quantified using the Pierce BCA Protein Assay Kit (Thermo Fisher Scientific) according to the manufacturer's instructions. The samples were then normalized to 1 mg per input and preactivated anti-Flag magnetic beads (Sigma-Aldrich) were added, followed by the incubation for 3 h at 4 °C on a rotating wheel. The beads were washed three times with lysis buffer. After removal of the supernatant, 56 μl of click-mix (170 μM TAMRA (5-TAMRA-Azide, CLK-FA008, Jena Biosciences), 230 μM copper sulfate, THPTA 1.15 mM, HCl 5 mM, sodium ascorbate 5 mM, in PBS)

were added per sample. Finally, 18 µl of elution buffer (4× Laemmli buffer supplemented with f.c. 10% β-mercaptoethanol) were added and the samples were boiled at 95 °C for 10 min before loading 20 µl of supernatant and analysis using SDS–PAGE. Before the transfer for immunoblotting and its analysis (see section 'Immunoblotting'), the SDS–PAGE gel was imaged on the ChemiDoc system using the Alexa 546 channel and Alexa 647 for the ladder.

## Ubiquitination assays

HEK293T LYN-Nluc-3×Flag cells were seeded into 10-cm culture dishes to reach around 70% confluency on the day of transfection. Transfections were performed using Lipofectamine 2000 (Thermo Fisher Scientific) according to the manufacturer's protocol. In brief, 30 µl of Lipofectamine 2000 was diluted in 720 µl Opti-MEM (Thermo Fisher Scientific) and mixed gently. In parallel, 12 µg of plasmid DNA (pRK5-HA-Ubiquitin, pRK5-HA-ubiquitin(K48R) or pRK5-HA-ubiquitin(K63R)) was diluted in 720 µl Opti-MEM. Both solutions were incubated separately at room temperature for 5 min, combined and incubated for 10 min. The resulting solution was then added dropwise to the cells. Cells were split the next day and subjected to treatments 72 h after transfection. After the treatments, cells were collected with ice cold PBS and, after an additional wash with ice-cold PBS, snap-frozen on dry ice. Cell pellets were lysed in 1 ml of lysis buffer containing 50 mM Tris-HCl (pH 7.4), 150 mM NaCl, 0.1% Triton X-100, 1 mM EDTA, 5 mM $MgCl_2$, 5% glycerol, freshly added protease inhibitors (Thermo Fisher Scientific) and Benzonase nuclease (Sigma-Aldrich).

For Flag pull-down assays, 500 µg of clarified lysate was incubated with 25 µl of anti-Flag magnetic beads (Sigma-Aldrich) for 3 h at 4 °C on a rotating wheel. The beads were washed three times with lysis buffer and bound proteins were eluted by boiling at 95 °C for 10 min in 70 µl of lysis buffer supplemented with 4× SDS sample buffer.

## Data plotting and statistical analysis

All data are represented as the mean of technical or biological replicates ± s.d. or ±confidence intervals. Datapoints were calculated as described in the respective sections.

Imaging data, all data related to the drug screen, proteomics, CRISPR screen, as well as in vitro kinase binding/inhibitory assay were plotted with seaborn (v.0.12.2) and matplotlib (v.3.4.2) in Python (v.3.7.6). Standard packages such as numpy (v.1.21.5), pandas (v.1.0.1) and scipy (v.1.4.1) were correspondingly used for data handling, processing, normalization, statistical calculations and/or data fitting. Immunoblot quantifications, MST results and data associated with flow cytometry (except for flow histograms) were plotted in GraphPad Prism (v.10.0.3). Statistical tests and data fitting for the corresponding datasets were calculated directly with in-build functions as detailed in the respective sections. Flow histograms were exported from FlowJo (v.10.6.2). Representative images of microscopy experiments were prepared using Fiji (ImageJ, v.2.1.1/1.53i).

## Preparation of BioID MS samples

Bait-mT expression was induced 24 h before initiating cell treatments using 1 µg ml⁻¹ doxycycline. The next day, the respective inhibitors (RI-4 2.5 µM, TAK285, 2.5 µM or SI-3, 156 nM) or vehicle control (DMSO, across all cell lines including the two GFP-mT versions) and 100 µM biotin (Sigma-Aldrich, B4501) were added to the cells for 1 h. For BLK-mT, an additional condition was generated including 2 h carfilzomib (1 µM) pretreatment before TAK285 addition. For RIPK2-mT, additional conditions were generated including the co-treatment with BafA1 (100 nM). After the treatments, 20 million cells per condition and replicate were collected by centrifugation followed by two washes in ice-cold PBS. The resulting cell pellets were snap-frozen on dry ice and stored at −80 °C until further processing.

The following protocol was adapted from a previous study[74]. All steps were carried out with Protein LoBind tubes (Eppendorf) and HPLC-grade reagents. In brief, for lysis, cell pellets were resuspended in 250 µl lysis buffer (PBS supplemented with 1% SDS (Sigma-Aldrich, 71736), 2 mM magnesium chloride (Invitrogen, AM9530G), protease inhibitors (Thermo Fisher Scientific, 78437) and benzonase (Merck, US170746-3)). The samples were vortexed and incubated at 37 °C (300 rpm, 30 min) followed by centrifugation at 18,000g (4 °C, 30 min). The supernatant was transferred to fresh tubes and the protein concentration was measured using the Pierce 660 nm protein assay reagent (Thermo Fisher Scientific, 22660) according to the manufacturer's instructions. Per sample, 1 mg of total protein was diluted up to a final volume of 300 µl with lysis buffer. Next, 30 µl of 50 mM TCEP (Sigma-Aldrich, 75259, diluted in $H_2O$) was added, the samples were vortexed and incubated for 1 h at 56 °C with shaking at 300 rpm. Then, 80 µl of 1 M HEPES (pH 7.5, AppliChem, A6916) was added, followed by 45 µl of 200 mM iodoacetamide (Sigma-Aldrich, I1149). The samples were again vortexed and incubated at 25 °C and 300 rpm. Pierce Streptavidin Agarose (Thermo Fisher Scientific, 20353) resin was prepared by centrifugation for 30 s followed by two PBS washes. Next the protein samples were added and incubated on a rotator for 1 h at room temperature in the dark. Finally, the samples were washed twice using 1× pre-washed Mini BioSpin columns (Bio-Rad, 7326207) with wash buffer 1 (0.2% SDS in 1× PBS), followed by 16 washes with wash buffer 2 (8 M urea in 1× PBS) and four washes with PBS. For elution, the slurry was resuspended in 2× digestion buffer (50 mM ammonium bicarbonate, 200 mM guanidine hydrochloride, 1 mM calcium chloride, in $H_2O$) and transferred again to a fresh tube. Subsequently, the supernatant was removed and 250 µl digestion buffer and freshly supplied 10 µl of trypsin solution (0.1 µg µl⁻¹, Promega, V5117) were added, before incubation overnight on a rotating wheel (14 h).

The next day, the beads were centrifuged briefly (30 s) and the supernatant was transferred into a fresh tube. Resin was washed once using 200 µl $H_2O$, which was added to the already separated supernatant. Peptides were cleaned up with self-made stage tips columns. These were prepared from 1 mm circles of an Epore C18 disc inserted into a 200-µl tip. On top of the C18 disc, 24 µl Oligo R3 solution (15 mg ml⁻¹ in acetonitrile (ACN)) was added before 1 min of centrifugation (1,000g). The column was then washed twice with 100 µl ACN (1,000g for 1 min) and equilibrated twice with 200 µl 0.1% TFA (3 min at 1,000g). The samples were acidified with 30% TFA (1% final concentration) before loading of the samples in two fractions onto the column (1,000g for 3 min). One wash with 200 µl 0.1% TFA (3 min at 1,000g) was followed by a double elution step using 50 µl elution buffer (90% ACN, 0.01% TFA, in $H_2O$) each. The eluted peptides were dried using a vacuum centrifuge (45 °C) and stored at −20 °C. Next, the samples were TMT-labelled with the TMTpro 18-plex Label Reagent Set (Thermo Fisher Scientific, A52045) according to the manufacturer's instructions. Subsequently, labelled peptides were pooled and fractionated using on-tip high-pH fractionation. Then, 1 ml of 20 mM ammonium formate (pH 10) was added per 320 µl of pooled sample and added again to self-made C18 columns, prepared as stated above except for the final wash steps, which were performed with 200 µl of 20 mM ammonium formate pH 10 instead of ACN. The samples were loaded in fractions of 250 µl followed by a wash with 200 µl of 20 mM ammonium formate pH 10. Centrifugation at each step was carried out for 3 min and 1,000g. Elution was carried out in five fractions (2 min at 1,000g) with buffers containing 20 mM ammonium formate (pH 10) and different percentages of ACN (16%, 20%, 24%, 28%, 80%). First, 50 µl was used per respective buffer, followed by 20 µl per buffer (2 min 1,000g each). Next, all fractions were dried using a vacuum centrifuge at 45 °C and the resulting dried peptides were stored at −20 °C until data acquisition.

## Sample preparation for full-proteome profiling

Per condition, 20 million cells were lysed in 300 µl lysis buffer (50 mM HEPES, pH 8 supplemented with 1 mM PMSF, protease inhibitor cocktail (Sigma-Aldrich) and 2% SDS). Cells were homogenized by pipetting

and incubated at room temperature for 20 min. Next, the samples were sonicated (Covaris S2 high-performance ultrasonicator) for 150 s. The lysates were clarified by centrifugation at 20,000g for 5 min at room temperature. Extracted protein amounts were determined by BCA (Pierce BCA Protein Assay, 23227). For each sample, 200 μg (K562) or 100 μg (NALM-6) of protein was digested using a filter-aided sample preparation (FASP) protocol essentially according to published procedures[75].

In brief, proteins were reduced by addition of DTT (final concentration 83.3 mM), followed by incubation at 95 °C for 5 min. After cooling the samples to room temperature, the samples were mixed with 200 μl freshly prepared 8 M urea in 100 mM Tris-HCl at pH 8.5 (UA-buffer) and added onto FASP filter units (Merck Millipore). For buffer-exchange, the samples were centrifuged at 14,000g for 15 min at 20 °C and residual SDS was washed by an additional washing step with 200 μl UA-buffer. All of the subsequent centrifugation steps were done at 14,000g for 15 min at 20 °C. Proteins were alkylated by addition of iodoacetamide (50 mM final concentration) and incubated for 30 min at room temperature in the dark. The samples were washed three times with 100 μl UA-buffer followed by three washes with 100 μl TEAB buffer (Sigma-Aldrich). Proteins were digested by addition of sequencing-grade trypsin at a ratio of 1:50 at 37 °C overnight.

To collect peptides, 50 μl of 50 mM TEAB buffer was added and samples were centrifuged. Filters were additionally washed with 50 μl of 0.5 M NaCl and the flowthroughs of both washing steps were pooled. Peptides were cleaned-up by C18 with peptide desalting spin-columns (Thermo Fisher Scientific). The peptides of each condition were labelled with TMTpro 18plex reagents (K562) or TMTpro 6plex (NALM-6) according to the manufacturer's instructions (Thermo Fisher Scientific). After 1 h of labelling, 1 μl of each channel was pooled together, quenched and cleaned-up by C18 and concentrated under reduced pressure. This test mix was measured by data-dependent acquisition (DDA) in the Orbitrap for both MS1 and MS2. Quantification was performed at the MS2 level. The test mix was used to calculate the median signal intensity of each TMTpro channel. The ratios to the lowest median channel intensity were derived and all channels were normalized to equalize the labelling efficiency. The pooled channels were quenched, and the samples were cleaned up by C18. As an additional quality control of channel normalization, another test pool was injected. After pooling all of the samples, an aliquot of 100 μl corresponding to roughly 450 μg was cleaned up by C18 and resuspended in 10 mM ammonium formate buffer pH 10. Peptides were separated on an C18 reversed-phase column (150 × 2.0 mm Gemini-NX, 3 μm C18 110 Å, Phenomenex) by liquid chromatography (LC) into 96 time-window-based fractions operating at 50 μl min⁻¹ constant flow rate. A total of 36 fractions were collected, using a previously described pooling strategy[76]. The samples were fractionated into glass vials with 5 μl 30% TFA to acidify samples after fractionation. The fractions were dried under reduced pressure and reconstituted in 0.1% TFA for MS analysis. Additional information with regard to the reagents is provided in Supplementary Table 5.

## LC–MS/MS data acquisition of BioID and full proteome samples

MS data were acquired on the Orbitrap Fusion Lumos Tribrid mass spectrometer (Thermo Fisher Scientific) coupled to the Dionex Ultimate 3000 RSLCnano system (Thermo Fisher Scientific) interfaced with the Nanospray Flex Ion Source (Thermo Fisher Scientific). Peptides were loaded on a trap column (PepMap 100 C18, 5 μm, 5 × 0.3 mm, Thermo Fisher Scientific) at a constant flow rate of 10 μl min⁻¹ with 0.1% TFA in HPLC-grade H₂O.

Next, the trap column was switched in-line, and peptides were separated on an analytical column (50 cm, 75 mm inner diameter) in-house packed with ReproSil-Pur 120 C18-AQ, 3 μm (Dr. Maisch HPLC) fitted to an ESI emitter fused silica (20 μm inner diameter × 7 cm length × 365 μm outer diameter; orifice inner diameter, 10 μm; CoAnn Technologies) kept at 50 °C. For the analysis, an analytical gradient of 190 min operated at a constant flow rate of 230 nl min⁻¹ was used. The HPLC was operated with buffer A (0.4% formic acid in HPLC-grade H₂O), and buffer B (0.4% formic acid in ACN).

The analytical gradient comprised the following steps: 0–4 min, constant 6% buffer B; 4–5 min, from 6 to 9% buffer B; 5–146 min, increase to 30% buffer B; 146–154 min, increase to 65% buffer B; and a flush at 100% buffer B. The column was re-equilibrated at 6% buffer B from 167–190 min. The samples were acquired in DDA mode using a maximum of ten dependent scans (TopN approach) with synchronous precursor selection (SPS) enabled. Peptides were ionized by applying a constant voltage of 1.8 kV. MS1 precursor survey scans for MS2 and MS3 levels were acquired with scan range of 400–1,600 $m/z$ and a resolution of 120,000 (at 200 $m/z$) in the Orbitrap. The automatic gain control (AGC) was set to 'standard' with a maximum injection time of 50 ms. Precursor ions were filtered by charge state (2–5) excluding undetermined charge states with a dynamic exclusion (60 s with a ±10 ppm window), and monoisotopic precursor selection. The MS1 precursor intensity threshold was set to $5.0 × 10^3$. For MS data analysis, a charge-state filter was used to select precursors for data-dependent scanning. In MS2 analysis, spectra were obtained using one charge state per branch (from $z = 2$ to $z = 5$) in a dual-pressure linear ion trap (ITMS2). Ions were isolated using a quadrupole isolation window with an isolation window of ±0.7. Fragmentation was achieved by collision-induced dissociation (CID) with a fixed normalized collision energy of 35% and an CID activation time of 10 ms. For MS2 scans, the normalized AGC target was set to 200% with a maximum injection time of 35 ms. For MS3 scans, precursor ions were isolated using SPS waveform with varying isolation windows for charge stats: 1.3 $m/z$ for $z = 2$, 1.2 $m/z$ for $z = 3$, 0.8 $m/z$ for $z = 4$ and 0.7 $m/z$ for $z = 5$. Fragment ions were further fragmented by high-energy collision-induced dissociation (HCD) at a fixed activation energy at 45% collision energy. The AGC target was set to 300% with a maximum injection time of 100 ms. The Orbitrap scan range was set to 100–500 $m/z$ at a resolution of 50,000. Xcalibur v.4.3.73.11 and Tune v.3.4.3072.18 were used to operate the instrument.

## Processing of BioID raw MS-injections

MS-raw files were processed with the Proteome Discoverer software (PD, Thermo Fisher Scientific, v.2.4.1.15). For the LYN dataset, a subset of 9 TMT-channels (126, 127N, 127C, 132C, 133N, 133C, 134N, 134C, 135N) were used, whereas, for the RIPK2 experiment, and the BLK and APH1A experiment the full channel set was processed. The three BioID datasets were processed independently.

The peptide identification search was performed using Sequest HT, searching for fully tryptic peptides with a maximum of two missed cleavages and a minimum peptide length of 6 and a maximum of 144 amino acids. The precursor mass tolerance was set to 10 ppm and fragment ion mass tolerance was restricted to 0.6 Da. Spectra were searched against the canonical human protein database obtained from UniProtKB (download 5 November 2021, 20,304 sequences) appended with an in-house-generated list of common laboratory contaminants (298 sequences) and streptavidin. As variable modification methionine oxidation (+15.994 Da), deamidation (0.984 Da), phosphorylation on serine, threonine and tyrosine (+79.966 Da), N-terminal specific acetylation (+42.011 Da), methionine loss (−131.040 Da) and acetylation with methionine loss (−89.030 Da) with a maximum number of three variable modification of the same type per peptide. Carbamidomethylation (+57.021 Da) of cysteine residues and TMT 18-plex labelling of peptide N termini and lysine residues (+304.207 Da) were used as static modifications. PSM and peptide FDR were controlled by Percolator at 1% respectively. The obtained results were filtered to include only spectrum matches with a Sequest HT cross-correlation factor (Xcorr) larger or equal to 0.9. Phosphosites needed a minimum site-probability of 75, corresponding to the high-confidence threshold. For protein abundance inference, only high-confidence proteotypic peptides were included.

Protein and peptide intensities were derived from TMTpro reporter ion intensities. The reporter abundances were based on signal-to-noise (S/N) values if applicable, otherwise reporter ion intensities were used. Correction of isotopic impurities was enabled. A co-isolation threshold for isolation interference of precursors was set to maximum 80%. Moreover, to remove noisy signals, an average TMTpro reporter ion S/N threshold smaller or equal to 10 was used with an additional SPS mass matches threshold of 65%, removing peptides with strong interferences. The obtained data were normalized using the sum total peptide amount and scaled to the average. For normalization and to derive protein abundances, all quantified peptides were used. Protein ratios and log₂[FC] values were directly calculated from the grouped protein abundances, without missing value imputation. Abundance changes were tested for their significance using ANOVA on individual proteins across biological triplicates. P values were corrected for multiple testing using the Benjamini–Hochberg procedure.

### Data analysis and representation of BioID data
The protein-level PD output was used for further analysis. For the LYN BioID experiments, 4,325 UniProtKB accessions were identified; for RIPK2 BioID experiments, 3,962 accessions were identified; and, for BLK, APH1A BioID experiments, 2,962 UniProtKB accessions were found. From the datasets, proteins flagged as contaminates and proteins without quantification values were removed, resulting in 3,564 UniProtKB accessions for the LYN dataset, 3,334 UniProtKB accessions for the RIPK2 dataset and 1,870 UniProtKB accessions for the BLK, APH1A dataset. For subsequent data analysis, the PD-derived normalized intensities, log₂[FC] and the adjusted P value (Benjamini–Hochberg corrected) were used. For the LYN experiment, the analysis focused on significantly changed proteins after SI-3 treatment versus the vehicle/baseline control (DMSO). To identify enriched proximity interactions, a combined threshold of a log₂[FC] ≥ 2 and an adjusted P ≤ 0.01 against GFP controls were used. The same thresholds were used to identify differentially changed interactions in the LYN SI-3-treated samples against LYN DMSO-treated control. For the RIPK2 and BLK/APH1A datasets, additional scoring of proximity interaction partners was performed using SAINTq[77]. For this, the total sum normalized protein intensities per replicate and condition were grouped together and scored against GFP-negative controls (DMSO). SAINTq was performed on protein level (parameters: normalise_control = false, input_level = protein, compress_n_ctrl = 3, and max score across bait replicates). For BLK/APH1A, proteins with a log₂[FC] ≥ 0.5, a SAINTq-score ≥ 0.99 and a BFDR ≤ 0.01 were considered to be high-confidence proximity interaction partners. For the RIPK2 dataset, a more stringent log₂[FC] cut-off of log₂[FC] ≥ 1 was used. As a further filter, the CRAPome[78] frequency was mapped to each prey protein, using for BLK/APH1A a 10% frequency and for RIPK2 a 20% frequency threshold. Bait proteins were excluded from the CRAPome filter. Moreover, for BLK/APH1A, prey proteins that were annotated as kinases and type I transmembrane proteins in UniProtKB were filtered. These annotated interactors were further filtered for significant changes against GFP negative controls, using the adjusted P value (Benjamini–Hochberg corrected) from the ANOVA hypothesis test performed within PD. The obtained proximity interactors were intersected between BLK and APH1A, revealing on one hand bait-specific and on the other hand shared preys. For the RIPK2 dataset, protein interactions for each condition (RI-4, 1 h; RI-4, 4 h; and BafA1, 18 h) were used as the input for gene set enrichment analysis for GO molecular function (2023) terms using Enrichr. All proteins covered in significantly enriched terms (adjusted P ≤ 0.05) were subset from the comparison of treated versus control (RI-4, 1 h and 4 h versus DMSO; and RI-4 + BafA1, 18 h versus BafA1). All interactions found in at least two conditions versus the GFP negative controls were selected for further visualization. The subset of obtained proximity interactors were grouped into broader molecular function terms. For visualization the log₂[FC] and adjusted P value against treatment controls (DMSO or

BafA1, respectively) were used. Data analysis and visualizations were generated employing the statistical software R (v.4.3.1). The resulting processed datasets are provided in Supplementary Data 5. Normalization results and additional individual volcano plots or scatter plots of SAINTq results are provided in Supplementary Fig. 6d–f (LYN), Supplementary Fig. 7b–d (BLK/APH1A) and Supplementary Fig. 8c–e (RIPK2).

### Processing and data analysis of full proteome profiling data
The full proteome datasets were processed in Proteome Discoverer v.2.4.1.15, deriving protein intensities using the TMTpro 18 or TMT 6-plex reporter ion quantities.

Peptide identification search was performed using Sequest HT searching for fully tryptic peptides of a minimum of 6 to up to 144 amino acids length and allowing for a maximum of 2 missed cleavage sites. Precursor mass tolerance was set to 10 ppm and fragment ion mass tolerance was restricted to 0.6 Da. The search was performed against the canonical human protein database obtained from UniProtKB (download 12 November 2020) appended with an in-house-generated list of common laboratory contaminants and streptavidin.

As variable modification methionine oxidation (+15.994 Da) and N-terminal specific acetylation (+42.011 Da), methionine loss (−131.040 Da) and acetylation with methionine loss (−89.030 Da) were set. The maximum number variable modification of the same type was limited to 3. As a static modification, carbamidomethylation (+57.021 Da) of cysteine residues and tandem mass tag (TMT) 18-plex/6-plex labelling of peptide N termini and lysine residues (+304.207 Da) were set. PSM and peptide FDR were controlled with Percolator at 1% respectively. Obtained results were filtered to include spectrum matches with a Sequest HT cross-correlation factor (Xcorr) ≥ 1 and strict Percolator target FDR filters. For further analysis, only peptides scored with high confidence and proteins identified with at least 1 proteotypic peptide were used. Protein and peptide intensities were derived from TMTpro reporter ion intensities. The reporter abundances were based on S/N values if applicable, otherwise reporter ion intensities were used. Correction of isotopic impurities was enabled. Co-isolation threshold of 70% for isolation interference of precursors was used. Moreover, to remove noisy signals, an average TMTpro reporter ion S/N threshold of ≤10 was used. A SPS mass match threshold of at least 65% was applied. For reporter-ion-based quantification, unique and razored peptides were considered. The obtained data were normalized using the sum total peptide amount. For normalization and to derive protein abundances, all quantified peptides per protein were used. Protein ratios and log₂[FC] values were calculated from the grouped protein abundances, without missing value imputation. To test for differentially abundant proteins, ANOVA for individual proteins across all biological replicates (n = 3) was performed. For further analysis of degradation selectivity in K562, the 7,665 protein groups with a high confidence score and quantitative values were used. For the NALM-6 cells, the 7,437 protein groups were used for further analysis. For each drug-treated condition the log₂[FC] and P values were derived against DMSO/baseline control conditions. Depending on the duration of the treatment, either the 8- or the 18-h negative control was used. The resulting processed datasets are provided in Supplementary Data 2.

### Kinobead profiling
Cells were lysed in 0.8% IGEPAL, 50 mM Tris-HCl pH 7.5, 5% glycerol, 1.5 mM magnesium chloride, 150 mM sodium chloride, 1 mM sodium orthovanadate, 25 mM sodium fluoride, 1 mM DTT, protease inhibitors (SigmaFast, Sigma-Aldrich) and phosphatase inhibitors (prepared in-house according to phosphatase inhibitor cocktail 1, 2 and 3 from Sigma-Aldrich). The cell lysate mixes used for compound profiling were generated either from COLO-205, K562, SK-N-BE(2), MV-4-11 and OVCAR-8 cell lysates (standard 5 CL (cell line) mix) or Jurkat and MCF7 cells mixed at equivalent ratios; the protein concentration was determined using the Bradford assay.

Kinobeads pull-down experiments were performed as previously described[79]. In brief, 2.5 mg of the cell lysate mixture was pre-incubated with increasing compound concentrations (DMSO, 3 nM, 10 nM, 30 nM, 100 nM, 300 nM, 1 μM, 3 μM, 30 μM) for 45 min at 4 °C in an end-over-end shaker in either of the two lysate mixes. Next, the lysates were incubated with Kinobeads (17 μl settled beads) for 30 min at 4 °C. The beads were washed and bound proteins were reduced with 50 mM DTT in 8 M urea, 40 mM Tris HCl (pH 7.4) for 30 min at room temperature. After alkylation with 55 mM CAA, proteins were digested with trypsin overnight at 37 °C. Peptides were desalted using C18 StageTips and dried down in a SpeedVac. Peptides were analysed using LC–MS/MS on the Dionex Ultimate3000 nano HPLC system coupled online to an Orbitrap Fusion Lumos (Thermo Fisher Scientific) mass spectrometer. Peptides were delivered to a trap column (100 μm × 2 cm, packed in-house with Reprosil-Gold C18 ODS-3.5 μm resin, Dr. Maisch, Ammerbuch) and washed at a flow rate of 5 μl min$^{-1}$ in solvent A (0.1% formic acid, 5% DMSO in HPLC-grade water). Peptides were then separated on an analytical column (75 μm × 40 cm, packed in house with Reprosil-Gold C18 3 μm resin, Dr. Maisch) using a 52-min gradient ranging from 4 to 32% solvent B (0.1% formic acid, 5% DMSO in ACN) in solvent A at a flow rate of 300 nl min$^{-1}$. The mass spectrometer was operated in a data-dependent mode, automatically switching between MS1 and MS2 spectra. MS1 spectra were acquired over a $m/z$ range of 360–1,300 m/z at a resolution of 60,000 in the Orbitrap using a maximum injection time of 50 ms and an AGC target value of $4 \times 10^5$. Up to 12 peptide precursors were isolated (isolation width of 1.7 Th, maximum injection time of 75 ms, AGC value of $5 \times 10^4$), fragmented by HCD using 30% normalized collision energy and analysed in the Orbitrap at a resolution of 15,000. The dynamic exclusion duration of fragmented precursor ions was set to 30 s.

Peptide and protein identification and quantification was performed using MaxQuant (v.1.5.3.30) by searching the tandem MS data against all canonical protein sequences as annotated in the UniProtKB reference database using the embedded search engine Andromeda. Carbamidomethylated cysteine was set as a fixed modification and phosphorylation of serine, threonine and tyrosine, oxidation of methionine and N-terminal protein acetylation as variable modifications. Trypsin/P was specified as the proteolytic enzyme and up to two missed cleavages were allowed. The minimum length of amino acids was set to seven and all data were adjusted to 1% PSM and 1% protein FDR. LFQ and match between runs were enabled within MaxQuant.

For the Kinobeads competition binding assays, protein intensities were normalized to the respective DMSO control and IC$_{50}$ and EC$_{50}$ values were deduced by a four-parameter log-logistic regression using an internal pipeline that uses the drc package[79] in R. An apparent dissociation constant ($K_{d,app}$) was calculated by multiplying the estimated EC$_{50}$ by a protein-dependent correction faction. The correction factor of a protein is defined as the ratio of the amount of protein captured from two consecutive pull-downs of the same DMSO control lysate. Targets of the compounds are annotated manually. A protein is considered a target if the resulting binding curve shows a sigmoidal curve shape with a dose-dependent decrease of binding to the beads. Moreover, the number of unique peptides and MSMS counts per condition as well as the protein intensity in the DMSO control are taken into account. The resulting fitted parameters in addition to the normalized intensities are provided in Supplementary Data 6.

## TAK285 chemoproteomics

To generate the TAK285 affinity matrix, the terminally amine-tethered TAK285 probe (synthesis is described in the Supplementary Methods) was immobilized to Sepharose beads as previously described[80].

For the competition assay, NALM-6 cell lysates were prepared as previously described[80]. The protein amount of cell lysates was determined using the BCA assay and adjusted to an Igepal concentration of 0.4% and protein concentration of 5 mg ml$^{-1}$ by diluting with Igepal-reduced lysis buffer. The cell lysate was pre-incubated with different doses of TAK285 or the DMSO vehicle control for 45 min at 4 °C on a shaker, followed by incubation with 18 μl TAK285 affinity matrix for 30 min at 4 °C on a shaker. The beads were washed (once with 1 ml of lysis buffer without protease inhibitors and with only 0.4% Igepal, twice with 1 ml of lysis buffer without protease inhibitors and with only 0.2% Igepal, three times with 1 ml of lysis buffer without protease inhibitors and without Igepal), and the captured proteins were denatured with 8 M urea buffer, alkylated with 55 mM iodoacetamide and digested with trypsin according to standard procedures.

The resulting peptides were desalted by StageTip desalting[81]. To construct a StageTip, five C18 discs were packed into a 200 μl pipette tip. The StageTips were activated with 200 μl ACN (all centrifugation steps at 250$g$), washed with 200 μl buffer B (0.1% formic acid in 50% ACN) and equilibrated with 200 μl buffer A (0.1% formic acid in double-distilled H$_2$O). The peptide samples were acidified to a final concentration of around 0.3% formic acid (pH > 2) and loaded on to StageTips. The loading step was repeated with the flow-through. Peptides attached to the C18 material were washed twice with 200 μl buffer A and eluted by adding twice 40 μl of buffer B and collecting the flow-through. The eluent was vacuum-dried and stored at −20 °C until LC–MS/MS measurement.

## LC–MS/MS measurement of the TAK285 competition assay

For proteomic data acquisition, a nanoflow LC–ESI-MS/MS setup, comprising a Dionex Ultimate 3000 RSLCnano system coupled to a Fusion Lumos mass spectrometer (both Thermo Fisher Scientific), was used in positive ionization mode. MS data acquisition was performed in DDA mode. For proteome analyses, half of the competition pull-down peptides were delivered to a trap column (Acclaim PepMap 100 C18, 3 μm, 5 × 0.3 mm, Thermo Fisher Scientific) at a flow rate of 5 μl min$^{-1}$ in HPLC-grade water with 0.1% (v/v) TFA. After 10 min of loading, peptides were transferred to an analytical column (ReproSil Pur C18-AQ, 3 μm, Dr. Maisch, 500 mm × 75 μm, self-packed) and separated using a stepped gradient from minute 11 at 4% solvent B (0.4% (v/v) formic acid in 90% ACN) to minute 61 at 24% solvent B and minute 81 at 36% solvent B at a 300 nl min$^{-1}$ flow rate. The nano-LC solvent A was 0.4% (v/v) formic acid HPLC-grade water.

MS1 spectra were recorded at a resolution of 60,000 using an AGC target value of $4 \times 10^5$ and a maximum injection time of 50 ms. The cycle time was set to 2 s. Only precursors with charge state 2 to 6 that fall in a mass range between 360 to 1,300 Da were selected and dynamic exclusion of 30 s was enabled. Peptide fragmentation was performed using HCD and a normalized collision energy of 30%. The precursor isolation window width was set to 1.3 $m/z$. MS2 spectra were acquired at a resolution of 30,000 with an AGC target value of $5 \times 10^4$ and a maximum injection time of 54 ms.

## Data analysis of the TAK285 competition assay

Protein identification and quantification was performed using Max-Quant (v.2.4.9.0) by searching the LC–MS/MS data against all canonical protein sequences as annotated in the Swiss-Prot reference database (downloaded April 2024) using the embedded search engine Andromeda. Carbamidomethylated cysteine was set as fixed modification and oxidation of methionine and amino-terminal protein acetylation as variable modifications. Trypsin/P was specified as the proteolytic enzyme, and up to two missed cleavage sites were allowed. Precursor tolerance was set to 10 ppm, and fragment ion tolerance was set to 20 ppm. The minimum length of amino acids was set to seven, and all data were adjusted to 1% peptide spectrum matches and 1% protein FDR. LFQ[82] and match between runs was enabled.

To search the proteomics data for dose-dependently competed proteins, we submitted the data to the CurveCurator pipeline[83]. This tool automatically calculates protein LFQ intensities at each competition concentration relative to the DMSO control, plots dose–response curves and applies customized statistics for calling proteins

dose-dependently regulated. The associated data are provided in Supplementary Data 7.

## BLK γ-secretase complex prediction and molecular dynamics simulations

The BLK−γ-secretase complex was predicted using AlphaFold3, with template information enabled for γ-secretase[43]. BLK was modelled starting at Gly2. The complex was prepared using CHARMM-GUI Membrane Builder[84]: the structure was automatically oriented using the PPM 2.0 method and inserted into a POPC bilayer using the replacement method. An N-terminal myristoylation was added at Gly2 during the setup process. The system was solvated with TIP3P water and neutralized with 0.15 M NaCl. GROMACS (v.2023.2) input files were generated according to CHARMM-GUI's standard protocol, comprising energy minimization (step 6.0), six-step equilibration (steps 6.1–6.6), and production dynamics (step 7), which were extended to 50 ns. Simulations were repeated in triplicate with different initial velocities. MM/GBSA binding free-energy estimates were computed, and interface contacts were analysed using GetContacts (https://getcontacts.github.io/). Depictions were generated with VMD (v.1.9.4).

## Reporting summary

Further information on research design is available in the Nature Portfolio Reporting Summary linked to this article.

## Data availability

Data associated with the drug screening such as hit scores as well as data associated with compounds or kinases are provided in Supplementary Data 1. Drug screening data have been deposited online (https://science.aithyra.at/KinDegData). Additional CRISPR−Cas9 screening data generated in the revision process have been deposited alongside. All processed sequencing and proteomics data are provided in Supplementary Data 2–7. Moreover, the proteomics data have been deposited to the ProteomeXchange Consortium via the PRIDE partner repository under dataset identifiers PXD062184 (in vivo biotinylation experiments), PXD053130 and PXD059599 (full proteome profiling) and PXD064676 (TAK285 chemoproteomics). Human protein fasta files were retrieved from UniProtKB (Taxonomic identified 9606, status reviewed, downloaded on 1 December 2019 or 29 April 2024; https://www.uniprot.org/) and have been deposited alongside the respective MS data. The Kinobeads data have been deposited at the ProteomeXchange Consortium via the MASSIVE partner repository under data set identifier MSV000095265 alongside with the used human protein fasta files (UniProtKB, Taxonomic identified 9606, status reviewed, downloaded on the 22 March 2016).

## Code availability

All analysis was performed with previously published analysis pipelines or was performed using standard data analysis processing and associated packages as described in the Methods.

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

**Acknowledgements** We thank the current and former members of the Winter laboratory, in particular M. Cigler for discussions and editorial contributions, as well as M. Schulmann for assistance in DNA preps; the staff at the Core Facility Flow Cytometry of the Medical University of Vienna for access to flow cytometry instruments and assistance with flow cytometry cell sorting as well as the CeMM Biomedical Sequencing Facility for NGS sample sequencing; T. Hannich of the MDP ProMet-facility at CeMM for technical assistance with proteomics data acquisition; J. Zuber at the Research Institute of Molecular Pathology for sharing iCas9 cell lines and plasmids; A.-C. Gingras for providing the BioID plasmids; G. Versteeg for providing the ubiquitin plasmids; S. Martens and M. Gyrd-Hansen for discussions. CeMM, the Winter laboratory and the Superti-Furga laboratory are supported by the Austrian Academy of Sciences; AITHYRA by the Austrian Academy of Sciences as well as the Boehringer Ingelheim Stiftung. The Winter laboratory is further supported by funding from the European Research Council (ERC) under the European Union's Horizon 2020 research and innovation programme (grant agreement 851478), as well as by funding from the Austrian Science Fund (FWF, projects P7909, P36746 and P5918723) and the Vienna Science and Technology Fund (WWTF, project LS21-015); N.S.S. by the FWF postdoctoral Esprit fellowship ESP 426 and Marie Skłodowska-Curie postdoctoral fellowship (grant agreement no. 101029199); S.L. by the EMBO postdoctoral fellowship (ALTF 236-2024); M.M.O. by a PhD fellowship from the Boehringer Ingelheim Fonds (BIF); and P.A. by the Generalitat de Catalunya (RIS3CAT Emergents VEIS: 001-P-001647 and 2021 SGR 00876) and the Spanish Ministerio de Ciencia, Innovación y Universidades (PID2020-119535RB-I00). A.C.-C. is a recipient of an FI fellowship (2020 FI_B 00094). S. Knapp and M.P.S. acknowledge the support by the Structural Genomics Consortium (SGC), a registered charity (no. 1097737) that receives funds from Bayer, Boehringer Ingelheim, Bristol Myers Squibb, Genentech, Genome Canada through Ontario Genomics Institute, EU/EFPIA/OICR/McGill/KTH/Diamond Innovative Medicines Initiative 2 Joint Undertaking (EUbOPEN grant 875510), Janssen, Pfizer and Takeda. S. Knapp is also supported by the German Cancer Research Center DKTK, the German Cancer Aid project TACTIC and the Frankfurt Cancer Institute (FCI). M.P.S. is funded by the Deutsche Forschungsgemeinschaft (DFG, German Research Foundation), CRC1430 (project ID 424228829).

**Author contributions** N.S.S. and G.E.W. conceptualized the study and wrote the manuscript with input from all of the authors. N.S.S. designed and performed most of the described experiments, data analysis and generated the figures. M.B. and A.C.-C. generated the drug screen scoring scheme. P.A. supervised the corresponding data analysis. K.K. performed experiments, quantified immunoblots and handled BioID experiments. X.G. performed MST experiments. F.F. assisted with the BioID experiments, performed the data analysis and generated the respective figure panels. A.R. and I.S. handled expression proteomics sample preparation and processing. M.A. oversaw MS experiments. E.B. and H.R. performed experiments. B.L.S. handled the AlphaFold3 structure predictions and associated molecular dynamics simulations. A.K. assisted with the drug screen, and performed the initial data processing and normalization. M.H. and C.S. supported CRISPR–Cas9 screens. M.P.S. performed NanoBRET experiments. F.B. generated the TAK285-tethered analogue. S.R. performed hit annotation. S.L. performed the TAK285 chemoproteomics and data analysis. P.P. performed and analysed the Kinobead profiling. H.I., C.S. and M.M.i.O. analysed sequencing data. S. Kubicek supervised the drug screen. S. Knapp

oversaw the NanoBRET assays. N.S.G. gave critical input for the manuscript. Y.S. oversaw MST experiments. G.S.-F. and B.K. supervised the proteomics experiments. G.E.W. has overall responsibility for the presented study.

**Competing interests** S. Kubicek, G.S.-F. and G.E.W. are scientific founders and shareholders of Proxygen and Solgate and shareholders in Cellgate Therapeutics. G.E.W. is on the scientific advisory board of Nexo Therapeutics. The G.E.W. and G.S.-F. laboratories received research funding from Pfizer. B.K. is a founder and shareholder of OmicScouts and MSAID. He has no operational role in either company. N.S.G. is a founder, scientific advisory board member and equity holder in Syros, C4, Allorion, Lighthorse, Voronoi, Inception, Matchpoint, CobroVentures, GSK, Shenandoah (board member), Larkspur (board member) and Soltego (board member).

The Gray laboratory receives or has received research funding from Novartis, Takeda, Astellas, Taiho, Jansen, Kinogen, Arbella, Deerfield, Springworks, Interline and Sanofi. The other authors declare no competing interests.

**Additional information**

**Correspondence and requests for materials** should be addressed to Georg E. Winter.

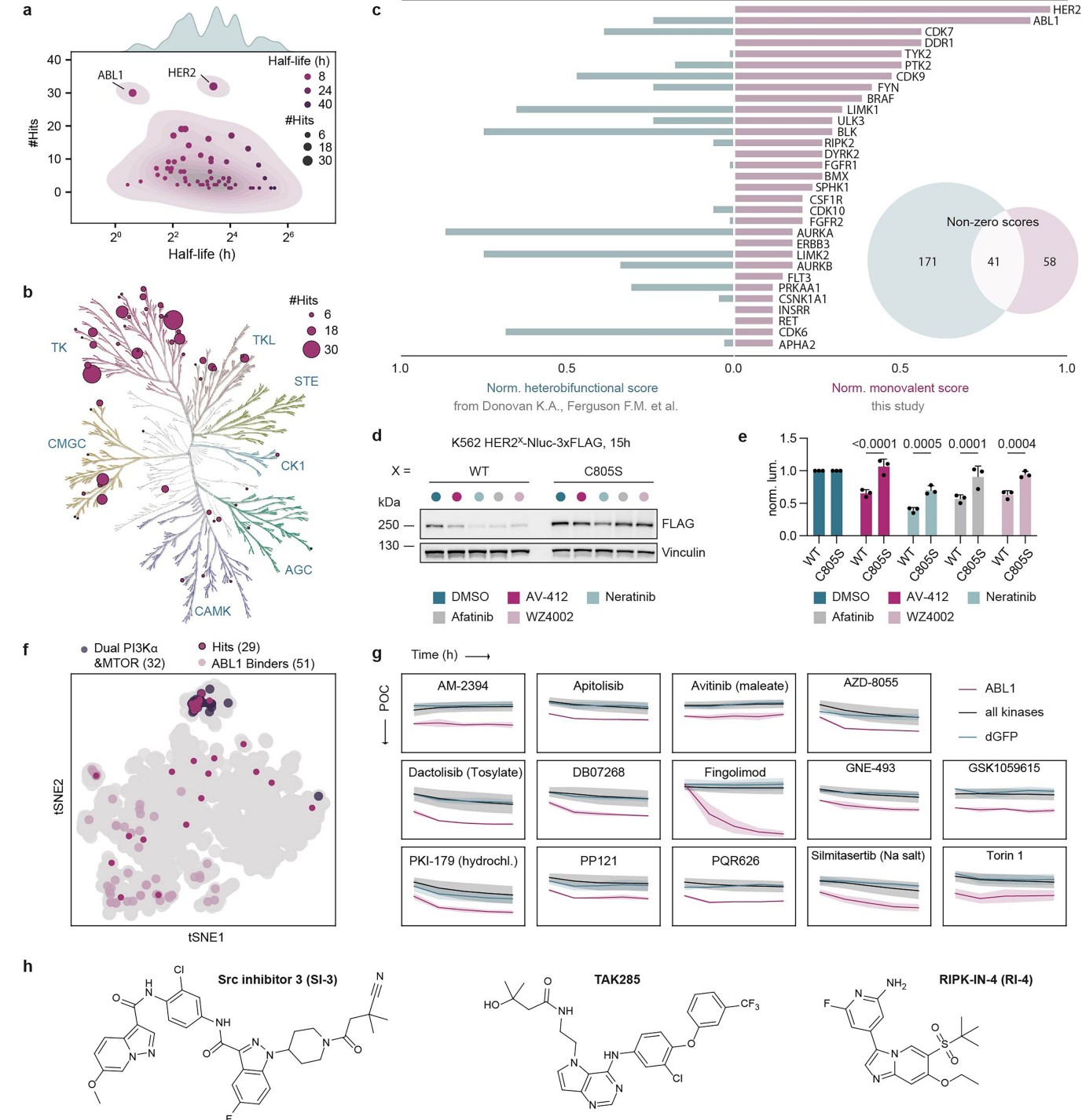

**Extended Data Fig. 1 | Neither half-life nor heterobifunctional degradability determine monovalent destabilization frequencies. a** Comparison of half-life per each kinase fitted from CHX screening data plotted against the summed hit scores. Top: Half-life KDE plot with adjusted bw value of 0.5. Middle: 2D KDE plot and scatter plot of number of hits (#Hits = score) and half-life (h). **b** Summed destabilization score (#Hits) per canonical kinase mapped onto the kinome tree (Illustration reproduced courtesy of Cell Signaling Technology, Inc. (www.cellsignal.com)). **c** Comparison of monovalent degrader scores to previously reported scoring frequencies by heterobifunctional degrader molecules (heterobifunctional score) adapted from Donovan, et al.[18]. The data was normalized to the highest scoring kinase per study and depicts the sorted top 32 downregulated kinases by monovalent small molecules. In total, 41 kinases were detected to be downregulated in at least one instance for

both studies. **d** Immunoblot analysis of cell lines in (e) (n = 3). **e** Luminescent reporter assay of K562 HER2$^{WT}$ or HER2$^{C805S}$ Nluc-3xFLAG reporter cell lines treated for 15 h with the indicated compounds (all 10 μM except for AV-412 (2.5 μM)) shown as normalized luminescence per genetic construct (two-way ANOVA, Sidak corrected) (n = 3). **f** tSNE plot of compound target landscape focusing on ABL1 hits in comparison to annotated ABL1 binders or dual PI3Kα and MTOR binders. **g** Drug screening data comparing ABL1 to the mean of all other kinases and the dGFP control. Continued in SI Fig S5. (m = 2, error bars correspond to CI for individual trajectories and SD for the mean of all kinases). **h** Chemical structures of the three selected examples for mechanism of action elucidation. All data shown as mean of the replicates ± SD unless specified otherwise; n = biological, m = technical replicates.

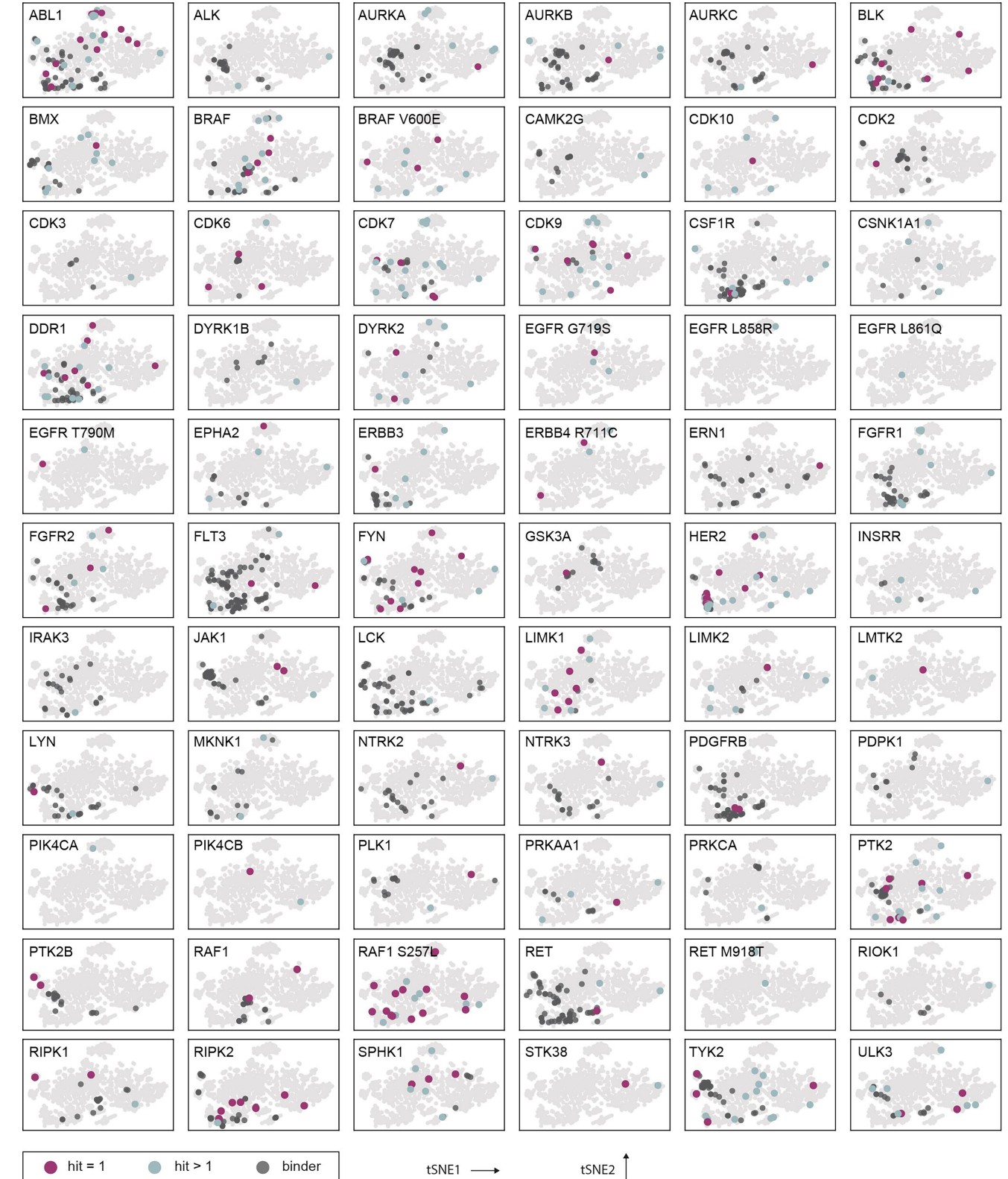

**Extended Data Fig. 2 | Hit and drug binding profile depicted as tSNE.** Overlay of the target compound space of all screened inhibitors as annotated in ChEMBL (light grey, see Supplementary Data 1) with all identified hits and annotated kinase binders per kinase shown in the indicated colour code.

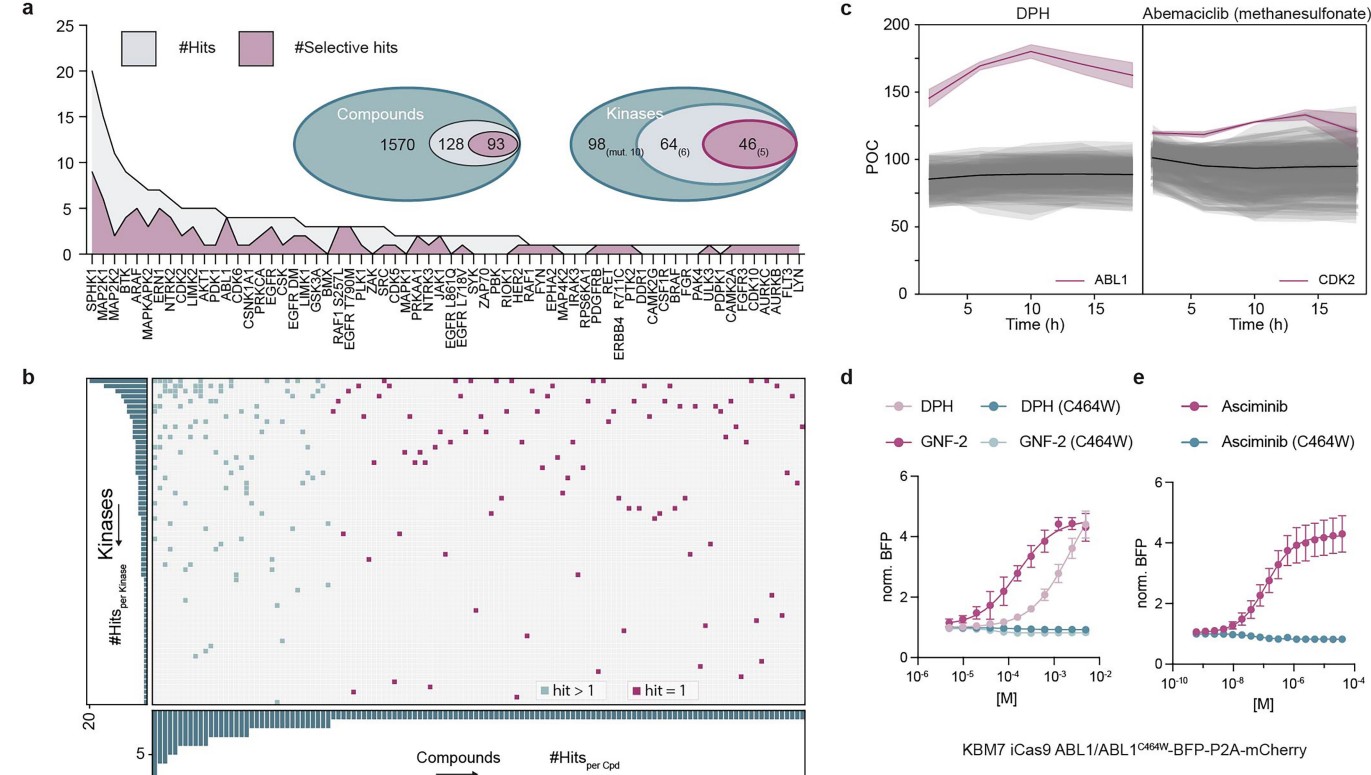

**Extended Data Fig. 3 | Global assessment of inhibitor induced kinase stabilization. a** Breakdown of hit scores across kinases and compounds. **b** Resulting binary kinase stabilization map sorted according to the observed stabilization frequencies across kinases and compounds including adjacent histograms of the summed scores in both dimensions. **c** Example trajectories of screening data for DPH stabilizing ABL1 selectively, as well as Abemaciclib stabilizing CDK2 selectively (m = 2, error bars correspond to CI for individual trajectories and SD for the mean of all kinases). **d** KBM7 iCas9 stability reporter validation for dose-dependent stabilization using DPH and GNF-2 for either ABL1^WT or ABL1^C464W constructs measured by flow cytometry (n = 3). **e** Same as in (d) shown for Asciminib. All data shown as mean of the replicates ± SD unless specified otherwise; n = biological, m = technical replicates.

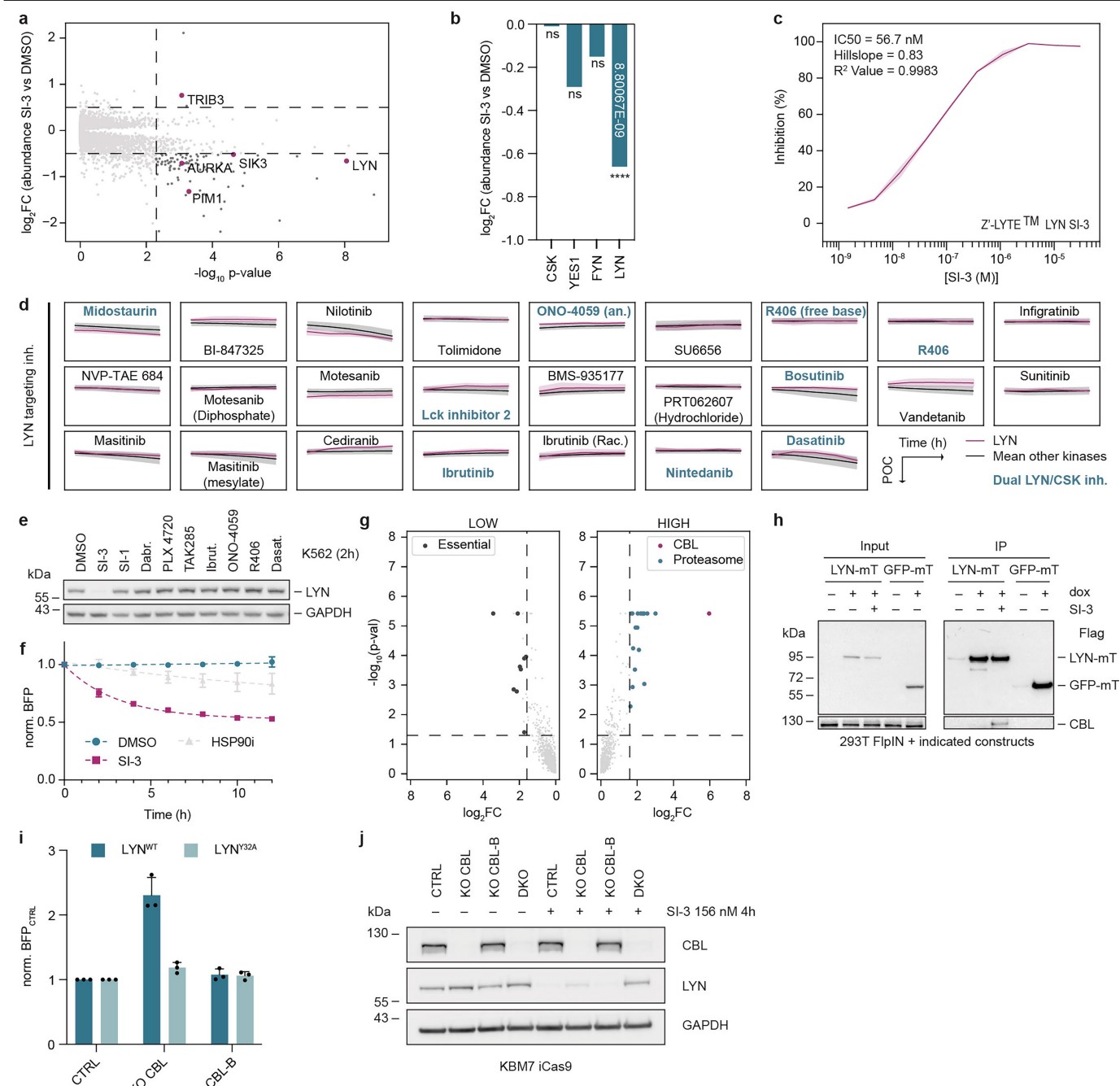

**Extended Data Fig. 4 | SI-3 selectively destabilizes LYN via a dual phosphodegron and dual E3 mechanism of action. a** Expression proteomics of K562 cells after 2 h of SI-3 treatment (156 nM) (One-way ANOVA), (n = 3). **b** Highlighted Src kinase family members and CSK abundance changes including p-values for the expression proteomics experiment shown in (a). **c** Z-LYTE' SI-3 Lyn inhibitory assay (m = 2, error bars = CI). **d** Drug screen trajectories of compounds annotated as LYN binders (Supplementary Data 1) depicted for the LYN reporter (pink) and compared to the mean response of all other kinase reporters (black). Compounds that are annotated to bind both LYN and CSK (ChEMBL) have further been highlighted in green. Axis annotations: y-limits (POC) = 0-150 and x-limits (Time (h)) = 0-20; LYN error bars = CI and all other trajectories are shown as SD. **e** Immunoblot of LYN endogenous protein levels

for 2 h drug treatments across a panel of compounds (2.5 µM) reported to engage LYN and CSK (n = 3). **f** Temporal destabilization profile of LYN stability reporter for DMSO, SI-3 (156 nM) or HSP90i (10 µM) (n = 3). **g** UPS-focused CRISPR/Cas9 screen data for vehicle (DMSO) treated cells (one-sided MAGeCK) (n = 2). **h** Immunoblot of representative enrichment for the respective BioID conditions (Fig. 2f) (n = 3). **i** Genetic k.o. (CTRL = sgAVVS1, KO CBL or KO CBL-B) of LYN^WT or LYN^Y32A stability reporter cell lines measured by flow cytometry. Depicted are baseline normalized values to each respective CTRL sample (n = 3). **j** Immunoblot for KO of CBL, CBL-B or DKO monitoring endogenous LYN levels upon SI-3 treatment (n = 3). n = biological replicates, m = technical replicates, all data shown as mean of replicates ± SD unless specified otherwise.

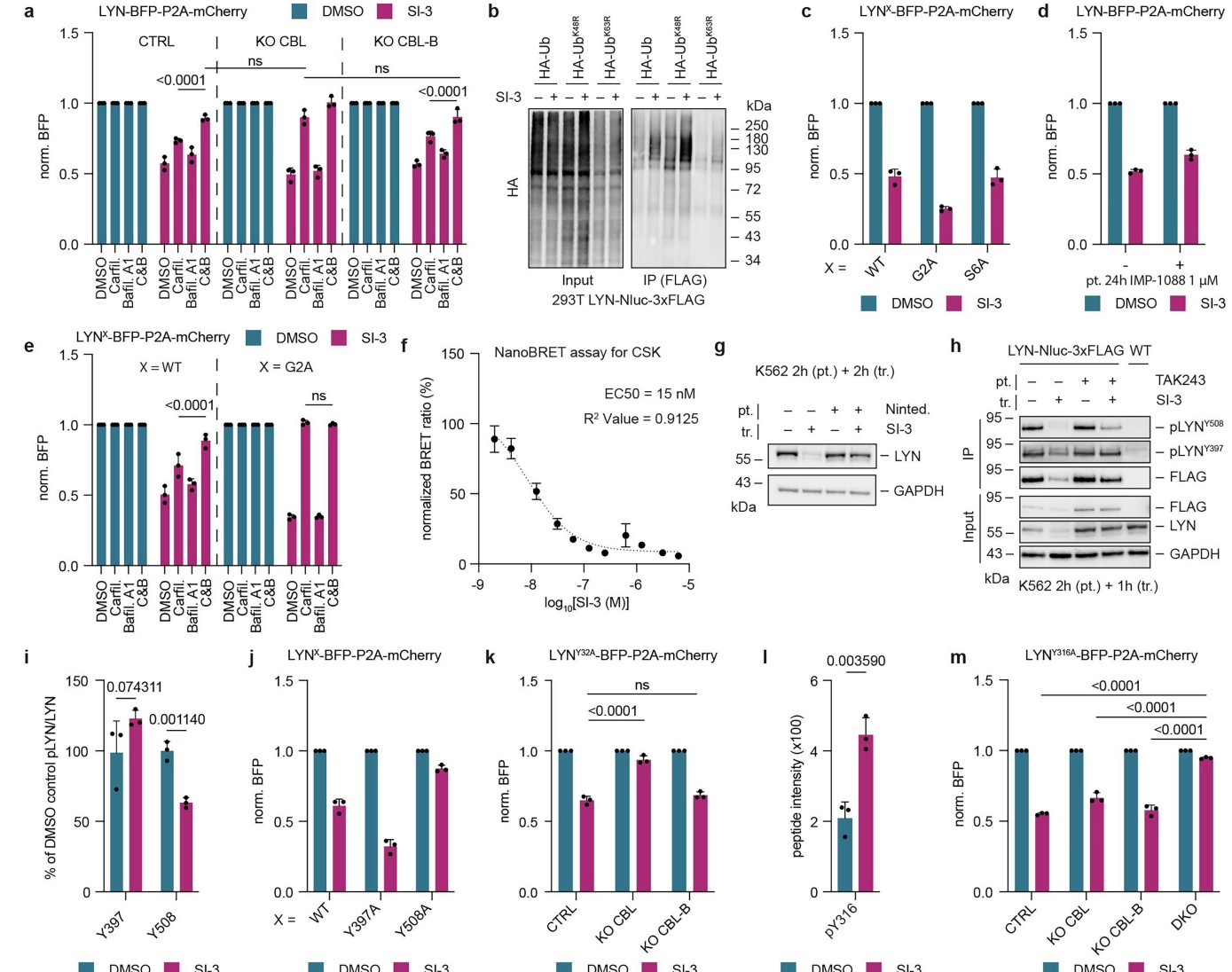

**Extended Data Fig. 5 | SI-3 selectively destabilizes LYN via a dual phosphodegron and dual E3 mechanism of action. a** Chemical rescue upon genetic k.o. for sg*AAVS1* (CTRL) or single CBL/CBL-B KO. Cells were pre-treated for 2 h with 1 µM Carfilzomib (Carfil.) or 100 nM Bafil. A1 or both (C&B) followed by 8 h of 156 nM SI-3 treatment and flow cytometric analysis (Two-way ANOVA with Tukey's test for multiple comparisons, ns > 0.9999, n = 3). **b** FLAG immunoprecipitation (IP) followed by HA immunoblot of lysates of 293 T FlpIn Lyn-Nluc-3xFLAG cells transfected with HA-ubiquitin (Ub, Ub[K48R] or Ub[K63R]) after 2 h of pre-treatment (pt.) with 1 µM Carfilzomib and 100 nM Bafil. A1, followed by 30 min of DMSO or SI-3 treatment (n = 3). **c** FACS stability reporter assay for LYN WT, G2A or S6A variants treated for 8 h with DMSO or 156 nM SI-3. **d** FACS stability reporter assay for LYN pre-treated for 24 h with IMP-1088 (1 µM) or DMSO, followed by 8 h of DMSO or 156 nM SI-3. **e** FACS stability reporter assay of LYN WT or G2A, pre-treated with rescue compounds from (a) followed by 8 h of DMSO or 156 nM SI-3 treatment (Two-way ANOVA with Tukey's test for multiple comparisons, ns > 0.9999, n = 3). **f** NanoBRET measurement of tracer displacement from CSK by SI-3 (n = 2, m = 2). **g** Immunoblot analysis of K562 cells pre-treated with Nintedanib (10 µM, 2 h)

followed by 2 h of SI-3 (156 nM) treatment (n = 3). **h** FLAG immunoprecipitation (IP) in K562 Lyn-Nluc-3xFLAG or parent (WT) control upon 2 h pre-treatment (pt.) with 1 µM TAK243 or DMSO followed by 1 h of DMSO or SI-3 treatment (n = 3). **i** Quantification of (h) using TAK243 pre-treated (degradation impaired) samples and comparing pY508 or pY397 LYN to total LYN. The quantification was performed on the IP fractions (FLAG IP) for the bands corresponding to LYN-Nluc-3xFLAG and normalized to the DMSO control lane (two-sided, unpaired t-test, n = 3). **j** FACS stability reporter assay for LYN WT, Y397A or Y508A treated for 8 h with DMSO or 156 nM SI-3 (n = 3). **k** Genetic k.o. (CTRL = sg*AVVS1*, KO CBL or KO CBL-B) for LYN[Y32A] stability reporter cell line and SI-3 degradation assessment (8 h, 156 nM) measured by flow cytometry (Two-way ANOVA with Tukey's test for multiple comparisons, ns > 0.9999, n = 3). **l** Phospho-peptide quantification of LYN pY316 measured in the BioID dataset and shown as normalized peptide intensity (two-sided, unpaired t-test, n = 3). **m** FACS stability assay for LYN Y32A upon KO of CBL, CBL-B or both (DKO) treated for 8 h with DMSO or 156 nM SI-3 (Two-way ANOVA with Tukey's test for multiple comparisons, n = 3). n = biological replicates, m = technical replicates, all data shown as mean of replicates ± SD unless specified otherwise.

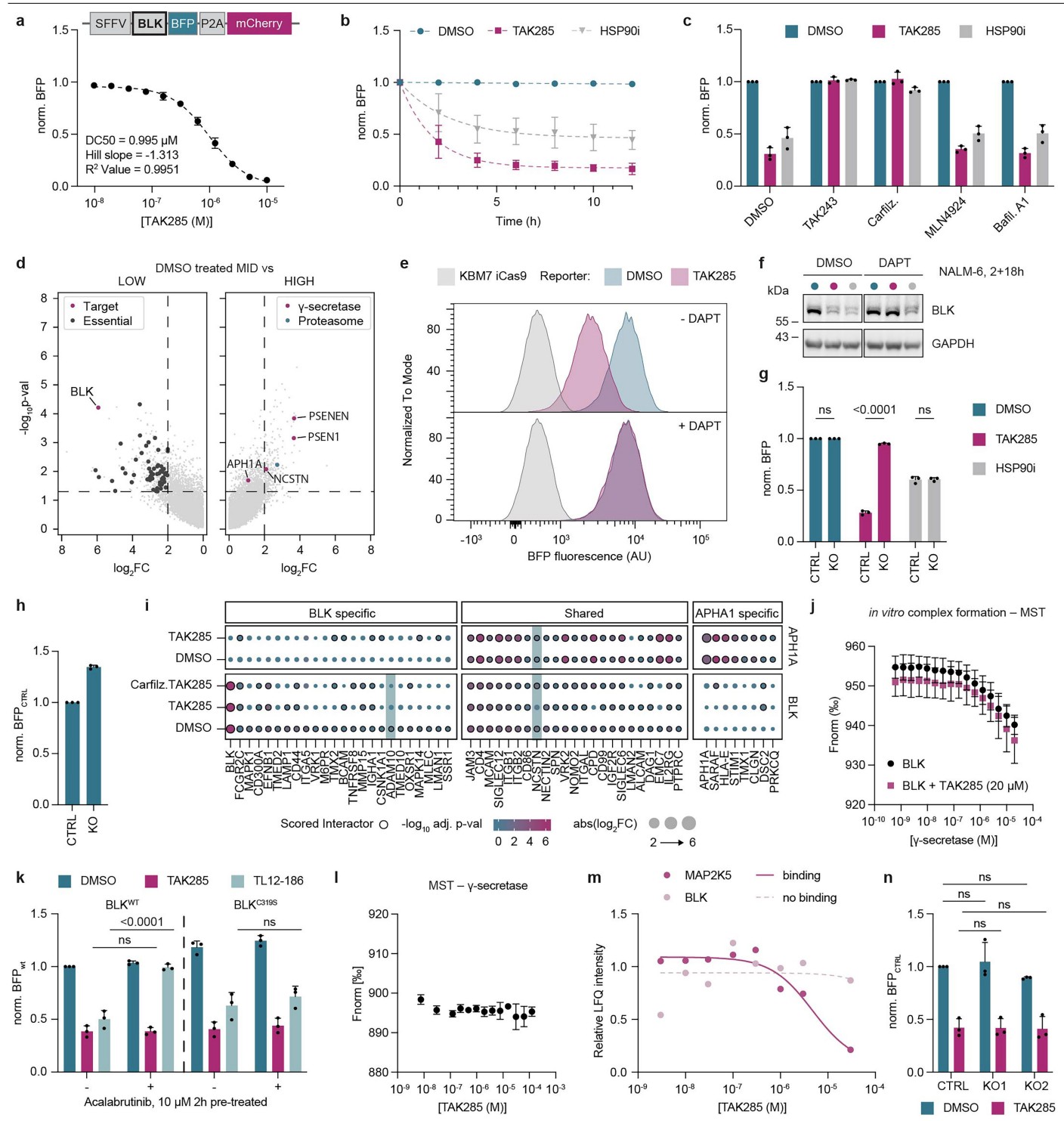

**Extended Data Fig. 6 |** See next page for caption.

**Extended Data Fig. 6 | TAK285 degrades BLK in a γ-secretase dependent manner. a** Schematic of the stability reporter design used in the cell line background KBM7 iCas9. Dose response titration of TAK285 to determine the degradation of the BLK stability reporter as measured by flow cytometry. **b** Flow cytometric analysis of the time-dependent destabilization of BLK (stability reporter). TAK285 was treated at 2.5 µM and HSP90i at 10 µM. $t^{1/2}$: TAK285 = 1.172 h, HSP90i = 1.740 h; $D_{max}$: TAK285 = 82.49%, HSP90i = 53.68%, $R^2$ value: TAK285 = 0.9994, HSP90i = 0.9902 (n = 3). **c** Chemical rescue of TAK285 and HSP90i mediated BLK stability reporter degradation (n = 3). TAK243 (1 µM), Carfilzomib (1 µM), MLN4924 (1 µM) and Bafil. A1 (100 nM) were pre-treated for 2 h, followed by 6 h of 2.5 µM TAK285 treatment prior to flow cytometry. **d** Matched genome-wide CRISPR/Cas9 data to Fig. 3b shown for vehicle control (DMSO) (one-sided MAGeCK) (n = 2). **e** Representative BLK stability reporter flow cytometry histogram plot for 2 h DMSO or 12.5 µM DAPT pre-treated cells followed by 6 h 2.5 µM TAK285 treatment. **f** Immunoblot of NALM-6 cells pre-treated for 2 h with DMSO or DAPT (12.5 µM) followed by 18 h 2.5 µM TAK285 or 10 µM HSP90i treatment (n = 3). **g** Genetic k.o. of *PSENEN* (KO; γ-secretase subunit) treated for 6 h with DMSO, TAK285 (2.5 µM) or HSP90i (10 µM) followed by flow cytometric analysis (Two-way ANOVA with Tukey's test for multiple comparisons, ns = >0.9999 (DMSO), 0.9998 (HSP90i); n = 3). **h** *PSENEN* k.o. (KO) baseline BLK stability values measured by flow cytometry and depicted normalized to CTRL (sg*AAVS1*) (n = 3). **i** Kinase and type I transmembrane protein interaction partners for BLK and APH1A mapped by BioID. Interaction partners were scored in baseline (DMSO) conditions against GFP controls and grouped as shared or bait-specific interactors. ADAM10 and the γ-secretase subunit NCSTN are highlighted in green. Interactors were ordered in descending $\log_2$FC. Dot size corresponds to $\log_2$FC of each protein against the GFP control. Black dot outlines indicate significantly scored interaction partners within each condition. The colour gradient represents the -$\log_{10}$ of the adjusted p-value (adj. p-val, BH) (n = 3). **j** MST measurements of complex formation for BLK and γ-secretase as the ligand in absence or presence of TAK285 (n = 3). **k** Chemical competition using 2 h of 10 µM Acalabrutinib pre-treatment followed by 6 h of 2.5 µM TAK285 or 1 µM TL-12-186 treatment and analysis by flow cytometry across two stability reporters: BLK$^{WT}$ or the cysteine mutant BLK$^{C319S}$ (Two-way ANOVA with Tukey's test for multiple comparisons, ns = >0.9999 (WT), 0.9004 (C319S); n = 3, depicted normalized to BLK$^{WT}$ as indicated in the axis label). **l** MST measurement for γ-secretase and TAK285 as ligand (n = 3). **m** Dose-ranging chemoproteomics (Kinobead profiling) in Jurkat and MCF7 cell lysates. MAP2K5 was identified as only binder (solid pink line, $K_D$ = 2.55 µM), while no binding was detected for BLK (dashed light pink line). **n** Flow cytometric analysis of BLK stability reporter cells with sgRNAs targeting either AAVS1 (CTRL) or MAP2K5 (KO1, KO2) (Two-way ANOVA with Tukey's test for multiple comparisons, ns from top to bottom: 0.7873, 0.9924, >0.9999, >0.9999; n = 3). All values represent the mean values ± SD; n = biological replicates, m = technical replicates.

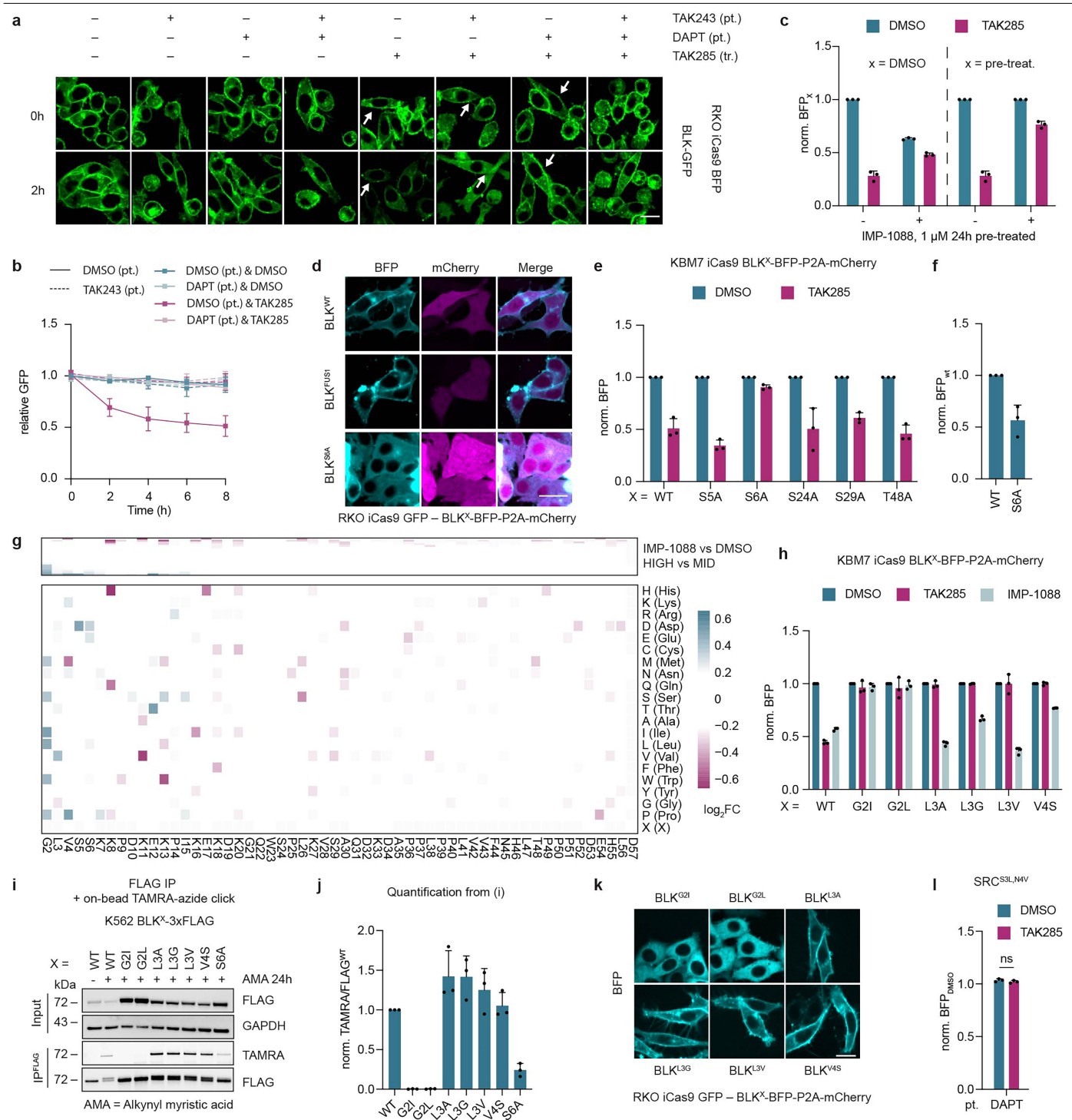

**Extended Data Fig. 7 |** See next page for caption.

**Extended Data Fig. 7 | TAK285 re-localizes BLK in a γ-secretase dependent manner. a** Representative microscopy images for a BLK-GFP clonal cell line (RKO iCas9-BFP) assessed for different treatment conditions. Cells were pre-treated (pt.) for 1 h with TAK243 (1 µM) and DAPT (12.5 µM) followed by DMSO or TAK285 (2.5 µM) treatment (tr.) for the indicated timeframe. White arrows highlight localization patterns (n = 2 (clonal cell lines), m = 2). See SI Fig S7e for additional example. **b** Quantified GFP intensity from images shown in (b) and additional timepoints. The mean total GFP abundance per cell was normalized per 0 h DMSO or TAK243 pre-treatment (n = 2 (clonal cell lines), m = 2). **c** Inhibition of myristoylation and measurement of BLK stability after TAK285 treatment. Myristoylation was inhibited 24 h prior to 6 h of 2.5 µM TAK285 treatment using the NMT1/2 inhibitor IMP-1088 (1 µM). Left: shown as DMSO-normalized data. Right: normalized per pre-treatment (x; DMSO or IMP-1088, respectively) (n = 3). **d** Representative microscopy images for localization of different BLK stability reporter pools (BFP-P2A-mCherry) in RKO iCas9 GFP cell line from Fig. 3e and (**e**). **e** Stability reporter mutant panel of all phospho-susceptible residues in the unique domain of BLK, assessed after 6 h 2.5 µM TAK285 treatment using flow cytometry (n = 3). **f** Baseline stability values for $BLK^{WT}$ or $BLK^{S6A}$ stability KBM7 iCas9 reporter cells from (**e**) measured by flow cytometry (n = 3). **g** DMS data for IMP-1088 treated (1 µM, 24 h) BLK DMS stability reporter library depicted as normalized $\log_2$FC to DMSO (n = 3). **h** Orthogonal validation of selected mutations treated with either DMSO, TAK285 (6 h, 2.5 µM) or IMP-1088 (1 µM, 24 h) and analysed by flow cytometry (n = 3). **i** On bead TAMRA azide click for individual BLK mutants after FLAG immunoprecipitation (IP, n = 3). **j** Quantification of (i). **k** Representative microscopy images for individual BLK stability reporter mutations (n = 3). **l** Flow cytometric analysis of KBM7 iCas9 stability reporter $SRC^{S3L, N4V}$ cells pre-treated with DAPT, followed by DMSO or TAK285 (2.5 µM, 6 h) treatment. Data matched to Fig. 3h (Two-way ANOVA with Tukey's test for multiple comparisons, ns = 0.8754) (n = 3). All values represent the mean values ± SD; n = biological replicates, m = technical replicates. Scalebars = 25 µm.

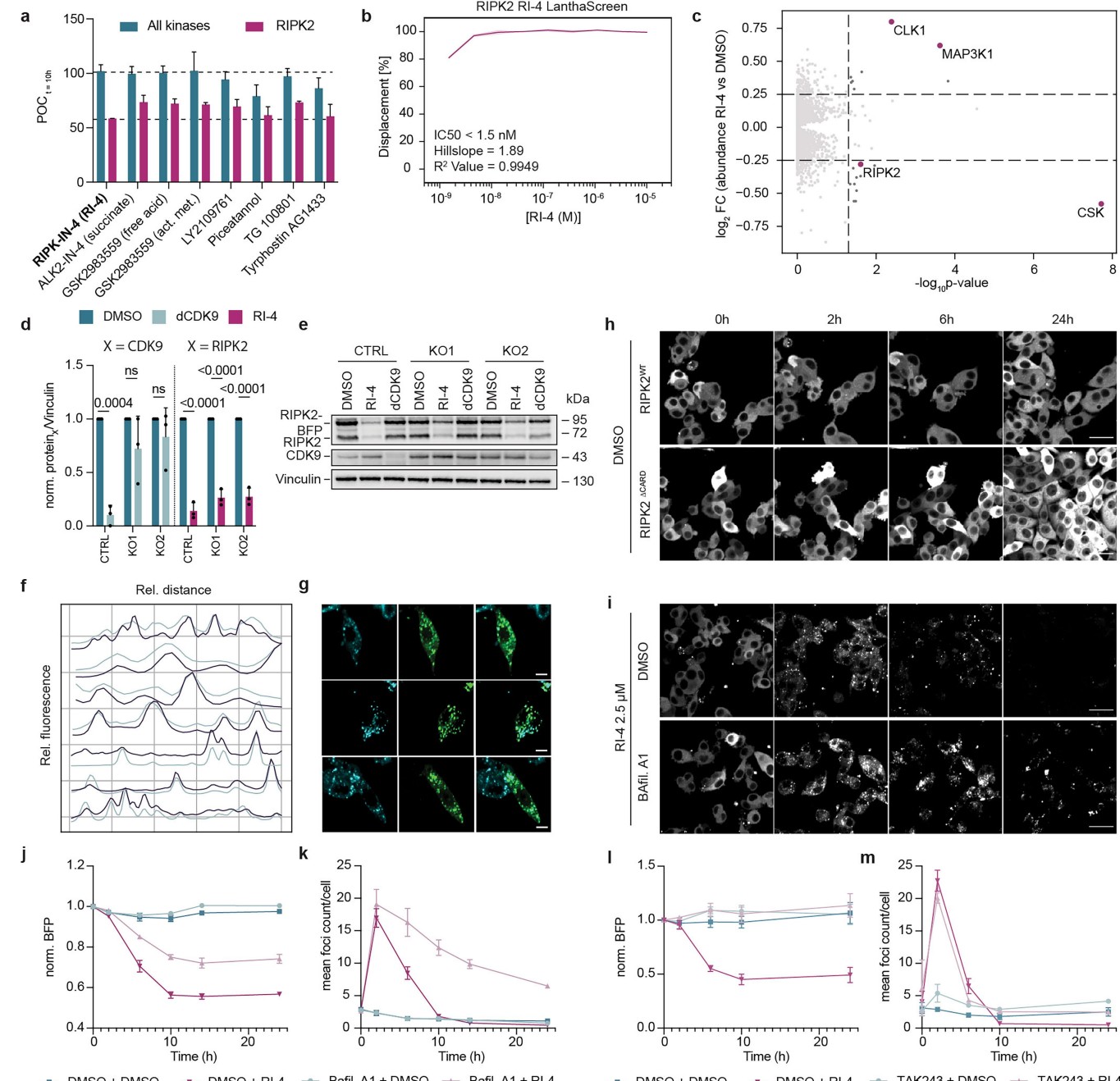

**Extended Data Fig. 8 | RI-4 degrades RIPK2 by inducing kinase multimerization, leading to its clearance via macro-autophagy. a** Drug screen data shown as mean values ± SD for all kinases vs RIPK2 only for the 10 h timepoint. Top and bottom dotted line indicated RI-4's degradation window (m = 2). **b** RI-4 LanthaScreen drug binding data. (m = 2, error bar = CI). **c** Expression proteomics of 10 µM RI-4 treated K562 for 18 h (One-way ANOVA), (n = 3). **d** Quantification of (e) and respective replicates for CDK9 or RIPK2 abundance for either dCDK9 or RI-4 treated samples, respectively. Values are shown in relation to Vinculin and normalized to DMSO (Two-way ANOVA with a Tukey's multiple comparison test, ns = 0.3953 (KO1); 0.8258 (KO2) (n = 3). **e** Example immunoblot of *PSMB5* genetic k.o. of RKO iCas9 RIPK2-BFP stability reporter harbouring cells expressing either a control sgRNA (CTRL, sg*AAVS1*) or sgRNAs targeting *PSMB5* (KO1/KO2) treated for 6 h with either DMSO, 2.5 µM RI-4 or 1 µM dCDK9 (n = 3). **f** Representative normalized fluorescent profiles of RIPK2-GFP and RIPK2-BFP expressing RKO iCas9 cells upon RI-4 treatment

(2 h, 2.5 µM) (n = 3). **g** Representative images of RIPK2 stability reporter cells co-expressing RIPK2-GFP upon RI-4 treatment (2 h, 2.5 µM) (n = 3). **h** Representative microscopy images (BFP channel) of RIPK2^FL (full length) and RIPK2^ΔCARD stability reporter in RKO iCas9 cells for vehicle control (DMSO) treatments matched treatments shown in Fig. 4h–j. **i** Representative microscopy images (BFP channel) of RIPK2 stability reporter upon RI-4 treatment in RKO iCas9 cells for either DMSO or Bafil. A1 pre-treated (2 h) cells followed by DMSO or RI-4 (2.5 µM) treatment for a total timeframe of 24 h. **j, k** Quantification of RIPK2 stability (j) or mean number of RIPK2 foci per cell (k) as shown in (i) (n = 3, m = 2). **l, m** Quantification of normalized BFP (l) or mean RIPK2 foci number (m) per cell of RIPK2 stability reporter cells (RKO iCas9) pre-treated (1 h) with DMSO or 1 µM TAK243 followed by DMSO or 2.5 µM RI-4 treatment (n = 3, m = 2). All values represent the mean values ± SD unless specified otherwise; n = biological replicates, m = technical replicates. Scale bars = 50 µm (h, i), 10 µm (g).

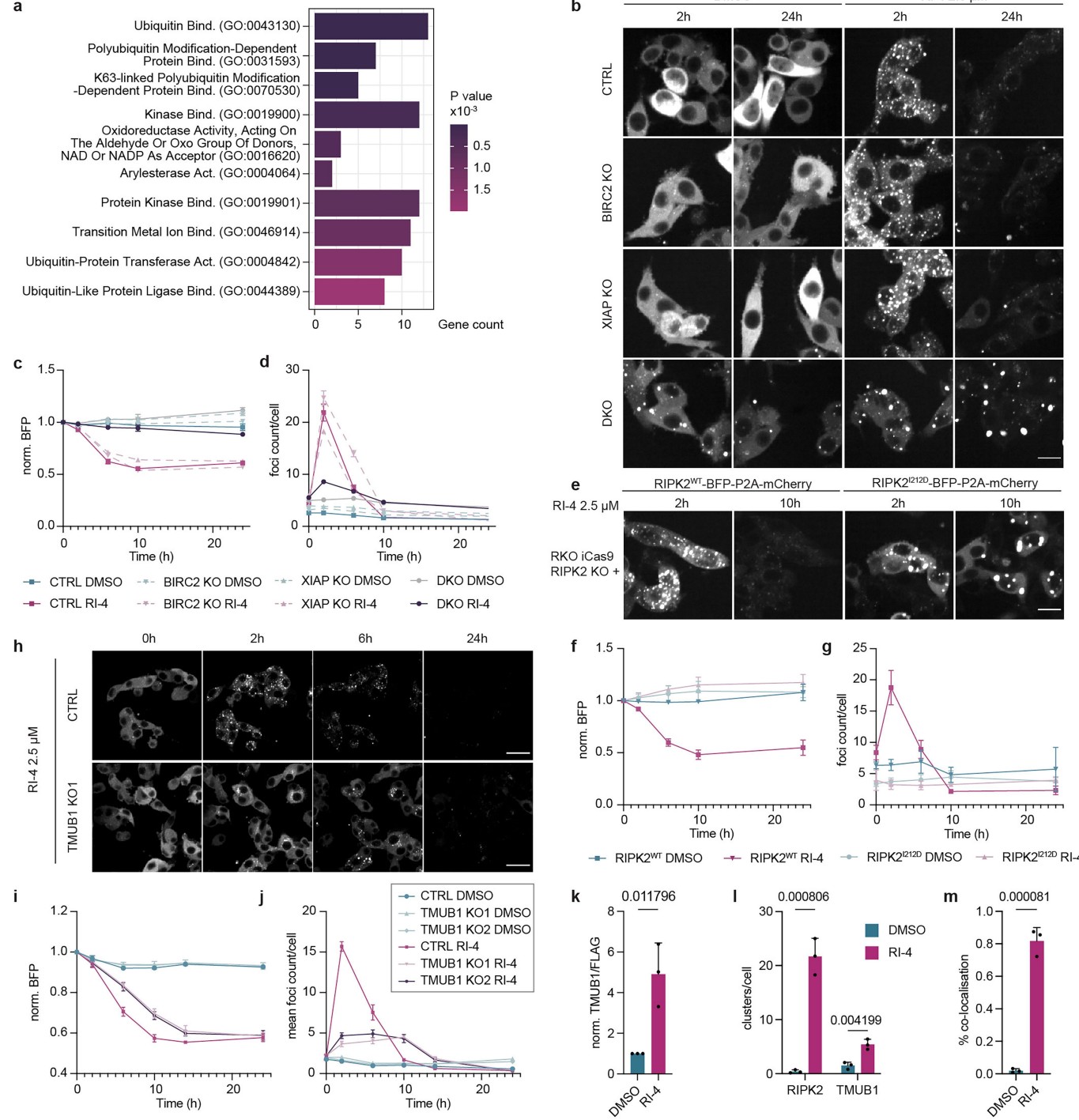

**Extended Data Fig. 9 | RI-4 induced, ubiquitin-mediated and TMUB1 facilitated RIPK2 degradation. a** Top 10 enriched GO terms (Molecular Function 2023) for scored interactors of RIPK2 miniTurbo vs GFP for 1 h of RI-4 (2.5 μM) treatment (one-sided Fisher Exact Test, significant terms p-val. <0.01). **b** Representative microscopy images (BFP channel) of RIPK2 stability reporter upon RI-4 (2.5 μM) treatment in RKO iCas9 cells expressing a control sgRNA (CTRL, sg*AAVS1*) or a sgRNA targeting *BIRC2 or XIAP* or both (DKO). Scale bar = 25 μm. **c, d** Quantification of RIPK2 stability (c) or mean number of RIPK2 foci per cell (d) as shown in b and the respective DMSO controls (n = 3). **e** Representative microscopy images (BFP channel) of RIPK2 stability reporter (WT or I212D) upon RI-4 (2.5 μM) treatment in RKO iCas9 RIPK2 KO cells. Scale bar = 25 μm. **f, g** Quantification of RIPK2 stability (f) or mean number of RIPK2 foci per cell (g)

as shown in e and the respective DMSO controls (n = 3). **h** Representative microscopy images (BFP channel) of RIPK2 stability reporter upon RI-4 (2.5 μM) treatment in RKO iCas9 cells expressing a control sgRNA (CTRL, sg*AAVS1*) or a sgRNA targeting *TMUB1* (TMUB1 KO1). Scale bar = 50 μm. **i, m** Quantification of RIPK2 stability (l) or mean number of RIPK2 foci per cell (m) as shown in (h) as well as for TMUB1 KO2 and the respective DMSO controls (n = 3, m = 2). **k** Quantification of bait normalized TMUB1 levels after co-immunoprecipitation from Fig. 4l and respective replicates. **l, m** Quantification of Fig. 4m for total number of foci (TMUB1 or RIPK2) as well as percentage foci co-localization. All values represent the mean values of the biological replicates ± SD; n = biological replicates, m = technical replicates. (k-m) Statistical significance was assessed using an unpaired two-tailed Student's t-test.

## Extended Data Table 1 | Screened kinases and hit number breakdowns

| Kinase | #Hits$^{Total}$ | #Hits$^{Selective}$ | Kinase | #Hits$^{Total}$ | #Hits$^{Selective}$ |
|---|---|---|---|---|---|
| ABL1 | 29 | 14 | INSRR | 3 | |
| AKT1 | | | IRAK3 | 1 | |
| ALK | 2 | | JAK1 | 3 | 2 |
| ARAF | | | LCK | 1 | |
| AURKA | 6 | 1 | LIMK1 | 11 | 6 |
| AURKB | 5 | 1 | LIMK2 | 6 | 1 |
| AURKC | 2 | 1 | LMTK2 | 2 | 1 |
| BLK | 9 | 7 | LYN | 2 | 1 |
| BMX | 9 | 1 | MAP2K1 | | |
| BRAF | 13 | 4 | MAP2K2 | | |
| BRAF$^{V600E}$ | 9 | 3 | MAP4K2 | | |
| BTK | | | MAPK1 | | |
| CAMK2A | | | MAPK4 | | |
| CAMK2G | 2 | | MAPKAPK2 | | |
| CDK10 | 6 | 1 | MKNK1 | 2 | |
| CDK2 | 1 | 1 | NTRK2 | 2 | 1 |
| CDK3 | 1 | | NTRK3 | 2 | 1 |
| CDK4 | | | PAK4 | | |
| CDK5 | | | PBK | | |
| CDK6 | 4 | 3 | PDGFRB | 2 | 2 |
| CDK7 | 18 | 4 | PDK1 | | |
| CDK9 | 16 | 6 | PDPK1 | 1 | |
| CHEK2 | | | PIK4CA | 1 | |
| CSF1R | 7 | 1 | PIK4CB | 2 | 1 |
| CSK | | | PLK1 | 2 | 1 |
| CSNK1A1 | 4 | | PRKAA1 | 4 | 1 |
| CSNK1D | | | PRKCA | 1 | |
| DDR1 | 19 | 7 | PTK2 | 17 | 7 |
| DYRK1B | 1 | | PTK2B | 2 | 2 |
| DYRK2 | 8 | 2 | RAF1 | 3 | 2 |
| EGFR | | | RAF1$^{S257L}$ | 21 | 14 |
| EGFR$^{G719S}$ | 4 | 1 | RET | 3 | 1 |
| EGFR$^{G719S\&T790M}$ | | | RET$^{M918T}$ | 3 | |
| EGFR$^{L718V}$ | | | RIOK1 | 1 | |
| EGFR$^{L858R}$ | 1 | | RIPK1 | 3 | 2 |
| EGFR$^{L861Q}$ | 1 | | RIPK2 | 9 | 8 |
| EGFR$^{T790M}$ | 2 | 1 | ROR2 | | |
| EPHA2 | 4 | 1 | RPS6KA1 | | |
| ERBB3 | 6 | 1 | SPHK1 | 8 | 3 |
| ERBB4$^{R711C}$ | 4 | 2 | SRC | | |
| ERN1 | 1 | 1 | STK24 | | |
| FGFR1 | 8 | | STK38 | 2 | 1 |
| FGFR2 | 6 | 3 | SYK | | |
| FGFR3 | | | TEC | | |
| FGR | | | TYK2 | 17 | 4 |
| FLT3 | 4 | 2 | ULK3 | 9 | 3 |
| FYN | 13 | 9 | YES1 | | |
| GSK3A | 1 | 1 | ZAK | | |
| HER2 | 32 | 16 | ZAP70 | | |

The kinase column indicates the kinase screened as Nluc-3xFLAG reporters in K562. Number of hits (#Hits) are indicated for the respective total and selective (only one kinase scoring per compound) counts across 1,570 compounds.

# Reporting Summary

## Statistics

For all statistical analyses, confirm that the following items are present in the figure legend, table legend, main text, or Methods section.

| n/a | Confirmed | |
|---|---|---|
| ☐ | ☒ | The exact sample size (*n*) for each experimental group/condition, given as a discrete number and unit of measurement |
| ☐ | ☒ | A statement on whether measurements were taken from distinct samples or whether the same sample was measured repeatedly |
| ☐ | ☒ | The statistical test(s) used AND whether they are one- or two-sided <br> *Only common tests should be described solely by name; describe more complex techniques in the Methods section.* |
| ☒ | ☐ | A description of all covariates tested |
| ☐ | ☒ | A description of any assumptions or corrections, such as tests of normality and adjustment for multiple comparisons |
| ☐ | ☒ | A full description of the statistical parameters including central tendency (e.g. means) or other basic estimates (e.g. regression coefficient) AND variation (e.g. standard deviation) or associated estimates of uncertainty (e.g. confidence intervals) |
| ☐ | ☒ | For null hypothesis testing, the test statistic (e.g. *F*, *t*, *r*) with confidence intervals, effect sizes, degrees of freedom and *P* value noted <br> *Give P values as exact values whenever suitable.* |
| ☒ | ☐ | For Bayesian analysis, information on the choice of priors and Markov chain Monte Carlo settings |
| ☒ | ☐ | For hierarchical and complex designs, identification of the appropriate level for tests and full reporting of outcomes |
| ☒ | ☐ | Estimates of effect sizes (e.g. Cohen's *d*, Pearson's *r*), indicating how they were calculated |

*Our web collection on statistics for biologists contains articles on many of the points above.*

## Software and code

Policy information about availability of computer code

**Data collection**

NanoGlo Lytic: Multilabel Plate Reader Platform Victor X3 model 2030 (PerkinElmer).
Drug screen: EnVision plate reader (Revvity).
Flow cytometry: BD LSRFortessa using BD FACSDiva software (v9.0), BD FACSAria Fusion using BD FACSDiva software (v8.0.2) or CytoFLEX SRT Benchtop Cell Sorter using CytExpert SRT-Software (v1.1.0.10007).
Western blotting/SDS gels: ChemiDoc Touch imaging system (BioRad) operated on Image Lab (v2.4.0.03).
NGS: HiSeq 3000 or NovaSeq 6000, Illumina (https://www.illumina.com/).
Mass spectrometry: Orbitrap Fusion Lumos Tribrid mass spectrometer coupled to a Dionex Ultimate 3000 RSLCnano system and operated via Xcalibur (4.3.73.11) and Tune (v3.4.3072.18).
Confocal microscopy: PerkinElmer Opera Phenix using Harmony (v4.9 and later versions).
MST: Monolith NT.115 (NanoTemper), MO.Affinity Analysis (v2.3)
Chemistry:  Interchim puriFlash XS 420Plus (column chromatography), Agilent 1260 Infinity II (HPLC), InfinityLab LC/MSD (LR-MS), Bruker Daltonik micrOTOF-QII (HR-MS), Bruker AV 400 HD (NMR)

**Data analysis**

Flow Cytometry Analysis: Flowjo (v10.6.2)
FACS-based CRISPR screens: pipelines for sgRNA quantification and statistical analysis are available on Github (https://github.com/ZuberLab/crispr-processnf/ tree/566f6d46bbcc2a3f49f51bbc96b9820f408ec4a3 and https://github.com/ZuberLab/crisprmageck- nf/tree/c75a90f670698bfa78bfd8be786d6e5d6d4fc455). Packages: fastx-toolkit (v0.0.14), Bowtie2 (v2.4.5), featureCounts (v2.0.1), MAGeCK (v0.5.9).
Western blot quantification: Image Lab (v6.1 build 7)
Image analysis: cellpose (0.6.5-foss-2020b), cellprofiler (4.1.3-foss-2020b), Fiji (ImageJ, 2.1.1/1.53i)
Data compiling, processing and statistical analyses: Microsoft Excel for Microsoft 365 (v16.86), R (v4.3.1), GraphPad Prism (v10.0.3), python (3.7.6), sklearn (v1.0.1), matplotlib (v3.5.3, v3.4.2), seaborn (v0.12.2), numpy (v1.21.5), pandas (v1.0.1), scipy (v1.4.1)

Mass Spectrometry: Proteome Discoverer (v2.4.1.15), MaxQuant (1.5.3.30/v2.4.9.0) , R (version 4.3.1), drc (10.1126/science.ade3925)
DMS: samtools (v1.17, v1.15.1), cutadapt (v4.4), FastqToSam (v3.0.0), Trim Galore (v0.6.6), bwa (v0.7.17), GATK (v4.1.8.1), pheatmap (v.1.0.12), R (v.4.1.0)
Structural predictions and molecular dynamics simulations: AlphaFold3 (V.3.0), CHARMM-GUI Membrane Builder, GROMACS (2023.2), GetContacts (https://getcontacts.github.io/), VMD 1.9.4

For manuscripts utilizing custom algorithms or software that are central to the research but not yet described in published literature, software must be made available to editors and reviewers. We strongly encourage code deposition in a community repository (e.g. GitHub). See the Nature Portfolio guidelines for submitting code & software for further information.

## Data

Policy information about availability of data

All manuscripts must include a data availability statement. This statement should provide the following information, where applicable:
- Accession codes, unique identifiers, or web links for publicly available datasets
- A description of any restrictions on data availability
- For clinical datasets or third party data, please ensure that the statement adheres to our policy

Data associated with the drug screening such as hit scores as well as data associated with compounds or kinases have been made available as Supplementary Data 1. Drug screening data have been deposited at https://science.aithyra.at/KinDegData. Additional CRISPR/Cas9 screening data generated in the revision process has been deposited alongside. All processed sequencing and proteomics data has been made available as Supplementary Data 2-7. Additionally, the proteomics data have been deposited to the ProteomeXchange Consortium via the PRIDE partner repository with the dataset identifiers PXD062184 for the in-vivo biotinylation experiments, PXD053130, and PXD059599 for the full proteome profiling and PXD064676 for the TAK285 chemoproteomics. Human protein fasta files were retrieved from UniProtKB (Taxonomic identified 9606, status reviewed, downloaded on the 01.12.2019 or 29.04.2024, https://www.uniprot.org/) and have been deposited alongside the respective MS data. The Kinobeads data have been deposited to the ProteomeXchange Consortium via the MASSIVE partner repository with the data set identifier MSV000095265 alongside with the utilized human protein fasta files (UniProtKB,Taxonomic identified 9606, status reviewed, downloaded on the 22.03.2016).

## Research involving human participants, their data, or biological material

Policy information about studies with human participants or human data. See also policy information about sex, gender (identity/presentation), and sexual orientation and race, ethnicity and racism.

| | |
|---|---|
| Reporting on sex and gender | N/A |
| Reporting on race, ethnicity, or other socially relevant groupings | N/A |
| Population characteristics | N/A |
| Recruitment | N/A |
| Ethics oversight | N/A |

Note that full information on the approval of the study protocol must also be provided in the manuscript.

## Field-specific reporting

Please select the one below that is the best fit for your research. If you are not sure, read the appropriate sections before making your selection.

☒ Life sciences　　☐ Behavioural & social sciences　　☐ Ecological, evolutionary & environmental sciences

For a reference copy of the document with all sections, see nature.com/documents/nr-reporting-summary-flat.pdf

## Life sciences study design

All studies must disclose on these points even when the disclosure is negative.

| | |
|---|---|
| Sample size | All presented data is based on cultured human cell lines. Sample sizes were not predetermined using statistical analyses. Sample sizes were based on prior experience in the field and our previous studies (Mayor-Ruiz et al, Mol Cell, 2019; Mayor-Ruiz et al, Nat Chem Biol, 2020). |
| Data exclusions | Individual drug screening trajectories were excluded as detailed in the methods. |
| Replication | Unless stated otherwise in figure legends or method sections, all experiments were done at least twice to ensure reproducibility. The number of independent biological experiments and technical replicates are specified in the respective figure legends. |
| Randomization | No animal or behavioral studies were conducted. Randomization was thus not necessary. All experiments were conducted in the presence of suitable positive and/or negative controls as indicated. |

| Blinding | No blinding was performed, as no subjective measurements were done. |
|---|---|

# Reporting for specific materials, systems and methods

We require information from authors about some types of materials, experimental systems and methods used in many studies. Here, indicate whether each material, system or method listed is relevant to your study. If you are not sure if a list item applies to your research, read the appropriate section before selecting a response.

## Materials & experimental systems

| n/a | Involved in the study |
|---|---|
| ☐ | ☒ Antibodies |
| ☐ | ☒ Eukaryotic cell lines |
| ☒ | ☐ Palaeontology and archaeology |
| ☒ | ☐ Animals and other organisms |
| ☒ | ☐ Clinical data |
| ☒ | ☐ Dual use research of concern |
| ☒ | ☐ Plants |

## Methods

| n/a | Involved in the study |
|---|---|
| ☒ | ☐ ChIP-seq |
| ☐ | ☒ Flow cytometry |
| ☒ | ☐ MRI-based neuroimaging |

## Antibodies

| Antibodies used | Immunoblotting: GAPDH (Santa Cruz Biotechnology, sc-365062), GAPDH (Santa Cruz Biotechnology, sc-47724), Vinculin (Szabo Scandic, SACSC-25336), FLAG (Cell Signalling Technology, #2368), LYN (Cell Signalling Technology, #2796), BLK (Cell Signalling Technology, #3262), RIPK2 (Cell Signalling Technology, 4142S), Phospho-Lyn (Tyr507) (Cell Signalling Technology, #2731), FIP200 (Cell Signalling Technology, #12436), CDK9 (Cell Signalling Technology, #2316), TMUB1 (Abcam, EPR14066), cCBL (Cell Signalling Technology, #2747), Phospho-LYN (Tyr397) (Cell Signalling Technology, #70926), HRP-conjugated Anti-Biotin (Cell Signalling, #7075), Peroxidase-conjugated Goat Anti-Rabbit IgG (Jackson ImmunoResearch 111-035-003), Peroxidase-conjugated Goat Anti-Mouse IgG (Jackson ImmunoResearch JAC115035003). |
|---|---|
| | Flow cytometry and FACS: APC anti-mouse CD90.1/Thy-1.1 antibody (1:400, no. 202526, BioLegend) and Human TruStain FcX™ Fc Receptor Blocking Solution (1:400, no. 422302, BioLegend). |
| Validation | LYN (Fig 2b, degradation assay)<br>pLYN (Tyr507 and Tyr 397) (Ext Fig 5h, functional assay)<br>BLK (Ext Fig 6f, degradation assay)<br>CDK9 (Ext Fig 8e, degradation assay)<br>FIP200 (Fig 4f, knock-out)<br>RIPK2 (Fig 4f, degradation assay)<br>TMUB1 (knock-out, data not shown)<br>cCBL (Ext Fig 4j, knock-out)<br>Anti-Biotin-HRP was validated by internal control experiments either without Biotin and/or without doxycycline induction of miniTurbo expression (data not shown).<br>Validations and multiple hundred references for Vinculin, GAPDH, FLAG and Peroxidase-conjugated AffiniPure Goat Anti-Rabbit IgG or Anti-Mouse can be found on the vendor sites and have been used in multiple, previous in-house studies (e.g. 10.1038/s41589-020-0594-x).<br>Target specificity for APC anti-mouse CD90.1/Thy-1.1 antibody (no. 202526, BioLegend) was verified by ectopic overexpression (data not shown) and as previously reported (10.1038/s41586-024-07089-6). |

## Eukaryotic cell lines

Policy information about cell lines and Sex and Gender in Research

| Cell line source(s) | KBM7 cells were obtained from T. Brummelkamp lab (Carette et al, Science, 2009), KBM7 iCas9 (originally obtained from Haplogen Bioscience), RKO iCas9-GFP and iCas9-BFP were gifted by J. Zuber (IMP - Research Institute of Molecular Pathology), NALM-6 were obtained from A. Villunger (10.1126/sciadv.ado6607). 293T and K562 were purchased from ATCC, 293T lentiviral packaging cells were obtained from Clontech and Flp-In™ T-REx™ 293 were obtained from Invitrogen. Jurkat cells (Clone E6.1), MCF-7, COLO-205 and MV-4-11 have been previously used in-house (10.1038/s41589-023-01459-3) or obtained from ATCC. |
|---|---|
| Authentication | All used cell lines were authenticated by vendors and routinely authenticated via cell morphology. Successfull CRISPR-based editing of cell lines was confirmed by cell-based degradation assays and/or immunoblotting. NALM-6 were additionally validated by STR profiling. |
| Mycoplasma contamination | All used cell lines were routinely tested and confirmed negative for mycoplasma contamination. |
| Commonly misidentified lines (See ICLAC register) | No commonly misidentified cell lines were used. |

# Plants

| | |
|---|---|
| Seed stocks | N/A |
| Novel plant genotypes | N/A |
| Authentication | N/A |

# Flow Cytometry

## Plots

Confirm that:

☒ The axis labels state the marker and fluorochrome used (e.g. CD4-FITC).

☒ The axis scales are clearly visible. Include numbers along axes only for bottom left plot of group (a 'group' is an analysis of identical markers).

☒ All plots are contour plots with outliers or pseudocolor plots.

☒ A numerical value for number of cells or percentage (with statistics) is provided.

## Methodology

**Sample preparation**

The detailed generation and sample preparation can be found in the SI Methods. In brief, stability reporter cell lines were generated by lentiviral transduction of the respective kinase-BFP-P2A-mCherry vectors. Next, cells were either sorted or used directly for experiments. For CRISPR/Cas9 screens and most canonical kinase reporter cell lines, sorted cell pools were additionally sorted as single clones and used after recovery and validation of degradation with the respective hit compounds. For experiments with sorted or unsorted pools of mutated kinase reporters, the respective canonical unsorted or sorted cell pools were used as a reference. sgRNA's were transduced by lentiviral delivery and used after selection with G418. sgRNA harboring cells were confirmed before and after selection by surface antigen staining. For functional degradation experiments, if required gene knock-outs were induced at the timepoints indicated in SI Table S3, and prior to flow cytometry cells were treated with the compounds indicated in the corresponding figure legends. In most instances, cells were directly measured on a BD LSRFortessa. For sorting, cells were stained and fixed as described in the methods and sorted to achieve at least 500 x or 1000 x sgRNA representation per replicate for the genome-wide or UPS focussed sgRNA libraries, respectively. The DMS screen was sorted at >1000 x representation per replicate.

Flow cytometric data analysis was performed in FlowJo v10.6.2. BFP and mCherry mean fluorescence intensity (MFI) values were normalized by background subtraction of the respective values from reporter-negative KBM7 iCas9 cells. Kinase stability was calculated as the ratio of background subtracted BFP to mCherry MFI, and is displayed normalized to the respective control condition as indicated in the figure legends. For the analysis only reporter positive cells were considered.

**Instrument**

Data acquisition was performed on a BD LSRFortessa (4 laser, 16 detector configuration; BD Bioscience). Cell sorting was performed on a BD FACSAria Fusion (5 lasers, 16 detectors; BD Bioscience) for CRISPR/Cas9 screens or a CytoFLEX SRT (4 lasers, 15 detectors; Beckman Coulter) for cell line generation.

**Software**

BD FACSDiva software (v8.0.2 and v9.0), Beckman Coulter CytExpert SRT (v 1.1.0.10007), Flowjo (v10.6.2)

**Cell population abundance**

For the FACS-based CRISPR/Cas9 screens, cells were sorted into high (5% of cells) or low (5%), and mid (30%) populations. Fractions were re-analyzed after collection for purity. In the case of > 5% of cross- contamination, samples were discarded before further processing. The identical process was applied for the FACS-based DMS screen.

**Gating strategy**

The forward scatter area vs. side scatter area plot was used to separate cell events from debris and dead cells. Forward scatter height vs. forward scatter area and/or side scatter width vs. side scatter height plots were used to separate single cells from aggregates. For cell populations that had not been sorted prior to the experiments (unsorted pools) or sorted pools with residual reporter negative cells, reporter positive cells were further gated in the Pacific Blue-A (BFP) vs PE-TexasRed-A (mCherry) scatter plots. For the sorting of fixed cells in the CRISPR/Cas9 stability screens, dead cells were excluded based on Zombia-NIR staining (BV786-A) vs FSC-A. Next, triple positive sgRNA (AF 647-A), iCas9-GFP (FITC-A/AF 488-A) and reporter (PE-TexasRed-A) cells were sorted into the respective low, high, and mid populations based on the BFP (BV421-A/Pacific Blue-A) vs mCherry (PE-TexasRed-A) scatter plots. These gates were dynamically adjusted to keep the percentage at 5% for high and low and 30% for MID populations.

A figure exemplifying the gating strategy for all flow cytometry experiments and FACS-based screens in provided in SI Fig S2.

☒ Tick this box to confirm that a figure exemplifying the gating strategy is provided in the Supplementary Information.

