## [Peer Review File · Nature]

Inhibitors supercharge kinase turnover via native proteolytic circuits

Corresponding Author: Dr Georg Winter

Version 0:

Reviewer comments:

Referee #1

(Remarks to the Author)

In the manuscript, Scholes et al. surveyed 1570 kinase inhibitors against 98 kinases, including 10 clinically relevant mutants, to systematically identify inhibitor-induced kinase destabilization. To date, several studies have shown inhibitor-induced degradation of kinases such as HER1 by neratinib or LMTK3 by C28. Chaperone deprivation is one of the suggested mechanisms for inhibitor-induced kinase degradation, but there is emerging evidence that the mechanisms can go beyond chaperone deprivation. However, there have not been systematic studies about the frequency, generalizability, and fundamental mechanistic principles of inhibitor-mediated kinase degradation, despite the key importance of kinase inhibitors in therapeutics.

Using a scalable luminescent reporter system expressing kinases fused to Nluc in K562 cells, the authors identified 232 compounds that downregulate protein levels of at least one kinase and 66 kinases affected by at least one compound. Most destabilized kinases were HSP90 clients (although see below). However, the authors suggest that many of the observed degradation events could not be explained by chaperone deprivation. The authors selected three kinases with a different HSP90 client status (LYN, BLK and RIPK2) for detailed follow up studies. The results point to three different mechanisms, where inhibitors potentiate endogenous degradation circuits by altering kinase activity (LYN), changing cellular localization (BLK) or inducing higher-order assemblies (RIPK2).

General comments:

Overall, the aim of the study is highly relevant and exciting, and the screening strategy is sound and well-controlled. The authors have done an impressive amount of work to identify inhibitors that degrade kinases and to characterize some of the mechanisms of action. However, the follow-up studies are either somewhat incomplete (RIPK2, BTK) or reveal mechanisms that have already previously been demonstrated (LYN). This unfortunately tempers my enthusiasm for this version of the study even if many experiments are technically very well done.

For example, unlike the RIPK2 case that likely involves on-target degradation, LYN and BLK degradation is not caused by direct inhibition of these kinases. Because LYN and BLK degradation are induced by inhibitors that might touch one of the upstream kinases or other targets, it is challenging to explain how they can achieve selectivity and how this could be used for e.g. rational design of "off-target degraders".

My suggestion for improving the manuscript (if the authors choose to revise it for Nature) would be to focus much less on the LYN degradation story (or even remove it) but follow up in more detail on the RIPK2 and BTK parts. For RIPK2, it would be important to show at the mechanistic level how this works. Does the inhibitor promote a direct interaction between RIPK2 higher-order structures and autophagy regulators (akin to BCL6 degraders) or is there another mechanism at play? For BTK, how exactly does the inhibitor affect myristoylation and what is the mechanistic connection between the compound and the gamma-secretase complex? Is this due to off-target, non-kinase inhibitory activity of the compound?

I really like the idea of finding mechanisms for inhibitor-induced degradation, and I commend the authors on pursuing

several leads. Their results that at least 2 of the 3 characterized compounds lead to kinase degradation indirectly rather than directly suggest that there is likely much more to be discovered beyond the kinome. The authors “only” looked at the degradation of 98 kinases with the compounds, but if one was to extrapolate this to the entire proteome, it is likely that a significant proportion of compounds would lead to degradation of non-kinase off-targets (network targets). This is an intriguing thought.

Other major points:

1. Kinase Degradation (KinDeg) map

1.1. The authors mention that many of the observed degradation events cannot be explained by chaperone deprivation. However, in Fig 1C, the authors describe that “both strong and weak clients of Hsp90 had a markedly higher prevalence of being destabilized than non or not-defined clients, pointing to an outsized contribution of chaperone deprivation to the observed degradation events”. To clarify this, the authors should discuss what fraction of hits could be due to the chaperone deprivation model. Moreover, the graph and the statement lack statistical support.

1.2. Kinases that do not interact with Hsp90 are generally more stable. Perhaps they are inherently less susceptible to degradation, explaining the trend that the authors observe? This could be an alternative explanation to the active chaperone deprivation model.

1.3. Maybe I missed this, but what fraction of degradation hits is due to network modulation vs direct targeting?

1.4. Which kinase inhibitors are included in 1570 inhibitors? How did authors select the inhibitors and the kinases? It would be interesting to see if the screening results have any trends based on the class or type of kinase inhibitors, for example, allosteric vs ATP-competitive, selective vs promiscuous, type I vs type II.

1.5. The authors state that the reporter half-life did not correlate with the frequency of scoring, but do not show the p-value of the correlation. Also, the statement seems quite strong, and one could also reasonably say that -0.381 is a decent correlation for such a dataset.

1.6. Did the authors find any kinase stabilizers?

2. LYN kinase

2.1. The authors suggest that SI-3, a CSK kinase inhibitor, induced rapid LYN destabilization by inhibiting CSK kinase, a negative regulator of SRC family kinases including LYN. However, it was already known that genetic depletion of CSK decreased the protein level of LYN kinase in a Cbl-dependent manner (PMID 32376886). Because SI-3 is a CSK inhibitor, it is predictable that SI-3 affects the abundance of LYN kinase. Unfortunately, this significantly reduces the novelty of the authors’ finding.

2.2. CSK is a negative regulator of SFKs (SRC family kinases), including LYN, FYN, and SRC. The authors showed that CSK inhibition by SI-3 leads to activation of LYN, followed by degradation via CBL. What about the other CSK inhibitors in the dataset? Could they induce LYN destabilization? If not, what makes SI-3 selectively affect LYN kinase over other SFKs and inhibitors?

2.3. In addition to LYN, SI-3 treatment affects several other proteins, as shown by proteome profiling (Extended Fig. 3A). CSKN1D and SIK3 are other potentially degraded kinases, as are non-kinases TPRN and AMD1, among others. Clearly, the compound affects many other targets. The authors should comment on this. Are these also due to activating their endogenous degradation circuits or due to off-target effects of the inhibitor?

2.4. If SI-3 supercharges LYN degradation circuits by altering kinase activity, as the author suggest, they should compare SI-3 to other CSK inhibitors to generalize this phenomenon.

2.5. The data for the neddylation inhibitor (MLN4934) is missing (Page 6).

2.6. In Fig 2G, did authors confirm depletion of CBL and CBL-B in either protein level or mRNA level?

2.7. In Fig 2J, the authors showed that double mutant of LYN (LYN Y32A+Y316A) was almost completely inert to SI-3-induced degradation, and the degradation of Y32A mutant is CBL dependent (Extended Data Fig 3N). What about the Y316A? Can Y316A recover SI-3-induced degradation of LYN in CBL or CBL-B KO conditions? Or are there another E3s that recognize the Y316 phosphodegron?

3. BLK kinase

3.1. In the Extended Data Table 1, 9 kinase inhibitors induce BLK degradation, and 7 inhibitors selectively affect BLK. But only 4 inhibitors (Afuresertib (AKT), BP-1-102 (STAT), 4-Hydroxyionchocarpin (p38 MAPK, JNK and ERK) and TAK-285 (HER1/2) are highlighted as a BLK degraders in the supplementary file. All these inhibitors selectively induced BLK degradation even though their targets are different. Did authors test if these inhibitors have shared mechanism to induced destabilization of BLK (at least in respect to myristoylation or gamma-secretase dependency)?

3.2. The authors show that TAK285-induced BLK degradation is dependent on UPS by using TAK243 and carfilzomib. When the myristoylation of BLK is reduced upon TAK285 treatment, is ubiquitination of BLK increased? Among the enriched genes in the CRISPR/Cas9 screen, were there any E3s that might recognize un-myristoylated BLK? Work by Elledge lab has suggested that ZYG11B and ZER1 are involved in degradation of such substrates (PMID: 31273098). Maybe the authors could look into these E3s?

3.3. Unlike BLK, expression levels of non-myristoylated (G2A) mutant forms of SRC were 4- to 6-fold higher than those of their myristoylated counterparts (PMID 20584982). To further support that non-myristoylated BLK is unstable (unlike SRC), the authors could compare the stability of BLK G2A to WT.

3.4. It would be good to show the specificity of TAK285 by whole proteome profiling as was done for the other two inhibitors. Are other myristoylated proteins also affected by the compound?

4. RIPK2

4.1 RIPK-IN-4 induced RIPK2 degradation is relatively clear compared to the other two follow-up stories. I wondered if other RIPK2 inhibitors like ponatinib (type II kinase inhibitor) or GSK583 (type I kinase inhibitor) also can induce RIPosome followed by turnover via autophagy. If they do not, what makes RIPK-IN-4 different from the other inhibitors, especially GSK583 that is structurally similar to RIPK-4-IN. Can authors predict the structure of RI-4 mediated oligomerized RIPK2?

4.2. The authors state that the CRISPR/Cas9 screen revealed an enrichment of hits involved in lysosomal degradation. The statement is missing statistics (is the enrichment significant?) and how the hit annotation was done (using GO categories, Uniprot annotations, or others?).

4.3. The RIPK2 Δ CARD variant also forms foci in Fig 4H.

4.4. The authors state that RIPK2 multimerizes or forms higher-order assemblies, but this is not shown. Figure 4 title states that the process is TMUB1 dependent, but TMUB1 deletion seems to just slow down the process.

Minor points:

1. Statistics are missing in most figures.
2. Figure legend of Fig. 2I is missing
3. On page 11 "BLK S6A strongly abrogated inhibitor-induced degradation (Fig 3J)", it is not 3J, it is 3I.

Referee #2

(Remarks to the Author)

The development of protein kinase targeted PROTACs is a "hot" field right now, and relevant to this it is known that some small molecule kinase inhibitors (KIs) not only inhibit kinase activity but also trigger degradation of their target protein kinase. Here, the authors set out to survey which KIs cause degradation of their target protein kinases, as assayed using Nanoluc PK-fusions, examining the protein levels of 98 protein kinases (88 WT and 10 mutant forms) in cells treated with 1570 monovalent kinase inhibitors. They found that 69 of these PKs exhibited a reduced protein level in response to at least one KI compound, with 232/1570 KIs causing degradation of at least one PK. There was a partial correlation between degradation and whether the PK was a strong or weak HSP90 chaperone client, but the degradation of several PKs could not be explained by chaperone deprivation, nor was there necessarily a good correlation with whether the PK had been shown to be sensitive to a designed heterobifunctional degrader.

This led the authors to follow up three examples of an unexplained KI-induced protein kinase degradation response in order to determine the mechanisms underlying KI-induced degradation: LYN by SRC inhibitor-3 (SI-3), BLK by TAK285, and RIPK2 by RIPK-IN-4 (RI-4). In the first case, SI-3 caused almost complete degradation of the LYN SFK within 2 hours, an effect that was ubiquitin dependent, being nearly fully inhibited by a combination of a proteasome inhibitor plus a lysosome inhibitor, and also demonstrated that the SI-3 effects were relatively selective for LYN compared to the other SFKs. By conducting a CRISPR screen in KBM7 cells stably expressing a LYN-Nanoluc reporter for genes required LYN degradation, they found a requirement for the CBL and CBLB RING domain E3 ligases, which have previously been implicated in SFK kinase degradation. Also, using BioID proximity labeling, they found that SI-3 induced the association of LYN with both CBL and CBLB, which were the strongest hits. Consistent with a role for CBL/CBLB E3 ligase activity, they found that combined knockout of CBL and CBLB but not either single knockout, prevented SI-3 induced LYN degradation. By assessing dose dependent changes in kinase activity using Kinobead profiling of cells treated with SI-3, they deduced that CSK rather than LYN was the primary SI-3 inhibitor target, and by making and using SI-3-resistant T266M CSK and T319I LYN gatekeeper mutants they showed that only expression of the SI-3-resistant CSK mutant prevented SI-3-induced LYN degradation, implying that loss of CSK-mediated inhibitory LYN Y508 phosphorylation was responsible for the response. Their model is that LYN kinase activation in the absence of CSK activity leads to its rapid CBL/CBLB-dependent ubiquitylation and degradation. They went on to identify pY32 in LYN as a possible phosphodegron for the CBL E3 ligase and pY316 as a second candidate CBL/CBLB phosphodegron.

In the second case, treatment with 1 μ M TAK285 caused highly selective degradation of BLK, another SFK, with a half-life of ~2 hours, in a ubiquitin and proteasome-dependent manner, and with kinetics similar to those observed upon treatment with an HSP90kinase chaperone inhibitor. A CRISPR screen in KBM7 cells identified subunits of the γ -secretase complex,

including NCSTN, as being important for TAK286-induced BLK degradation, and, consistently, a proximity-labeling interaction between BLK and NCSTN was detected. However, because γ -secretase exclusively carries out intramembrane cleavage of Type 1 TMD proteins, it seemed unlikely to cleave BLK directly. Inhibition of BLK N-terminal myristoylation with IMP-1088 reduced TAK285-induced BLK degradation, implying a requirement for BLK membrane association. They went on to show that swapping just the BLK N-terminal domain with that of SRC rescued TAK285-mediated BLK degradation, implicating this region in degradation. TAK285-induced BLK degradation was not blocked by pretreatment with the covalent BLK inhibitor acalabrutinib, suggesting that TAK285 acts independently of direct binding to the BLK ATP pocket, consistent with an inability to detect BLK binding to TAK285 using a bead-based assay. Focusing on the BLK N-terminal region they found that an S6A mutation, which decreases myristoylation, causing strongly reduced BLK membrane association, as did treatment with the IMP-1088 myristoylation inhibitor. They also showed that TAK285 treatment led to a loss of BLK myristoylation and BLK relocation, which was prevented by a γ -secretase inhibitor. Consistent with release into the cytoplasm being causative in NLK degradation they found that both the S6A mutant BLK and BLK in IMP-1088 treated cells were reduced. They propose that TAK285, through an unknown adaptor protein, induces membrane-associated BLK to interact with γ -secretase, which then somehow releases BLK into the cytoplasm where it is degraded spontaneously in a ubiquitin-dependent manner.

Lastly, they analyzed how the RIPK-IN-4 RIPK inhibitor induces degradation of RIPK2, a protein kinase involved in NOD1/2 pattern recognition receptor signaling to inflammatory pathways. They showed that RI-4 causes slow RIPK2 degradation over 18 hours, with CSK being the only other PK stabilized by RI-4 treatment. An analogous CRISPR KBM7 cell screen in this system identified components of the lysosome as being important in RI-4-induced RIPK2 degradation, with TMUB1, a UBL domain protein implicated in quality control of TMD proteins at ER contact sites, being a top hit. Consistent with lysosomal degradation being important, bafilomycin A1 reduced RI-4-induced RIPK2 degradation, as did knockout of the FIP200 macroautophagy adaptor protein. They went on to show that RIPK2 aggregates appeared in response to RI-4 treatment, which were cleared in a manner blocked by the bafilomycin A1 autophagy inhibitor, and reduced by genetic depletion of TMUB1. They conclude that RI-4-triggered aggregation of RIPK2 results in slow autophagy dependent degradation of RIPK2, which is a novel protein kinase depletion mechanism.

The authors' kinome wide KinDeg screen to identify small molecule protein kinase inhibitors that also trigger degradation of their target protein kinases has led to some intriguing and unexpected findings, which have potential implications for use of small molecule protein kinase inhibitors in the clinic. To illustrate different mechanisms through which KIs can induce protein kinase degradation, they analyzed three KI/protein kinase pairs in depth - LYN/SI-3, BLK/TAK285 and PIPK3/RI-4. No common principles emerged from the three cases studied, which in one sense is presumably the point the authors are trying to make. However, as a result the paper is somewhat disjointed. In all three examples, there are unanswered questions. For instance, the mechanism through which TAK285 induces loss of BLK myristoylation and its degradation in a γ -secretase manner remains unclear, and the mechanisms underlying RI-4 induced aggregation and clearance of RIPK2 by autophagy-dependent lysosomal degradation are not well worked out.

Points: A. LYN/SI-3

1. CBL/CBLB mediated ubiquitylation of LYN would be expected to primarily cause proteasome-dependent LYN degradation, raising the question why LYN degradation is partially lysosome dependent? Does CBL-mediated degradation of LYN require it to be membrane associated? CBL-induced degradation of RTKs involves their lysosomal degradation. The LYN protein is membrane associated through two different N-terminal lipid modifications - could this explain the partial dependence of the SI-3 effect on lysosomal degradation? What effect does IMP-1088 or S6A LYN have on SI-3 induced LYN degradation?
2. Have the authors demonstrated that LYN becomes (poly)ubiquitylated in SI-3 treated cells, and, if so, determine Ub sites and Ub linkage types on LYN?
3. Figure S3M: Why didn't the authors assay LYN pY397 levels in SI-3 treated cells? If their model that SI-3 inhibition of CSK results in LYN activation is right, then pY397 levels should increase (unless SI-3 prevents LYN autophosphorylation at Y397). A LYN Y397F mutant could be used as a control that should show reduced degradation according to their model.
4. Based on a prior report, the authors propose that pY32 serves as a phosphodegron for CBL in response to SI-3, but did not demonstrate directly that SI-3 induced CBL-mediated LYN degradation requires its "SH2" domain and its ability to bind to pTyr - this could be tested with appropriate CBL mutants. With regard to the proposal that pY32 acts as a CBL-targeting phosphodegron in LYN, it would be reassuring if they showed that CBL/CBLB can bind a LYN pY32 peptide (as well as pY316 peptide) - this was not done in the cited study (ref. 33). The sequence dependence of the CBL "SH2" domain pTyr binding has been examined in several studies, with (N/D)XpY(S/T)XXP and RA(V/I)XNQP(S/T) being reported as consensus. Also, Park's group showed that DpYR is needed for CBL-dependent pY1003 MET RTK degradation. Neither the pY32 nor the pY316 site conforms well to these consensus sequence and this point deserves discussion. Moreover, in the crystal structure Y316 is in a beta strand on the surface of the LYN catalytic domain and would not be very accessible to phosphorylation or CBL "SH2" domain binding (but perhaps Y316 phosphorylation would affect the local structure).
5. What tyrosine kinases phosphorylate Y32 and Y316 in LYN when it is activated by SI-3? Scholes et al. (ref 33) proposed that Y32 is phosphorylated by an activated SFK when CSK is inhibited, but did not show this experimentally. PhosphoSitePlus indicates that Y32 is an EGFR target, but the new Kinase Library tool in PhosphoSitePlus predicts BRK, an SFK relative, as far and away the top hit for Y32 (BRK is also negatively regulated by C-terminal Tyr phosphorylation although the tyrosine kinase responsible seems to be unknown). pY316 has been frequently detected in HTP

phosphoproteomic screens, but an upstream kinase has not been reported. Intriguingly, the KL tool scores CSK as the top hit for Y316, which means the level of LYN pY316 should decrease if SI-3 inhibits CSK (did they show through an in vitro assay that SI-3 inhibits CSK kinase activity?).

6. Figure 2I: A description of panel I is missing from the Figure 3 legend.

B. BLK/TAK285:

1. They propose that TAK285, through an unknown adaptor protein, induces membrane-associated blk to interact with γ -secretase, which then somehow releases BLK into the cytoplasm where it is degraded spontaneously. The main unresolved issue is how TAK285 induces BLK association with γ -secretase - is the myristoyl group directly involved in some way other than as a membrane anchor? To test whether plasma membrane association of BLK per se is needed, BLK could be inducibly anchored to the plasma membrane or possibly a C-terminal CAXX box could be added. In addition, if released soluble BLK is constitutively degraded by the UPS, why is membrane-associated BLK protected?

2. As shown using the TAMRA assay, the loss of the myristoyl (Myr) group from BLK is triggered by TAK285 treatment, but it is unclear how the Myr group is lost, or whether it is just the Myr group that is lost. Also, is it possible that γ -secretase hydrolyzes the Myr-Gly bond (the Myr-Gly linkage is an amide linkage like the peptide bond) or cleaves another peptide bond near the N-terminus, which might in turn expose a cryptic N-end rule degron?

3. Since other SFKs are myristoylated but are unaffected by TAK285, this suggests that the specificity of BLK degradation is either inherent in the unique N-terminal sequence of BLK or in the interaction of the hypothesized adaptor protein with BLK. The authors replaced the BLK N-terminal sequence with that of SRC and showed that this prevented BLK degradation, but did not do the opposite experiment of swapping the BLK N-terminal sequence into SRC to determine whether this is sufficient to induce SRC degradation (or degradation of another irrelevant protein).

4. Figure 3J: As a minor point, the total level of BLK protein was increased in lane 4 with a proteasome inhibitor + TAK285 combination but not seen in lane 2 with the same combination – this deserves some comment,

C. RPK2/RI-4:

1. It is unclear how RI-4 induces RIPK2 aggregate formation, or how these aggregates are recognized by the autophagy machinery. Do the authors have any mechanistic insights? Are other proteins involved in aggregate formation? In this regard, the authors have implicated TMUB1 in the process, but it is unclear whether it interacts with RIPK2, perhaps in a ubiquitin-dependent manner.

2. Does autophagy-dependent degradation of RIPK2 require its ubiquitylation? Was the effect of the TAK243 Ub E1 inhibitor on RIPK2 degradation tested in this system?

3. Did the authors make an RI-4-resistant RIPK2 mutant to show that the effects of RI-4 are due to inhibition of RIPK2 kinase inhibition. Likewise, is a kinase-dead RIPK2 mutant degraded upon RI-4 treatment? This information could be relevant to how RIPK2 phosphorylation is required for aggregate formation, which might be relevant to recognition by a phosphodependent autophagy receptor protein, such as a phosphodependent LIR domain receptor.

4. If autophagy is required for RIPK2 degradation, as the authors conclude., one would have expected autophagy genes, such as FIP200, to be identified in the CRIPSR screen, but this does not appear to have been the case. Some discussion is called for.

5. If CSK degradation is induced by RI-4, as the authors state, then based on the results they obtained in the LYN/SI-3 system, LYN should also have been destabilized – was that the case?

Other points: 1. Line 105: The authors discuss whether the 68 kinases might be correlated with them being clients for the HSP90 chaperone, but do not mention whether they are clients for the kinase-dedicated chaperone CDC37.

2. Line 182: Do the ABL1 degraders degrade ABL2 and also BCR-ABL? Does the asciminib allosteric ABL inhibitor also induce ABL degradation?

3. TAK243 was used as a Ub E1 inhibitor to block protein ubiquitylation, but the target for TAK243 does not seem to have been specified in the text.

Minor point: 1. Line 91: The actual number of small molecule kinase inhibitors in clinical use worldwide is actually significantly higher than the number of 80 given in Roskowski's list of FDA-approved KIs, since several additional KIs have been approved by other regulatory agencies around the world (see - <https://www.guidetopharmacology.org/GRAC/LigandListForward?type=Approved&database=all>.)

(Remarks to the Author)

The manuscript from Winter and colleagues describes a systematic screen for small molecules that cause kinase destabilization. Using K562 cell lines engineered to express 1 of nearly 100 kinase-luciferase fusions, approximately 1500 compounds were screened at multiple time points in each cell line for decreased luminescence, amassing an impressive dataset of compound-induced destabilization events. The power of this dataset is in the systematic approach the authors took and their well-thought controls, which made it possible to identify unique ligand-destabilization relationships with a high probability of reflecting true-positive pharmacological events of biological interest. They uncover a large number of HSP90 chaperone-dependent events, recapitulating this well-known pharmacology at a possibly larger scale. They also identify 3 events for closer study, SI-3-induced LYN destabilization, TAK285 and BLK, and RIP-IN-4 and RIPK2. SI-3 was found to inhibit CSK, which activates LYN, which activates a natural mechanism of LYN degradation that is dependent on CBL. BLK degradation was found to be caused by the intrinsic instability of BLK in the cytoplasm, wherein TAK285 caused decreased myristoylation and decreased membrane localization of BLK through an unknown mechanism involving gamma secretase. RIPK2 degradation was found to be caused by autophagy of compound-induced assemblies resembling RIPosomes that are caused by RIPK2 activation, which are also degraded by autophagy (with contributions from TMUB1 to foci formation). This Reviewer has minimal comments about the data, which are impressive in both scale and rigor. Editorially, the decision on this manuscript will depend on the level of novelty assigned to the three follow-up studies in comparison to the large body of work already documenting various compound-induced destabilization events for kinases (ref. 6 in the manuscript).

In this Reviewer's reading of the authors' text, the CSK-dependent degradation of LYN, which is the most developed finding of the manuscript seems to have been a known mechanism, whereas the latter two findings seem more novel but less developed.

The BLK findings are intriguing and could potentially be revealing of an unrecognized function of gamma secretase, although the mechanism by which gamma secretase contributes is not settled by these experiments. How is gamma secretase impacting BLK membrane localization? What is the direct target of TAK285 leading to degradation? The authors suggest this for future work.

The RIPK2 findings are also interesting in that they revealed a mirroring of pathogen-induced RIPosome formation, but it is not clear how the RIP-IN-4 compound is inducing RIPosomes, which are naturally caused by RIPK2 activation. Are RIPK2 the degraders that were previously assigned as inhibitors actually acting as activators? Do all 9 RIPK2 inhibitors/degraders act through this mechanism? Are there any RIPK2 inhibitors that do not cause assembly and autophagy or is this universal? What do the authors think is causing CSK degradation by the RIPK2 compound, which appears to be even more substantial than the RIPK2 degradation (Ext 5c). Is there anything about the CLK1/MAP3K1 stabilization worth mentioning?

Version 1:

Reviewer comments:

Referee #1

(Remarks to the Author)

The authors have made commendable efforts to address all the reviewers' comments and have added a substantial number of new experiments.

The authors have done an incredible amount of follow-up work to identify the mechanisms of action of the three follow-up compounds. In particular, the BLK degradation story is remarkable, and it is indeed a shame that despite the authors' heroic efforts, they could not identify the target of the small molecule. It would have been so interesting to know how exactly TAK285 activates gamma-secretase to cleave BLK! But it would feel wrong to ask the authors to continue these efforts within this manuscript, which is already bursting at seams. Similarly, the RIPK2 degradation story is significantly improved and provides compelling evidence for the molecular mechanism behind RIPK-IN-4.

The inclusion of stabilizer part in Extended Data Fig. 3 was an interesting addition. The number of stabilization events (64 kinases and 128 different compounds) is quite significant compared to the destabilization events (66 kinases and 232 compounds), suggesting that about one-third of all stability changes involve kinase stabilization. This further argues for indirect "network" effects with many compounds.

My main remaining issues have to do with the larger context of the manuscript. In the end, it presents three distinct stories which are very loosely connected. As previously pointed out, the novelty of LYN degradation via CSK inhibitors is limited. Considering the length of the manuscript, I feel that it would be only strengthened by focusing on the more "developed" stories. I still think that this story would be better served as a separate manuscript. Moving the LYN part in the supplements would likely bury it completely. That said, I will let the editor and the authors discuss whether to include this part in the final manuscript if it proceeds to publication.

The second aspect is the practical conclusions of this work. Again, the amount of work put into this is truly impressive, but in the end the take-home message remains unclear. The authors mention that their results highlight "supercharging" endogenous degradation (or stabilization?) circuits. But this notion is quite vague and it is not at all clear to me what supercharging actually means, even after reading the manuscript. The underlying mechanisms are clearly very distinct, and it feels like an overreach to put them all under the umbrella of "supercharging".

Overall, I lean towards recommending accepting this manuscript for publication despite these shortcomings. Future studies by the Winter lab will hopefully solve the mystery of BLK degradation!

Below are additional comments related to the revision:

LYN degradation.

In the original manuscript, the authors demonstrated that SI-3 induces LYN degradation by CBL/CBL-B and suggested that Y316A is a new phosphodegron for CBL along with Y32, a known phosphodegron. To address more possible mechanisms underlying LYN degradation by SI-3, the author performed additional experiments including ubiquitination (K48, K62) assay, membrane association (IMP-1088 treatment) and Y508A mutant rendered LYN resistant to SI-3 degradation. As a result, the manuscript included a large volume of experimental data.

The authors wanted to emphasize that SI-3 is the only CSK inhibitor that induces LYN degradation. So, they added comparative Western blot of set of LYN binders. However, the manuscript still lacks a mechanistic explanation as to why SI-3 uniquely exerts this effect, which was requested in the earlier review comment. That said, as discussed above, this is not the most significant story of the manuscript so pursuing this further is not in my opinion essential at this point.

BLK degradation.

The authors proposed that TAK285 induces BLK degradation via γ -secretase-mediated cleavage of the N-terminal myristoylation, supported by KO screening and BioID. In the revised version, deep mutational scanning (DMS) was used to further investigate the N-terminal residues of BLK, and potential interactions with γ -secretase were explored through alpha fold prediction. The authors also showed, using MST, that TAK285 does not directly enhance the interaction between BLK and γ -secretase.

However, the key mechanistic question remains still unresolved: how TAK285 promotes γ -secretase-dependent cleavage of BLK in the absence of direct binding to either protein as the authors also mention in the Discussion.

Of note, the findings in Reviewer Fig. R5 are intriguing. The fact that 4-Hydroxylonchocarpin acts as a γ -secretase-dependent degrader, whereas FIN-1 does not, may offer mechanistic insights and deserves further exploration.

RIPK2 degradation.

The RIPK2 section was relatively well addressed in the original submission. The addition of immunoprecipitation data showing a RIPK-IN-4 induced interaction between RIPK2 and TMUB1 strengthens the proposed mechanism.

Testing of additional compounds including GSK583, RIP inhibitor 2, WEHI-345, Regorafenib, Gefitinib, GSK2983559, and LY2109761 was a valuable addition. Interestingly, only RIPK-IN-4 and GSK2983559 promoted foci formation and interaction with TMUB1. LY2109761 —despite not being a classical RIPK inhibitor— also induced foci formation and TMUB1 binding, which could offer new clues into the underlying mechanism.

Minor points

Extended Data Fig. 5i: Y507 should likely be Y508

Extended Data Fig. 3d,e: The label "GK" should be corrected to "C464W", as the gatekeeper residue in ABL is T315 and the corresponding mutation in the myristoyl pocket is C464W.

Figure 4k is difficult to understand in the absence of X axis labels. Does the X axis represent something or is the scatter there just to separate points better on the Y axis?

Referee #2

(Remarks to the Author)

The authors have done a commendable job of addressing the very large number of points raised in the three reviews, generating an overwhelming 44-page rebuttal (!), which includes a massive amount of new experimental data designed to address many of the issues the reviewers raised. They have added a significant amount of these new data in the revised version (e.g. in Figure 3 and in several of the supplementary figures). Many of the new results satisfactorily address concerns I had raised.

In my view, the most exciting new finding is that in response to TAK285 treatment γ -secretase cleaves the N-terminal myristoyl group from BLK, with the N-terminal 7 residues of BLK being sufficient to confer this response when grafted onto another totally unrelated protein such as BFP. Nevertheless, despite extensive efforts the authors failed to define molecularly how the TAK285 kinase inhibitor promotes γ -secretase cleavage of the BLK myristoyl group - does TAK285 activate cleavage of any other known γ -secretase target proteins? For instance, they were unable to identify the key kinase target for TAK285, determine the novel N-terminal residue of γ -secretase-cleaved demyristoylated BLK, or identify an E3 Ub

ligase responsible for soluble BLK degradation. With regard to the new data generated for kinase inhibitor-induced LYN and RIPK2 degradation, the authors' identification of BIRC2/cIAP and XIAP as E3 ligases necessary for RI-4-induced degradation of RIPK2 and the additional evidence that RIPK2 aggregates are cleared by autophagy strengthen the paper.

Points: 1. Although this might be beyond the scope, it would strengthen the authors' conclusions if they provided direct biochemical evidence that purified γ -secretase cleaves the myristoyl group from purified Myr-BLK. Their modeling of the Myr-BLK N-terminal sequence into the active site of γ -secretase and the alignment of the Myr-Gly bond with the catalytic Asp residues is quite persuasive, but modeling alone does not prove that the cleavage occurs at the Myr-Gly amide bond, and it would be important to demonstrate that the N-terminus of the soluble BLK is in fact Gly as is proposed; in the rebuttal the authors say they failed to define the novel N-terminus of released BLK, but perhaps this could be accomplished with the product of a biochemical cleavage assay. This information is relevant to the mechanism through which the soluble BLK is degraded in the cytoplasm. In their rebuttal, the authors mention the negative results they obtained with UBR1/2/4/5 KO intended to determine if released BLK is degraded through an N-end rule Ub ligase-dependent mechanism. Here it is worth noting that, if Gly were the new N-terminus (as posited in Fig R8), the released BLK protein will not be targeted by the N-end rule degradation pathway because N-terminal Gly is a strongly stabilizing N-end rule amino acid residue ($t_{1/2} = 30$ h). Based on the high stability of the G2A BLK, which is not membrane anchored, the Gly2 residue does seem to be important for degradation induced by TAK285. It should be a priority to determine the identity of the E3 ligase for soluble BLK and what it recognizes in the N-terminal 7 residues of BLK. In this regard, based on the known hydrophobic γ -secretase cleavage sites in APP that generate A β peptides, is it possible that γ -secretase cleaves the Leu-Val (or Val-Ser bond) bond in BLK? Consistent with this idea, substitution of Ser3Asn4 in the c-SRC N-terminal region with Leu3Val4 rendered it a target for TAK285/ γ -secretase induced degradation. If Myr-BLK were cleaved before Ser5, then it would be rapidly degraded by the N-end rule pathway, and if c-SRC Leu3Val4 were cleaved before Lys5, it would also be rapidly degraded by the N-end rule pathway.

2. Although beyond the scope of this paper, it would also be interesting to know if γ -secretase can remove the myristoyl group from other myristoylated SFKs, and possibly other myristoylated proteins, and, if so, what are the functional consequences for turnover of the demyristoylated protein. They showed that demyristoylation of BLK is dependent on the sequence of the first 7 aa of BLK and particularly the Leu3, Val4 and Ser6 residues ruling out most SFKs as targets, but can they predict what other Myr proteins might be targets for γ -secretase? For example, can a bioinformatics exercise be conducted on mammalian proteins with a Gly2 residue to predict additional possible myristoylated protein targets for γ -secretase based on the sequence determinants needed for Myr-BLK cleavage?

Referee #3

(Remarks to the Author)

The authors have returned an outstanding revision of their original manuscript with a great deal of new data and, importantly, new insights. I agree with the authors' assertion that the scale of their experiments is deeply helpful for understanding the frequency of these events (i.e. inhibitor-induced destabilization), which has clear implications for seeking out these effects in the process of drug discovery or, alternatively, avoiding them. I would call attention to the increased clarity surrounding gamma secretase mediated destabilization of BLK, which will be of keen interest to many and was thoroughly investigated with multiple orthogonal, unbiased, and data-rich experiments. Though the authors have been unable to identify the proximal target of TAK285 that induces gamma secretase to cleave BLK, the additional data in the revision will provide be a compelling start for future experiments. The increased mechanistic understanding of RIPK2-induced degradation is equally impressive. The authors have clearly put much effort into this revision and responded faithfully to our comments. The manuscript will be of widespread interest, and I would recommend its acceptance so the rest of the community can benefit from this excellent resource and collection of findings.

Referee #4

(Remarks to the Author)

I co-reviewed this manuscript with one of the reviewers who provided the listed reports.

Referees' comments:

Referee #1 (Remarks to the Author):

In the manuscript, Scholes et al. surveyed 1570 kinase inhibitors against 98 kinases, including 10 clinically relevant mutants, to systematically identify inhibitor-induced kinase destabilization. To date, several studies have shown inhibitor-induced degradation of kinases such as HER1 by neratinib or LMTK3 by C28. Chaperone deprivation is one of the suggested mechanisms for inhibitor-induced kinase degradation, but there is emerging evidence that the mechanisms can go beyond chaperone deprivation. However, there have not been systematic studies about the frequency, generalizability, and fundamental mechanistic principles of inhibitor-mediated kinase degradation, despite the key importance of kinase inhibitors in therapeutics.

Using a scalable luminescent reporter system expressing kinases fused to Nluc in K562 cells, the authors identified 232 compounds that downregulate protein levels of at least one kinase and 66 kinases affected by at least one compound. Most destabilized kinases were HSP90 clients (although see below). However, the authors suggest that many of the observed degradation events could not be explained by chaperone deprivation. The authors selected three kinases with a different HSP90 client status (LYN, BLK and RIPK2) for detailed follow up studies. The results point to three different mechanisms, where inhibitors potentiate endogenous degradation circuits by altering kinase activity (LYN), changing cellular localization (BLK) or inducing higher-order assemblies (RIPK2).

General comments:

Overall, the aim of the study is highly relevant and exciting, and the screening strategy is sound and well-controlled. The authors have done an impressive amount of work to identify inhibitors that degrade kinases and to characterize some of the mechanisms of action. However, the follow-up studies are either somewhat incomplete (RIPK2, BTK) or reveal mechanisms that have already previously been demonstrated (LYN). This unfortunately tempers my enthusiasm for this version of the study even if many experiments are technically very well done.

For example, unlike the RIPK2 case that likely involves on-target degradation, LYN and BLK degradation is not caused by direct inhibition of these kinases. Because LYN and BLK degradation are induced by inhibitors that might touch one of the upstream kinases or other targets, it is challenging to explain how they can achieve selectivity and how this could be used for e.g. rational design of “off-target degraders”.

My suggestion for improving the manuscript (if the authors choose to revise it for Nature) would be to focus much less on the LYN degradation story (or even remove it) but follow up in more detail on the RIPK2 and BTK parts. For RIPK2, it would be important to show at the mechanistic level how this works. Does the inhibitor promote a direct interaction between RIPK2 higher-order structures and autophagy regulators (akin to BCL6 degraders) or is there another mechanism at play? For BTK, how exactly does the inhibitor affect myristoylation and what is the mechanistic connection between the compound and the gamma-secretase complex? Is this due to off-target, non-kinase inhibitory activity of the compound?

I really like the idea of finding mechanisms for inhibitor-induced degradation, and I commend the authors on pursuing several leads. Their results that at least 2 of the 3 characterized

compounds lead to kinase degradation indirectly rather than directly suggest that there is likely much more to be discovered beyond the kinome. The authors “only” looked at the degradation of 98 kinases with the compounds, but if one was to extrapolate this to the entire proteome, it is likely that a significant proportion of compounds would lead to degradation of non-kinase off-targets (network targets). This is an intriguing thought.

We would like to thank Reviewer #1 for their kind remarks and positive feedback. We likewise believe that our discoveries will significantly expand beyond kinases and the kinome and are likely generalizable over many protein classes (as seen also with inhibitors/degraders of IDO1, see Hennes *et al.* biorxiv 2024 (under consideration in *Nature Chemistry*).

We also very much appreciate the feedback on our manuscript. As will be more apparent in the ensuing sections, we did follow this Reviewer’s advice, performed additional analyses and added additional mechanistic data to both RIPK2 and BLK and have generated a considerable amount of data (some of which we could not even add to the manuscript due to space constraints). On a high level and before going into details on specific critiques raised, we want to share our perspective on some of the points raised:

- With regards to working up the mechanism of action of drug-induced LYN degradation, we completely agree with the notion that aspects of this mechanism have been described before and that it is fair to argue that the compound taps into an established mechanism of action. However, we would like to clearly state that, from a pharmacologic and chemical-biology centric perspective, the intriguing novelty is that we identify a compound that can co-opt this mechanism. There are many (!) kinase inhibitors that bind SRC family kinases, yet SI-3 is unique in its capability of inducing LYN degradation given its differential potency on CSK over LYN that was most strikingly revealed by our chemoproteomics and cellular target engagement studies. Hence, we felt that the unbiased mechanism of action study around SI-3 still warrants publication (including some modifications we have made in response to reviewer requests).
- We would have loved to also employ structural approaches, akin to the BCL6 study mentioned by this Reviewer, to further increase our understanding of the mechanistic underpinnings of RIPK2 and/or BLK degradation. In fact, we even attempted structural studies involving the γ -secretase in collaboration with the Shi lab but did not manage to acquire conclusive data. Having stated this, we would also like to put our work in relation to the BCL6 manuscript or other recent degrader mechanism of action studies published in *Nature* (such as Xie *et al.*, *Nature* 2025 and Yeo *et al.*, *Nature* 2025). In all of these cases, the stories dissected the mechanism of a single small molecule. Each of these compounds had also already been identified to act as degrader before, so (while all these papers are beautiful and impactful), they assigned one mechanism to a degrader phenotype that had previously been established. In contrast, the goal of this study is to contribute to a conceptual understanding of an entire class of degrader mechanisms where we have identified >400 new degrader events that have not been reported before, could make systems-level conclusions and finally went into mechanistic detail with three independent (never reported) drug-induced degrader mechanisms. Based on the results of these validation experiments, and supported by data reported in Hennes *et al.*, biorxiv 2024, we found that inhibitor-induced degradation events tap into native turnover mechanisms as a common denominator. Given that, I would hope that this Reviewer agrees that we can’t provide the same mechanistic depth on each of these findings compared with the aforementioned studies.

Other major points:

1. Kinase Degradation (KinDeg) map

1.1. The authors mention that many of the observed degradation events cannot be explained by chaperone deprivation. However, in Fig 1C, the authors describe that “both strong and weak clients of Hsp90 had a markedly higher prevalence of being destabilized than non or not-defined clients, pointing to an outsized contribution of chaperone deprivation to the observed degradation events”. To clarify this, the authors should discuss what fraction of hits could be due to the chaperone deprivation model. Moreover, the graph and the statement lack statistical support.

We apologize for the misleading wording. We have rephrased the section to clarify the two separate findings: (I) chaperone deprivation is one likely mechanism for inhibitor-induced kinase degradation, specifically for client kinases. This is reflected by the significant enrichment of HSP90 clients amongst our hits. Nevertheless, (II) our mechanistic work-up for RIPK2, BLK and LYN highlights that, even for HSP90 client kinases, inhibitor-induced destabilization can also occur via mechanisms that are unrelated to chaperone deprivation. This has been clarified in the revised version of the manuscript (p.4).

Prediction of mechanisms acting via chaperone deprivation on a case-by-case basis is no straightforward task. As exemplified in the paper, BLK is susceptible to chaperone deprivation induced degradation (downregulated upon HSP90i treatment) but at least two of the identified inhibitor-degrader hits do not induce degradation via this pathway.

In summary, HSP90 client status provided the most significant predictor for inhibitor-induced degradability, highlighting chaperone deprivation as a central mechanism. Beyond the scope of this manuscript, future work in the lab will address individual chaperone deprivation examples on a more mechanistic level, particularly focusing on E3 recognition after HSP90 dissociation.

We added the p-values to **Fig 1c** as requested.

1.2. Kinases that do not interact with Hsp90 are generally more stable. Perhaps they are inherently less susceptible to degradation, explaining the trend that the authors observe? This could be an alternative explanation to the active chaperone deprivation model.

To test this, we compared the distribution of hit kinase half-lives derived from our screening data per client category using a Kruskal-Wallis test and did not find any significant differences (p-value 0.17008).

1.3. Maybe I missed this, but what fraction of degradation hits is due to network modulation vs direct targeting?

This is a great point raised by Reviewer 1 which in fact we did not address in depth in the index version of our manuscript. To provide generalizable insights, we have made considerable efforts in extending our computational analyses, as outlined in the following:

Overview: First, we collected publicly available kinase-inhibitor binding data and assessed the number of hits vs published binders. Secondly, we constructed a kinase protein-protein

interaction network and together with the concatenated binding data evaluated the probability of a network driven degradation event. In total, we could attribute 86 out of 404 hits (21%) to either being likely driven by network effects (57) or induced via direct compound binding (29). The rather low number of such “mappable” events stems from the rather incomplete binding data. Despite our best efforts to collate as many databases as possible, comprehensive binding data such as resulting from Kinobead profiling is not available for most of the inhibitors used in our screen. For example, only 56 out of 232 hit compounds have such data available (see **Supplemental Data Table 1**). In addition, we leveraged the inherent polypharmacology of kinase inhibitors to increase the number of engaged kinases by maximizing the drug concentration during the screening process while avoiding toxicity or other screening artefacts by performing a drug pre-screen (see **Materials and Methods**). Hence, these numbers, especially with respect to events directly caused by *validated* binding events, are likely an underrepresentation of the true number.

(I) Construction of drug binding profiles: To construct the comprehensive drug binding profiles, we gathered kinase-inhibitor binding data from the CC v.2024 (Duran-Frigola et al. Nat Biotechnol 2020, 836,654 compounds x 6,933 features, 6,191 features exc. ‘classes’), DrugBank (7,063 compounds x 4,677 protein targets), Douglass et al. Cell Reports Medicine 2022 (32 compounds x 46 protein targets), Heinzke et al. Scientific Data 2024 (binarized at max. pChEMBL 6; 413,171 compounds x 972 protein targets) and Reinecke et al. Nat Chem Biol 2023 (following the binarization procedure applied in Comajuncosa-Creus et al. Nat Commun 2024, 163 compounds x 73 protein targets). In total, we collected binding data for 876,497 compounds encompassing 9,835 targets (9,093 targets excluding ‘classes’) and 5,488,587 compound-target pairs (1,321,623 pairs excluding ‘classes’).

We identified at least one target for 158 out of 232 compound hits (mean: 10.2 targets, median: 5 targets), totaling 271 destabilization events (out of 404) and 51 kinases (out of 66). Indeed, hits were slightly enriched in compounds with binding data (i.e. having at least one reported target; OR 1.39, p-val <0.05). Overall, 29 out of 271 destabilization events (10.7%) could be explained by direct target binding, encompassing 26 hits and 12 kinases (**Reviewer Fig R1**).

Fig. R1: Number of degraded kinases (y-axis, blue) and binding targets (orange) for each hit compound (x-axis). Mutated kinases are depicted with blue bars with black edges.

(II) Kinase network construction: To assess network mediated effects, we gathered human pathway data from KEGG, totaling 362 pathways and 8,538 genes. We built a global network (i.e. with all the pathways merged) with the Networkx Python package, considering the gene, enzyme, group and ortholog entry types (nodes) and their corresponding relations (edges). The

network was randomized 100 times for comparison, preserving the node degree and randomizing the edges. We then computed the shortest path between each drug target and the degraded proteins using both the global and the randomized networks. As expected, in some cases, the shortest path distance between inhibitor targets and destabilized proteins was lower in the real network than in the randomized counterparts, showing a potential effect through the network connectivity (**Reviewer Fig R2a**). We explored different values to define whether a network effect could be considered significant (**Reviewer Fig R2b**). We finally fixed the threshold on 20%, meaning we considered significant those target-destabilized protein pairs where the length of the shortest path in the real network is lower than 80% of the randomized networks for this specific pair. Using this threshold and once we exclude the 29 direct hits from above, we found that 57 of the 242 destabilization events (23.6%) could potentially be explained through network modulation effects (**Reviewer Fig R2b**). In addition for 68 (28.1%) we could not identify a path between drug targets and degraded proteins in this global network. Please, note that, to make a more accurate assessment, we would need global networks with directionality information (i.e. which proteins are up- or down-stream of the target kinases). However, unfortunately, this information is scarce and not systematic, making it not useful for global assessments.

We at this point in time have not included above analysis in the manuscript. Given that many of the assayed inhibitors lack unbiased binding data, these analyses are inherently incomplete. This results in caveats to the assigned numbers which, despite our best efforts, may not reflect the true proportion of direct and network-mediated hits. In addition, as showcased by LYN, SI-3 clearly engages LYN, however, degradation is driven by SI-3's capacity to preferentially engage the upstream kinase CSK. In this scenario guilt-by-association hence would be incorrect. Globally, we find that features can be deduced only if sufficient (well-characterized) chemical matter is identified. For example, in both top scoring kinases HER2 and ABL1, we identified sufficient hits either annotated as direct binders (HER2) or binding to a common network-associated target (dual PI3K α and mTOR binders for ABL1). With increasing numbers of globally profiled inhibitors, we are optimistic that future studies can and will address this important question.

Fig R2. a Normalized distributions of shortest paths between hit drug targets and degraded proteins in the global network (blue) and the x100 randomized ones (orange). For 68 out of 242 destabilization events, no path was found between both entities, leading to 174 paths. **b** Number of degradation events (y-axis) that can be explained by network modulation in terms of the established threshold to define a target-degraded protein pair as significant (having a shortest path in the real global network longer than the threshold value among the x100 randomized networks).

1.4. Which kinase inhibitors are included in 1570 inhibitors? How did authors select the inhibitors and the kinases?

We chose a commercially available drug screening library with good kinase target distribution, structural diversity and cell permeability and biologic activity. As mentioned in the manuscript, compounds that were affecting cellular viability or showing assay interference were excluded. Kinases were manually selected based on several criteria. First, kinases with chemical binders and clinical relevance formed the core group (such as HER2, BTK, MET). Secondly, we added a set of understudied kinases, also commonly referred to as the dark kinome (such as DYRK1B/2, RIOK1, CDK10, LMTK2; <https://doi.org/10.1093/nar/gkaa853>). In total, we attempted to cover all kinase families, and with exception of the STE branch, assessed at least one member family each. Finally, we pre-tested expression levels of each of our available kinase constructs, omitting constructs yielding too low luminescent levels (e.g. ERBB4) from the screen.

It would be interesting to see if the screening results have any trends based on the class or type of kinase inhibitors, for example, allosteric vs ATP-competitive, selective vs promiscuous, type I vs type II.

This is an interesting point raised by Reviewer 1, which we gladly followed up in detail. In brief, we determined selective/promiscuous inhibitor classification based on the number of targets associated to each inhibitor. The extent to which hit compounds are enriched in promiscuous inhibitors mostly depends on the established threshold used to classify an inhibitor as selective or promiscuous. Since no “true” cut-off exists, we have plotted the continuous data below (see **Reviewer Fig R3a, b**) and we overall find a biphasic OR distribution. As an example, setting the threshold to 20, the observed OR is around 2.2 with a p-value of <0.05, indicating an enrichment for promiscuous probes. In compound numbers this translates to 14 out of 156 hits ($\approx 9.0\%$) being promiscuous and 34 out of 798 of the non-hits ($\approx 4.3\%$) being promiscuous. The Fisher test was done with Scipy (Virtanen et al Nature Methods 2020). In sum, our analysis reveals a modest enrichment for promiscuous inhibitors. Since it is rather context (cutoff-) dependent, we opted against including this statement in the revised version of the manuscript. If this Reviewer thinks it should be included, we are happy to do so in alignment with the editor.

Fig R3. **a** Number of destabilized kinases (y-axis) in terms of the number of binding targets (x-axis) for each hit compound. **b** Fisher Test Odds Ratio (y-axis) indicating the enrichment of hit compounds in promiscuous inhibitors at different values of the number of targets used to classify an inhibitor as promiscuous. Bars are colored according to the Fisher Test p-value.

Further, given the lack of comprehensive binding mode annotations, we manually annotated our hit compounds for their putative Type I vs II and allosteric binding mode utilizing the available literature data, structural properties of the inhibitors as well as structural data where

available. We have added this dataset to the manuscript (see **Supplementary Data Table 1**). To test for enrichment of any specific binding mode, we cross-assessed our hits with the PKIDB database. No enrichment for either inhibitor type could be detected. We have added the corresponding information to the manuscript on **p.4**.

1.5. The authors state that the reporter half-life did not correlate with the frequency of scoring, but do not show the p-value of the correlation. Also, the statement seems quite strong, and one could also reasonably say that -0.381 is a decent correlation for such a dataset.

Thank you for this comment. We have now rephrased the relevant sentence for clarity and have included the p-value for the reported correlation (**p.3**). While the correlation is statistically significant, the effect size is relatively modest. We believe that this finding is more plausibly explained by a slight bias toward shorter reporter half-lives rather than indicating a strong biological relationship.

1.6. Did the authors find any kinase stabilizers?

Indeed, our results have revealed several kinase stabilizers. To follow up on this Reviewer's request, we have now significantly extended the analysis and have also conducted experimental validation. Corresponding figures, documents and methods have been added on **p.4**, **Supplementary Data Table 1** and **Extended Data Fig 3**.

Akin to our original destabilization analysis, we systematically analyzed the number of significantly deviated time points of our kinases per treatment. We identified 204 stabilization events, totaling 64 kinases and 128 different compounds.

To cross-validate our findings on protein stabilization, we focused our efforts on ABL1, which was stabilized by DPH. DPH is a small molecule that binds to the allosteric, myristic acid binding pocket of ABL1. We could successfully confirm stabilization in our stability reporter setup in a dose-dependent manner. Further, stabilization was abrogated by mutating C464W a critical residue within the myristoyl binding pocket (<https://pmc.ncbi.nlm.nih.gov/articles/PMC7566958/>) highlighting a mechanism of action induced by direct drug binding. We next sought out to test other allosteric ABL1 inhibitors (asciminib and GNF-2) to generalize this phenomenon. Both had been excluded in our drug screen (see **Materials & Methods**) due to their toxicity in the BCR-ABL driven screening cell line K562. Both asciminib and GNF-2 displayed identical behavior to DPH, stabilizing ABL1 in a dose dependent manner, which was again abrogated by the C464W mutation (see **Extended Data Fig 3d, e**). For the remainder of stabilization events, we are currently dissecting multiple leads for a follow-up study. All associated data has, however, already been made available in above-described documents and datasets.

2. LYN kinase

2.1. The authors suggest that SI-3, a CSK kinase inhibitor, induced rapid LYN destabilization by inhibiting CSK kinase, a negative regulator of SRC family kinases including LYN. However, it was already known that genetic depletion of CSK decreased the protein level of LYN kinase in a Cbl-dependent manner (PMID 32376886). Because SI-3 is a CSK inhibitor, it is predictable that SI-3 affects the abundance of LYN kinase. Unfortunately, this significantly reduces the novelty of the authors' finding.

As mentioned above already, we believe that the novelty of our findings emerges from the identified compound “SRC inhibitor 3 (SI-3)” and less from the biology. Critical for phenocopying above-described intrinsic LYN degradation via the CSK-mitigated activity-stability switch is the establishment of a small molecule that is sufficiently selective for CSK over LYN. There is a large panel of inhibitors that target CSK, but with SI-3 we have uncovered a small molecule that uniquely fulfils this key feature and has a sufficient preference *in cellulo* to inhibit CSK over LYN. We thus report the first full chemical solution to tap into LYN’s intrinsic activity-stability switch. SI-3 can hence be utilized as a tool compound to explore the biological context of rapidly ablating LYN protein levels without requirement of genetic engineering (as e.g. needed for the chemo-genetic system referenced in the original manuscript PMID 31282857) and on timescales in the order of minutes.

Engineering CSK selectivity has been a challenge and also already tackled by the field of TPD by engineering heterobifunctional degraders such as recently described in PMID 33275901. However, despite selectively degrading CSK as assessed by expression proteomics, DB-3-291 does not induce LYN degradation. Given the use of Dasatinib as the warhead for kinase engagement, we would hypothesize that Dasatinib still inhibits LYN, thus protecting it from activation analogue to our chemical rescue experiment shown in **Extended Data Fig 5g** using Nintedanib.

Finally, SI-3 mediated LYN degradation is, to the best of our knowledge, the first degrader molecule to this date shown to act via two degradation pathways. Intriguingly, in addition to the E3 ligase CBL, we found that membrane anchorage is required for lysosomal processing (added as **Extended Data Fig 5e**). We further uncovered Y316, a residue frequently phosphorylated in proteomics/phospho-proteomics upon endogenous LYN activation that acts in conjunction with Y32 as phospho-degron.

2.2. CSK is a negative regulator of SFKs (SRC family kinases), including LYN, FYN, and SRC. The authors showed that CSK inhibition by SI-3 leads to activation of LYN, followed by degradation via CBL. What about the other CSK inhibitors in the dataset? Could they induce LYN destabilization? If not, what makes SI-3 selectively affect LYN kinase over other SFKs and inhibitors?

We cross-checked all inhibitors in our library. Out of 25 LYN inhibitors, 8 have also been reported to inhibit CSK (see data in **Extended Data Fig 4d** and relevant text section on p.6), but fail downregulate LYN in our screening campaign. To orthogonally validate this, we selected a subset of these inhibitors known to target LYN and CSK (ONO-4059, R406, Dasatinib and Ibrutinib), as well as manually selected inhibitors reported to engage CSK more potently than LYN (e.g. TAK285, PLX-4720). We blotted for endogenous LYN levels after 2h of treatment but did not identify any reduction (added as **Extended Data Fig 4e**). This again highlights the unique capacity of SI-3 to exquisitely downregulate LYN stemming from the large differential inhibitory capacity of CSK compared to LYN. Finally, considering other CSK regulated kinases, none of the other family members have been associated with such a rapid depletion upon CSK ablation (e.g. shown in <https://elifesciences.org/articles/46043>).

2.3. In addition to LYN, SI-3 treatment affects several other proteins, as shown by proteome profiling (Extended Fig. 3A). CSKN1D and SIK3 are other potentially degraded kinases, as are non-kinases TPRN and AMD1, among others. Clearly, the compound affects many other

targets. The authors should comment on this. Are these also due to activating their endogenous degradation circuits or due to off-target effects of the inhibitor?

We thank Reviewer 1 for bringing this to our attention. In our experience of having profiled many small molecules for proteome-wide effects, there are certain sets of proteins that are frequently affected (up- or downregulated) in patterns that are unrelated to the target or mode of action of the compound. In many cases, these are short-lived proteins, which is here the case for TPRN and AMD1 (see <https://doi.org/10.1016/j.molcel.2021.09.015>). Large-scale proteomics studies have in fact started to map sets of these “frequent hitter” proteins, which for instance also include the aforementioned CSNK1D (see [decryptM, https://www.proteomicsdb.org/protein/55930/proteinddfp](https://www.proteomicsdb.org/protein/55930/proteinddfp)).

2.4. If SI-3 supercharges LYN degradation circuits by altering kinase activity, as the author suggest, they should compare SI-3 to other CSK inhibitors to generalize this phenomenon.

As shown through additional experiments (see 2.2), no other CSK inhibitor prompted LYN degradation, highlighting the uniqueness of SI-3. We hence could not follow up this request.

2.5. The data for the neddylation inhibitor (MLN4934) is missing (Page 6).

We apologize for the oversight and have replaced the respective **Fig 2d** with the complete set of rescue compounds.

2.6. In Fig 2G, did authors confirm depletion of CBL and CBL-B in either protein level or mRNA level?

Yes, we confirmed CBL depletion by immunoblotting (added as **Extended Data Fig 4j**) and confirmed the genetic disruption of CBL-B via TIDE (**Reviewer Fig R4**).

Fig R4. TIDE analysis of CBL-B and DKO (CBL and CBL-B) for sgRNA targeted CBL-B DNA locus.

2.7. In Fig 2J, the authors showed that double mutant of LYN (LYN Y32A+Y316A) was almost completely inert to SI-3-induced degradation, and the degradation of Y32A mutant is CBL dependent (Extended Data Fig 3N). What about the Y316A? Can Y316A recover SI-3-induced degradation of LYN in CBL or CBL-B KO conditions? Or are there another E3s that recognize the Y316 phosphodegron?

For the Y316A mutant we observed continued dependence on both E3 ligases. We have added the data to the manuscript in **Extended Data Fig 5k**.

3. BLK kinase

3.1. In the Extended Data Table 1, 9 kinase inhibitors induce BLK degradation, and 7 inhibitors selectively affect BLK. But only 4 inhibitors (Afuresertib (AKT), BP-1-102 (STAT), 4-Hydroxylonchocarpin (p38 MAPK, JNK and ERK) and TAK-285 (HER1/2) are highlighted as a BLK degraders in the supplementary file.

We thank the reviewer for identifying this detail. During the export of the trajectory files as a summarizing .pdf, some of the text boxes must have been disrupted hence disabling the word search function in a subset of cases. While all plots for BLK were included (9) in the original file, we have now edited the file to configure all text boxes correctly.

All these inhibitors selectively induced BLK degradation even though their targets are different. Did authors test if these inhibitors have shared mechanism to induced destabilization of BLK (at least in respect to myristoylation or gamma-secretase dependency)?

We expanded the scope of our search amongst our BLK hit compounds (see **Reviewer Fig R5**) and identified 4-Hydroxylonchocarpin, a chalcone reported to act as ERK activator as additional γ -secretase dependent degrader. Further we could show that FIIN-2, an FGFR inhibitor degraded BLK independently of the γ -secretase. Since BLK is sensitive to chaperone deprivation, FIIN-2 could exert its function via this route, but we did not investigate this further for this manuscript.

Given severe space constraints, we can show these data only as a Reviewer Figure. If this Reviewer believes it is essential to show these data in the final version of the manuscript, we are prepared to make the required changes in alignment with the editor.

Fig R5. Flow cytometric assessment of KBM7 iCas9 BLK stability reporter cells treated for 2h with DMSO or 12.5 μ M DAPT, followed by 8h with 2.5 μ M TAK285, 10 μ M 4-OH (4-Hydroxylonchocarpin), 2.5 μ M FIIN-2 or 10 μ M HSP90i (Two-way ANOVA with a Tukey's multiple comparison test, ns > 0.9999, n = 3).

3.2. The authors show that TAK285-induced BLK degradation is dependent on UPS by using TAK243 and carfilzomib. When the myristoylation of BLK is reduced upon TAK285 treatment, is ubiquitination of BLK increased?

We attempted HA-ubiquitin overexpression coupled to BLK-FLAG co-immunoprecipitation in HEK293T cells to test for enhanced ubiquitination upon TAK285 treatment. However, even in the case of the positive control compound (PROTAC TL12-186) we could not detect any differences in ubiquitination due to technical difficulties. Of note, TAK285-induced BLK degradation (but not TAK285-induced localization into the cytosol) can be rescued with the E1 inhibitor TAK243, insinuating that ubiquitination is not required for upstream γ -secretase

recognition and cleavage but for downstream proteolytic processing (see **Extended Data Fig 7a**, from initial submission).

Among the enriched genes in the CRISPR/Cas9 screen, were there any E3s that might recognize un-myristoylated BLK? Work by Elledge lab has suggested that ZYG11B and ZER1 are involved in degradation of such substrates (PMID: 31273098). Maybe the authors could look into these E3s?

In the original genome-scale CRISPR screen, we could not identify a putative E3 ligase amongst the top 30 most significant hits. To answer the Reviewer's question, we performed another CRISPR screen, but in this case using a ubiquitin focused library to enhance the resolution within E3 space and allowing us to screen at two different timepoints after dox induction of gene editing: 96h (similar to the original genome-wide screen) and an extended timeframe of 168h prior to drug treatment. In both setups we could recover E3s, but the data did not suggest a strong genetic rescue. We attempted the validation of individual hits but did not identify any rescue with meaningful effect size (see below, **Reviewer Fig R6a-e**).

We agree with the Reviewer's point that the elusive E3(s) would likely be involved in recognizing the remnant N-terminal residue after cleavage. Hence, we tested all N-end rule E3 ligases (UBR1/2/4/5) and the suggested E3 pair ZYG11B and ZER1 (as double knockout) in arrayed experiments (see **Reviewer Fig R6f**). Since UBR4 showed a mild rescue for the treatment with the myristoylation inhibitor IMP-1088, we also cross-tested a set of double KOs, combining hits from the ubiquitin focused CRISPR screens and UBR4 (see **Reviewer Fig R6e**). However, none of the generated KOs, provided sufficient rescue. As genetic screens often fail to resolve genetic redundancies, proteomics discovery approaches can offer an alternative. Unfortunately, also an additional round of BioID experiment failed to identify candidate E3s for mechanistic follow-up studies beyond the ones highlighted in **Reviewer Fig R6**.

In sum, despite our best efforts (2 sets of pooled genetic screens, hypothesis-driven genetic perturbations and proteomics-based approaches), we could not identify the UPS effector(s) responsible for BLK turnover after γ -secretase cleavage, pointing to a deconvoluted mechanism that would require further in-depth research independent from this manuscript. Given the severe space constraints, we are not able to include these negative data into the Extended or Supplementary Material.

Fig R6. UBI-focused CRISPR/Cas9 screening data for sgRNAs enriched in low and high sorted fractions compared to sgRNAs identified in the mid fraction in KBM7 iCas9 BLK stability reporter. Putative hits are highlighted in pink per condition (dox 96h or 168h, DMSO (6h), TAK286 (2.5 μ M, 6h) IMP-1088 (1 μ M, 24h)), essential (low) in dark grey and proteasome subunits (high) in teal. **e-f** KOs for respective genes or gene pairs analyzed after 168h of Cas9 induction (via dox) in KBM7 iCas9 BLK stability reporter cells for DMSO, TAK285 (2.5 μ M, 6h) or IMP-1088 (1 μ M, 24h) treatments by flow cytometry (n = 3). **(e)** Hits from UPS-focused CRISPR/Cas9 screen +/- KO of UBR4. **(f)** Targeted KO of all putative N-end rule E3 ligases.

3.3. Unlike BLK, expression levels of non-myristoylated (G2A) mutant forms of SRC were 4- to 6-fold higher than those of their myristoylated counterparts (PMID 20584982). To further support that non-myristoylated BLK is unstable (unlike SRC), the authors could to compare the stability of BLK G2A to WT.

We thank this Reviewer for pointing this out. As a consequence, we generated the suggested BLK G2A mutant which, as hypothesized, is not degraded by TAK285 and has a higher stability than BLK WT (see **Reviewer Fig R7a, b**). Further and as expected, BLK G2A is neither myristoylated nor anchored into the membrane (**Reviewer Fig R7 c, d**) hence explaining the lack of additional destabilization when subjected to TAK285 or IMP-1088 treatment. Of note, this increased stability is also in line with our hypothesized mechanism. Mutating the second glycine residue, ablates the myristic modification (see **Reviewer Fig R8**) but also generates a novel, and as shown for SRC, more stable N-terminus. Collectively, this supports that cleavage of myristic acid by γ -secretase releases a novel N-terminus, such as the terminal glycine, which is then recognized by yet to be determined E3 ligase(s).

Fig R7. a Flow cytometric assessment of stability reporters in KBM7 iCas9 cells for indicated BLK variants treated with either DMSO, 6h TAK285 (2.5 μ M) or 24h of IMP-1088 (1 μ M) (n = 3). **b** Cells from (a) normalised to BLK^{WT} stability reporter (n = 3). **c** FLAG IP for 24h AMA treated K562 BLK^{WT}-3xFLAG or BLK^{G2A}-3xFLAG cells followed by on-bead TAMRA azide click (n = 3). **d** Representative images of BLK^{G2A} stability reporter transduced into RKO iCas9 cells (scalebar 25 μ m).

As we will describe in detail below, to decode BLK's sequence requirements of TAK285-mediated degradation, we additionally performed a FACS-based saturated mutagenesis screen. Within this dataset, we also identified G2 as one hotspot that impairs both TAK285 as well as IMP-1088-mediated BLK degradation. Specifically, when normalized to vehicle control G2I and G2L scored as our top enriched mutations (see **Fig 3f, Extended Data Fig 7g**). We thus characterized these mutations extensively for the manuscript, first cross-validating that G2I and G2L, as G2A, are resistant to both TAK285 as well as IMP-1088-mediated degradation (**Extended Data Fig 7h**). Next, as expected imaging revealed that G2 mutants are cytosolic (**Extended Data Fig 7k**) and not myristoylated (**Extended Data Fig 7i, j**). To note, G2A scored as top stabilizing residue for the DMSO (vehicle control) condition in the screening setup (see **Supplementary Data Table 4**). However, in the manuscript, data are depicted for TAK285 and IMP-1088 normalized to DMSO to counteract changes in baseline stability (**Fig 3f, Extended Data Fig 7g**) thus yielding G2I and G2L as top scoring mutations with respect to drug induced stability changes. For brevity we have only included these two mutations in the main and extended figure panels.

Fig R8. Model and overview for N-terminal processing of BLK in steady state and upon TAK285 induced degradation. **Top:** Sequence of BLK unique domain. Cellular processing leads to M1 being cleaved off and G2 being myristoylated, generating the membrane anchored BLK. **Bottom:** TAK285 induces γ -secretase processing of peptide bond between the myristic acid and G2 (see AlphaFold3 model in **Fig 3i**) revealing a *neo* N-degron that leads to proteasomal degradation of BLK.

Since we were interested in the detailed genetic dissection of TAK285 mediated degradation, we generated several fusion and truncation variants of BLK and SRC. First, we could show that fusion of BLK’s unique domain to SRC enables TAK285-mediated, γ -secretase dependent SRC degradation (added in **Fig 3e**). Moreover, when “engrafted” on BFP, the unique domain itself is sufficient to generate a BFP fusion that is degraded by TAK285 again in a γ -secretase dependent mechanism (see **Fig 3g**). Given the importance of these first 57 amino acids, we hence performed aforementioned saturated mutagenesis screen in which each individual position is mutated to all other possible amino acids. We coupled each variant to our stability reporter system and akin to our CRISPR screen setup sorted for the cellular populations with “high” or “low” BFP levels after vehicle, TAK285 or IMP-1088 treatment (added as **Fig 3f**, **Extended Data Fig 7g**).

The results highlighted a clear importance of residue G2 for both TAK285 as well as IMP-1088 mitigated BLK degradation, which culminates on the loss of the myristic acid modification, membrane dissociation and simultaneous occurrence of a *neo* N-terminal residue as detailed in above sections.

Focusing on TAK285-specific residues, we identified L3, V4 and S6 as key positions. As already highlighted in the initial manuscript submission, S6 is critical for myristoylation status. Corroborating our initial findings of this variant’s cytosolic localization (moved to **Extended Data Fig 7d**) as well as decreased baseline stability (moved to **Extended Data Fig 7f**), we measured the myristoylation status of S6A using the alkynyl myristic acid chemical probe. Indeed, compared to wt BLK, S6A has strongly impaired myristoylation (added as **Extended Data Fig 7i, j**). Critically, we extended our truncation analysis further and showed that residues #1-7 are sufficient to retain TAK285-mediated BLK degradation (see **Fig 3g**).

Fig R9. Sequence alignment of BLK and SRC N-termini. Light teal highlights identical residues in both proteins, which are critical players impacting myristoylation status. Residues in light pink highlight diverging residues which are critical for γ -secretase processing.

Focusing on two additional key positions in our DMS dataset, L3 and V4, we identified multiple mutants that are (a) resistant to TAK285-mediated degradation (**Extended Data Fig 7h**), while (b) retaining sensitivity to IMP-1088 treatment. Further, the assessed mutants are (c) myristoylated to at least the same degree as BLK (**Extended Data Fig 7i, j**) and in line with this data (d) localized to the membrane (**Extended Data Fig 7k**). When compared to SRC's sequence we noted that L3 and V4 form an aliphatic patch in BLK (**Reviewer Fig R9**), which we hypothesized may be critical for γ -secretase processing. We hence mutated the corresponding residues in SRC to those of BLK. Indeed, we were thus able to generate a SRC variant that was degraded by TAK285 in a γ -secretase dependent manner (see **Fig 3h**, **Extended Data Fig 7l**).

Our initial BioID data already pointed towards an intrinsic affinity between BLK and γ -secretase (**Extended Data Fig 6i**). We now further substantiate evidence for this interaction *in vitro* with data from MST measurements with recombinant proteins (see **Extended Data Fig 6j**). This motivated the development of a structural model via Alphafold3 (see **Fig 3i**). After minimal relaxation of the system via MD simulation, we found that within the resulting complex, BLK's unique domain is positioned within the active site of γ -secretase. Specifically, the peptide bond between the myristic acid and G2 is coordinated by two hydrogen bonds formed by the two catalytic residues D385 and D275 of γ -secretase. Furthermore, we noted that L3 is positioned within a neighboring hydrophobic pocket, corroborating our mutational data and this residue's important role to enable γ -secretase processing upon TAK285 treatment. Of note, we also attempted structural elucidation but could only observe heterogenous complex formation, which disabled the generation of a high-resolution structure.

Given the structural prediction, the extensive mutagenesis and truncation data as well as new literature evidence highlighting that γ -secretase can cleave myristic acid from small molecules (<https://doi.org/10.1021/acscchembio.4c00432>), our mechanistic hypothesis thus places TAK285 as small molecule that induces the γ -secretase dependent cleavage of myristic acid from BLK, releasing it into the cytosol where the generated neo N-terminus serves as an N-degron (see **Reviewer Fig R8**).

Since the peptide bond of the myristic acid to G2 is specifically coordinated by the two active site aspartic acids, we hypothesized that this is the site for initial cleavage by γ -secretase. As shown in the original submission, myristic acid is lost from BLK (**Fig 3i**) upon TAK285 treatment and in a γ -secretase dependent manner, which supports this notion. Given the positioning of BLK on the cytosolic side of the secretase complex (rather than being a transmembrane protein), as well as the per-residue decomposition of binding free energy (see **Reviewer Fig R10**), in which loss of the myristic acid drastically increases the energetic contribution of G2, it is most likely that cleavage results in the release of BLK with a remnant glycine into the cytosol. Despite our best efforts, we were, however, not able to identify via MS the specific cleavage site.

Fig R10. Per-residue decomposition of binding free energy (ΔG) calculated using MM/GBSA from 50 ns MD simulations. Simulations were run on the predicted model either starting from the mature BLK (myristoylated (MYR) G2 (teal) or the non-myristoylated BLK starting from G2 (pink).

Finally, we set out to identify the kinase mediating BLK degradation. Previous chemical competition experiments (moved to **Extended Data Fig 6k**) already informed that direct BLK binding is not important. To exclude involvement of direct binding at the effector (γ -secretase), we conducted MST experiments, where TAK285 was dose-dependently titrated against purified γ -secretase. However, no binding was detected (**Extended Data Fig 6l**). Collectively, these data are in line with a network-mediated effect of TAK285. To elucidate this, we performed multiple attempts to map potential targets using chemoproteomics. In addition to the Kinobead profiling data from the index submission, we now have synthesized a tethered analog of TAK285 for immobilization to conduct direct affinity enrichment followed by proteomics analysis. While this analog retained BLK degradation activity (data not shown), in-depth analysis failed to reveal targets beyond MAP2K5 which we had already identified via Kinobeads profiling. We hence set out to validate MAP2K5, but genetic KO did not alter TAK285 degradation (added as **Extended Data Fig 6n**).

Given that TAK285 is a kinase inhibitor, we would suspect phosphorylation to be the most likely PTM to occur and hence enable γ -secretase processing of BLK. The mutational data would suggest S6 as one potential site. We hence re-visited our genome-wide CRISPR/Cas9 screening data, mapping all kinases, all concatenated chemo-proteomic hits, all as well as the top 10 kinases predicted to phosphorylate S6 within BLK (see **Reviewer Fig R11**). However, no noteworthy hits could be identified. MS identification as well as experiments with synthetic peptides likewise could not resolve if S6 phosphorylation indeed acts as potential *steric switch* to tune γ -secretase processing of BLK. Future work will be required to deconvolute the directly acting target of TAK285-induced BLK degradation.

Fig R11. Annotated genome-wide CRISPR/Cas9 data for the TAK285 treatment condition. **a** Mapping of all kinases onto the screening data. **b** Mapping of all chemoproteomics hits onto the data. **c** Mapping of top 10 putative BLK S6 targeting kinases.

3.4. It would be good to show the specificity of TAK285 by whole proteome profiling as was done for the other two inhibitors. Are other myristoylated proteins also affected by the compound?

In response to this comment, we performed whole proteome profiling in both the original K562 screening cell line (data not shown) as well as in NALM-6 cells (see now **Fig 3a**). Due to lineage specific expression, BLK was only detected in the latter cell line model. We identified very selective degradation (5 proteins downregulated after 18h of 2.5 μ M drug treatment. Specifically, no other kinase nor any putative myristoylated protein (NTHL1, ETV3, TBC1D31, NAGLU) were downregulated.

4. RIPK2

4.1 RIPK-IN-4 induced RIPK2 degradation is relatively clear compared to the other two follow-up stories. I wondered if other RIPK2 inhibitors like ponatinib (type II kinase inhibitor) or GSK583 (type I kinase inhibitor) also can induce RIPosome followed by turnover via autophagy. If they do not, what makes RIPK-IN-4 different from the other inhibitors, especially GSK583 that is structurally similar to RIPK-4-IN.

We assessed a full set of RIPK2 inhibitors covering published examples of both Type I and Type II RIPK2 inhibitors. The suggested small molecule Ponatinib had to be excluded from the analysis due to its autofluorescence in the BFP channel (see **Reviewer Fig R12a**). Overall, the three type I inhibitors GSK583, RIP inhibitor 2 and WEHI-345 (PMID: 36959850, 10.1016/j.chembiol.2015.07.017) did not show any foci formation nor degradation in our RIPK2 RKO iCas9 stability reporter cell line (see **Reviewer Fig R12b, c**). We could, however, observe foci formation and degradation for Regorafenib and Gefitinib, reported Type II and Type I inhibitors of RIPK2, respectively. However, contrary to RIPK-IN-4, foci appeared and were cleared at a slower rate. This highlights a general susceptibility of RIPK2 to multimerize into qualitatively distinct, RIPosome-like structures with different clearance efficiencies. Further, these data suggest that drug-induced RIPK2 multimerization and degradation are independent of the type of inhibitor (Type I vs II).

Fig R12. Assessment of the degradation capacity of a panel of type I and II RIPK2 inhibitors. **a** BFP autofluorescence of Ponatinib. **b** Temporal trajectories of normalized BFP fluorescence per cell of RIPK2 stability RKO iCas9 cells and indicated compounds (2.5 μ M) **c** Quantified foci number per cell and compound.

Orthogonal to assessing published RIPK2 inhibitors, we extended our analysis to other hits from within our drug screening data. We focused on two compounds GSK2983559 and LY2109761, for which we could identify literature evidence supporting direct RIPK2 engagement. We confirmed RIPK2 degradation as well as assembly formation, completely phenocopying RI-4 mediated RIPK2 degradation. We thus assessed TMUB1 dependency and again confirmed TMUB1's critical role in facilitating RIPosome formation and clearance (see **Reviewer Fig R13a, b**). Of note, in response to Reviewer 2, we also generated data that show that TMUB1 interacts with RIPK2 following RI-4 treatment (co-IP, see new panel **Fig 4I**). In line with these data, GSK2983559 and LY2109761 both also immunoprecipitated TMUB1 to a similar degree (see **Reviewer Fig R13c**). In contrast, the slow-acting degraders shown in (**Reviewer Fig R12b, c**) are not able to engage this interaction (see **Reviewer Fig R13c**), substantiating TMUB1's role as facilitator in this process.

Fig R13. Assessment of the degradation capacity of two additional hit compounds. **a** Temporal trajectories of normalized BFP fluorescence per cell of RIPK2 stability RKO iCas9 cells and indicated compounds (2.5 μ M) ($n = 3$). **b** Quantified foci number per cell and compound from data in (a). **c** FLAG Co-immunoprecipitation of K562 RIPK2-Nluc-3xFLAG cells across evaluated hit compounds ($n = 3$, data from identical membrane).

Can authors predict the structure of RI-4 mediated oligomerized RIPK2?

Structurally, as mentioned and shown in the original manuscript RIPK2 CARD-CARD domain stacking must be one of the driving forces for multimerization. We, however, hypothesized that the assemblies consist of a heterogeneous set of proteins akin to other inflammasomes (<https://www.nature.com/articles/cdd201733>) which stymies structural elucidation or predictions with tools we have at our disposal (cryo-ET is not accessible to us and such studies could not be completed in the timeframe allocated to this revision).

To follow the Reviewer's request and better understand RI-4-induced RIPK2 assemblies, we performed time-resolved BioID for RIPK2 (1h, 4h after RI-4 treatment and 18h after Bafil. A1 or RI-4 and Bafil A1. treatments). The results are included on p.13-14 and corresponding figures **Fig 4k**, **Extended Data Fig 9a** and **Supplementary Data Table 5**. We identified many previously established RIPosome components, particularly those associated with an antagonizing function (e.g. TNFAIP3, CYLD, N4BP1 and CCDC50). Furthermore, we found clathrin associated proteins such as EPS15 and proteins associated with ubiquitin dependent vesicle-mediated transport such as GGA3. Our top scoring interactor across all timepoints was SQSTM1/p62, a well-established autophagy receptor. This finding is well aligned with our data that knockout of the scaffolding protein FIP200 prevents RI-4-induced RIPK2 degradation. FIP200 is a SQSTM1/p62 interactor (Turco *et al.*, 2019) and essential to bridge the recognition of autophagic cargo to the autophagy initiation machinery.

GO enrichment of the drug-induced RIPK2 interactome further highlighted the importance of ubiquitination for this process. Indeed, the top three (1h, added to **Extended Data Fig 9a**) or four (4h, see **Reviewer Fig R16**) most significant GO molecular function terms (2023) were ubiquitin associated terms. For the 1h timepoint, we further identified Kinase Binding (GO:0019900) and Protein Kinase Binding (GO:0019901) among the top 10 most enriched terms.

Given the important role of ubiquitination in the process and following requests from Reviewer 2, we finally set out to identify potential E3 ligases involved in the process of drug-induced RIPosome formation and/or degradation. For this we performed targeted KO of E3s identified as interactors in the BioID and also conducted an orthogonal UBI-focused CRISPR/Cas9 screen (not included in the manuscript due to space constraints, see corresponding data in detail in **Reviewer Fig R19-20**). In sum our data identified that double

knock-out of the two IAP domain E3 ligases XIAP and cIAP1 (DKO) rescues RIPosome clearance, while RIPosome formation is not impaired (**Extended Data Fig 9b-d**). Of note, individual XIAP knock-out enables partial RIPK2 multimerization supporting its previously established role for maintaining RIPK2 in its cytosolic state (PMID: 31350258). Finally, the point mutant RIPK2^{I212D} was reconstituted as stability reporter in RKO iCas9 RIPK2 KO cells, and completely phenocopied the DKO (added as **Extended Data Fig 9e-g**). This nicely corroborates with literature evidence revealing I212 as one of the critical residues for interacting with IAP domains (PMID: 32954645) and hence outlines a critical role for the interaction with the two identified E3 ligases XIAP and cIAP1.

In sum, we have followed up on this request from Reviewer 1 by conducting time-ranging BioID experiments, pooled genetic screens and focused genetic validation experiments to describe the heterogenous composition of RI-4 induced assemblies (which is not amenable for structural elucidation outside of cryo-ET) and to identify factors relevant for their clearance.

4.2. The authors state that the CRISPR/Cas9 screen revealed an enrichment of hits involved in lysosomal degradation. The statement is missing statistics (is the enrichment significant?) and how the hit annotation was done (using GO categories, Uniprot annotations, or others?).

We apologize for the missing details. We performed GO enrichment and have updated the relevant text section to:

“Systems-level analysis of the identified effector genes revealed an enrichment of hits that are involved in lysosomal degradation (GO Biological Process 2025: endosomal vesicle fusion [GO: 0034058, p-val 0.003409]; phagolysosome assembly [GO0001845, p-val 0.003959]).”

4.3. The RIPK2 Δ CARD variant also forms foci in Fig 4H.

RIPK2 Δ CARD indeed forms a minor set of foci, which likely stems from the presence of endogenous RIPK2. In total there is still a very striking difference between WT and Δ CARD constructs supporting the notion that CARD-CARD domain stacking is required for the multimerization phenotype we observe. As stated in 4.1, not RIPK2 alone but a multitude of proteins assemble together to enable RIPK2 degradation. Indeed this type of mechanism is well established for other systems such as Tau fibril formation and subsequent clearance via macroautophagy (10.1126/sciadv.adm8449). Tau fibrils act as “seeds”, and concerted interactions with the autophagosome machinery particularly p62 enable coordinated clearance – or failure of clearance in disease settings.

4.4. The authors state that RIPK2 multimerizes or forms higher-order assemblies, but this is not shown.

To assess RIPK2 self-interactions, we added a second fluorescent reporter to our RIPK2 stability reporter. Both RIPK2-GFP and RIPK2-BFP assembled into identical clusters as highlighted in fluorescent cross-section profiles (see **Extended Data Fig 8f, g**). Noteworthy, the identified clusters containing both RIPK2 fusions are too large to be explained by a simple dimeric assembly. Moreover, we have consistently changed the wording in the manuscript to higher-order assemblies, avoiding the terminology of multimerization.

Figure 4 title states that the process is TMUB1 dependent, but TMUB1 deletion seems to just slow down the process.

We have altered the figure title to account for the fact that TMUB1 acts as a facilitator. We also extended our experiments to assess TMUB1's role in RI-4 mitigated RIPK2 degradation (as also mentioned in point 4.1). Co-immunoprecipitation of RIPK2-Nluc-3xFLAG revealed that TMUB1 interactions are unique to inhibitors (RI-4, GSK2983559 and LY2109761) that induce foci that are subsequently turned over (**Fig 4l, Extended Data Fig 9k, Reviewer Fig R12c**). Secondly, we performed IF staining for TMUB1 unveiling TMUB1-RIPK2 co-localized assemblies upon RI-4 treatment (**Fig 4m, Extended Data Fig 9l, m**). Altogether, the data we have accumulated, point to a critical role of TMUB1 in facilitating the turnover of RIPK2 assemblies.

Minor points:

1. Statistics are missing in most figures.
2. Figure legend of Fig. 2I is missing
3. On page 11 “BLK S6A strongly abrogated inhibitor-induced degradation (Fig 3J)”, it is not 3J, it is 3I.

We thank Reviewer 1 for the attention to detail in going through our manuscript. All minor points have been addressed in the revised version of the paper.

Referee #2 (Remarks to the Author):

The development of protein kinase targeted PROTACs is a “hot” field right now, and relevant to this it is known that some small molecule kinase inhibitors (KIs) not only inhibit kinase activity but also trigger degradation of their target protein kinase. Here, the authors set out to survey which KIs cause degradation of their target protein kinases, as assayed using Nanoluc PK-fusions, examining the protein levels of 98 protein kinases (88 WT and 10 mutant forms) in cells treated with 1570 monovalent kinase inhibitors. They found that 69 of these PKs exhibited a reduced protein level in response to at least one KI compound, with 232/1570 KIs causing degradation of at least one PK. There was a partial correlation between degradation and whether the PK was a strong or weak HSP90 chaperone client, but the degradation of several PKs could not be explained by chaperone deprivation, nor was there necessarily a good correlation with whether the PK had been shown to be sensitive to a designed heterobifunctional degrader.

This led the authors to follow up three examples of an unexplained KI-induced protein kinase degradation response in order to determine the mechanisms underlying KI-induced degradation: LYN by SRC inhibitor-3 (SI-3), BLK by TAK285, and RIPK2 by RIPK-IN-4 (RI-4). In the first case, SI-3 caused almost complete degradation of the LYN SFK within 2 hours, an effect that was ubiquitin dependent, being nearly fully inhibited by a combination of a proteasome inhibitor plus a lysosome inhibitor, and also demonstrated that the SI-3 effects were relatively selective for LYN compared to the other SFKs. By conducting a CRISPR screen in KBM7 cells stably expressing a LYN-Nanoluc reporter for genes required LYN degradation, they found a requirement for the CBL and CBLB RING domain E3 ligases, which have previously been implicated in SFK kinase degradation. Also, using BioID proximity labeling, they found that SI-3 induced the association of LYN with both CBL and CBLB, which were the strongest hits. Consistent with a role for CBL/CBLB E3 ligase activity, they found that combined knockout of CBL and CBLB but not either single knockout, prevented SI-3 induced LYN degradation. By assessing dose dependent changes in kinase activity using Kinobead profiling of cells treated with SI-3, they deduced that CSK rather than LYN was the primary SI-3 inhibitor target, and by making and using SI-3-resistant T266M CSK and T319I LYN gatekeeper mutants they showed that only expression of the SI-3-resistant CSK mutant prevented SI-3-induced LYN degradation, implying that loss of CSK-mediated inhibitory LYN Y508 phosphorylation was responsible for the response. Their model is that LYN kinase activation in the absence of CSK activity leads to its rapid CBL/CBLB-dependent ubiquitylation and degradation. They went on to identify pY32 in LYN as a possible phosphodegron for the CBL E3 ligase and pY316 as a second candidate CBL/CBLB phosphodegron.

In the second case, treatment with 1 μ M TAK285 caused highly selective degradation of BLK, another SFK, with a half-life of ~2 hours, in a ubiquitin and proteasome-dependent manner, and with kinetics similar to those observed upon treatment with an HSP90kinase chaperone inhibitor. A CRISPR screen in KBM7 cells identified subunits of the γ -secretase complex, including NCSTN, as being important for TAK286-induced BLK degradation, and, consistently, a proximity-labeling interaction between BLK and NCSTN was detected. However, because γ -secretase exclusively carries out intramembrane cleavage of Type 1 TMD proteins, it seemed unlikely to cleave BLK directly. Inhibition of BLK N-terminal myristoylation with IMP-1088 reduced TAK285-induced BLK degradation, implying a requirement for BLK membrane association. They went on to show that swapping just the BLK N-terminal domain with that of SRC rescued TAK285-mediated BLK degradation, implicating

this region in degradation. TAK285-induced BLK degradation was not blocked by pretreatment with the covalent BLK inhibitor acalabrutinib, suggesting that TAK285 acts independently of direct binding to the BLK ATP pocket, consistent with an inability to detect BLK binding to TAK285 using a bead-based assay. Focusing on the BLK N-terminal region they found that an S6A mutation, which decreases myristoylation, causing strongly reduced BLK membrane association, as did treatment with the IMP-10880 myristoylation inhibitor. They also showed that TAK285 treatment led to a loss of BLK myristoylation and BLK relocalization, which was prevented by a γ -secretase inhibitor. Consistent with release into the cytoplasm being causative in NLK degradation they found that both the S6A mutant BLK and BLK in IMP-1088 treated cells were reduced. They propose that TAK285, through an unknown adaptor protein, induces membrane-associated BLK to interact with γ -secretase, which then somehow releases BLK into the cytoplasm where it is degraded spontaneously in a ubiquitin-dependent manner.

Lastly, they analyzed how the RIPK-IN-4 RIPK inhibitor induces degradation of RIPK2, a protein kinase involved in NOD1/2 pattern recognition receptor signaling to inflammatory pathways. They showed that RI-4 causes slow RIPK2 degradation over 18 hours, with CSK being the only other PK stabilized by RI-4 treatment. An analogous CRISPR KBM7 cell screen in this system identified components of the lysosome as being important in RI-4-induced RIPK2 degradation, with TMUB1, a UBL domain protein implicated in quality control of TMD proteins at ER contact sites, being a top hit. Consistent with lysosomal degradation being important, bafilomycin A1 reduced RI-4-induced RIPK2 degradation, as did knockout of the FIP200 macroautophagy adaptor protein. They went on to show that RIPK2 aggregates appeared in response to RI-4 treatment, which were cleared in a manner blocked by the bafilomycin A1 autophagy inhibitor, and reduced by genetic depletion of TMUB1. They conclude that RI-4-triggered aggregation of RIPK2 results in slow autophagy dependent degradation of RIPK2, which is a novel protein kinase depletion mechanism.

The authors' kinome wide KinDeg screen to identify small molecule protein kinase inhibitors that also trigger degradation of their target protein kinases has led to some intriguing and unexpected findings, which have potential implications for use of small molecule protein kinase inhibitors in the clinic. To illustrate different mechanisms through which KIs can induce protein kinase degradation, they analyzed three KI/protein kinase pairs in depth - LYN/SI-3, BLK/TAK285 and PIPK3/RI-4. No common principles emerged from the three cases studied, which in one sense is presumably the point the authors are trying to make. However, as a result the paper is somewhat disjointed. In all three examples, there are unanswered questions. For instance, the mechanism through which TAK285 induces loss of BLK myristoylation and its degradation in a γ -secretase manner remains unclear, and the mechanisms underlying RI-4 induced aggregation and clearance of RIPK2 by autophagy-dependent lysosomal degradation are not well worked out.

We thank the reviewer for the helpful comments. While the mechanisms are very diverse, we do see a common denominator and overarching logic. In classical TPD the small molecules act as a proximity-inducing bridge between the target protein and a component of the cellular degradation machinery. In contrast, all inhibitors we have nominated for mechanistic follow-up studies induce kinase degradation by inducing endogenous, less stable protein "states". These mechanisms are of course very diverse, reflecting the many layers that control endogenous protein levels/stability and manifest, in the validated examples, via altering activity, localization and multimerization of the kinase target. Of note, our parallel study on small-molecule degraders of the enzyme IDO1 further expand that mechanistic repertoire and

extends it to another protein family. In this case, inhibitors of IDO1 mimic a state of IDO1 that is not engaged with its co-factor Heme and hence more efficiently turned over via its native E3 ligase KLHDC3 (Hennes *et al.*, biorxiv 2024). Collectively, these findings open up a new way of engineering degrader modalities, which is dependent on a detailed mechanistic understanding of the homeostasis of the target under investigation.

This reviewer is correct that, in validating and working up the findings around RIPK2, BLK and LYN, some questions pointing to the exact molecular mechanism remain open when compared to previous degrader mechanism of action studies published in *Nature*, such as Slabicki *et al* (BCL6 degradation, *Nature* 2020), Xie *et al.* and Yeo *et al.* (both on CoREST degradation). We think it would however also be fair to put our manuscript into context with these findings. In all of these published cases, stories dissected the mechanism of a single small molecule. Each of these compounds had already been identified to act as degrader before, so (while all these papers are beautiful and impactful), they assigned one mechanism to a degrader phenotype that had previously been established. In contrast, the goal of this study is to contribute to a conceptual understanding of an entire class of degrader mechanisms where we have identified >400 new degrader events that have not been reported before, could make systems-level conclusions and finally went into mechanistic detail with three independent, never reported drug-induced degrader mechanisms. Given that, we would hope that this Reviewer agrees that we cannot provide the same mechanistic depth on each of these findings compared with the aforementioned studies.

Points: A. LYN/SI-3

1. CBL/CBLB mediated ubiquitylation of LYN would be expected to primarily cause proteasome-dependent LYN degradation, raising the question why LYN degradation is partially lysosome dependent? Does CBL-mediated degradation of LYN require it to be membrane associated? CBL-induced degradation of RTKs involves their lysosomal degradation. The LYN protein is membrane associated through two different N-terminal lipid modifications - could this explain the partial dependence of the SI-3 effect on lysosomal degradation? What effect does IMP-1088 or S6A LYN have on SI-3 induced LYN degradation?

We thank Reviewer 2 for bringing up this point. As highlighted in **Extended Data Fig 5a**, CBL is the E3 that enables the plasticity for both the lysosomal and proteasomal route. This is also in line with previous evidence that CBL is able to mark proteins for lysosomal degradation (e.g. the RTK EGFR, PMID 12754251). To further address this point, we have performed additional experiments. These showed that LYN retains its degradation capacity via SI-3 also when we perturb membrane association using a G2A mutation, or the suggested S6A mutation (see **Extended Data Fig 5c**). Equally, degradation capacity is retained after IMP-1088 treatment (see **Extended Data Fig 5d**). Interestingly, LYN G2A (non-membrane associated LYN) is only degraded via the proteasomal route, underpinning that membrane association is required for lysosomal degradation (see **Extended Data Fig 5e**). We have added the accompanying data to the manuscript as above-mentioned panels and corresponding text sections (**p.7**).

2. Have the authors demonstrated that LYN becomes (poly)ubiquitylated in SI-3 treated cells, and, if so, determine Ub sites and Ub linkage types on LYN?

To address this question, we performed HA-Ubiquitin overexpression experiments followed by FLAG-immunoprecipitation of HEK293T LYN-Nluc-3xFLAG cell lines. First, we identified increased total ubiquitination of LYN after drug treatment (added as **Extended Data Fig 5b**). Secondly, K48R and K63R ubiquitin mutants both retained this signature, yielding increased HA-signal after SI-3 treatment. The presence of mixed ubiquitin chains is in line with the plasticity to be able to engage both the lysosomal and proteasomal system.

3. Figure S3M: Why didn't the authors assay LYN pY397 levels in SI-3 treated cells? If their model that SI-3 inhibition of CSK results in LYN activation is right, then pY397 levels should increase (unless SI-3 prevents LYN autophosphorylation at Y397). A LYN Y397F mutant could be used as a control that should show reduced degradation according to their model.

To address this point, we extended our data and assessed both Y397A and Y508A mutant degradation in our stability reporter setup. Moreover, we assessed LYN pY397 levels akin to how we have determined pY508 levels in the original version of the manuscript.

While the Y397A mutant reporter was still degraded by SI-3, Y508A was fully resistant to SI-3 mediated degradation (added as **Extended Data Fig 5j**). pY397 levels remained unchanged (updated **Extended Data Fig 5h, i**). Prior evidence in disease contexts highlights that LYN Y508F mutations are sufficient for alleviating the inhibitory state of the kinase (<https://doi.org/10.1038/s41467-023-36941-y>), we thus suspect that the steric release is the critical limiting step for enabling the phosphorylation of the two identified phosphodegrons. Furthermore, both SRC or remaining endogenous LYN^{wt} may be able to act as critical drivers of the identified enhanced phosphorylation on Y316.

4. Based on a prior report, the authors propose that pY32 serves as a phosphodegron for CBL in response to SI-3, but did not demonstrate directly that SI-3 induced CBL-mediated LYN degradation requires its "SH2" domain and its ability to bind to pTyr - this could be tested with appropriate CBL mutants. With regard to the proposal that pY32 acts as a CBL-targeting phosphodegron in LYN, it would be reassuring if they showed that CBL/CBLB can bind a LYN pY32 peptide (as well a pY316 peptide) - this was not done in the cited study (ref. 33). The sequence dependence of the CBL "SH2" domain pTyr binding has been examined in several studies, with (N/D)XpY(S/T)XXP and RA(V/I)XNQP(S/T) being reported as consensus. Also, Park's group showed that DpYR is needed for CBL-dependent pY1003 MET RTK degradation. Neither the pY32 nor the pY316 site conforms well to these consensus sequence and this point deserves discussion. Moreover, In the crystal structure Y316 is in a beta strand on the surface of the LYN catalytic domain and would not be very accessible to phosphorylation or CBL "SH2" domain binding (but perhaps Y316 phosphorylation would affect the local structure).

Overall, we believe that the functional dependence of LYN degradation on Y32 and Y316, both of which are conserved tyrosines and known phosphorylation sites, supports their function as putative phosphodegrons. We also assessed Y316 positioning within LYN's predicted AlphaFold structure to validate surface exposure. In line with our hypothesis, pY316 would be exposed to the surface and thus in theory accessible to E3 recognition.

Future work will indeed be required, and we thank the reviewer for the suggestion of these excellent experiments to address direct CBL binding. Given space constraints we face in this study (we already show a total of 129 display panels), and given the feedback by Reviewer 1

to generally de-prioritize our efforts on further elucidating the MoA of LYN degradation, we have decided not to pursue further biochemical characterization.

5. What tyrosine kinases phosphorylate Y32 and Y316 in LYN when it is activated by SI-3? Scholes et al. (ref 33) proposed that Y32 is phosphorylated by an activated SFK when CSK is inhibited, but did not show this experimentally. PhosphoSitePlus indicates that Y32 is an EGFR target, but the new Kinase Library tool in PhosphoSitePlus predicts BRK, an SFK relative, as far and away the top hit for Y32 (BRK is also negatively regulated by C-terminal Tyr phosphorylation although the tyrosine kinase responsible seems to be unknown). pY316 has been frequently detected in HTP phosphoproteomic screens, but an upstream kinase has not been reported. Intriguingly, the KL tool scores CSK as the top hit for Y316, which means the level of LYN pY316 should decrease if SI-3 inhibits CSK (did they show through an in vitro assay that SI-3 inhibits CSK kinase activity?).

This is indeed an interesting point raised by Reviewer 2. Given the considerations raised in (4.) above, further elucidating all nuances of SI-3-induced LYN degradation would, however, exceed the scope of this manuscript.

Of note, CSK inhibition was reported in the original publication of the compound (IC50 4 nM, 10.1021/acsmchem-lett.9b00354) and we could confirm CSK engagement via NanoBret assay.

6. Figure 2I: A description of panel I is missing from the Figure 3 legend. We apologize for this oversight and have added the missing legend.

B. BLK/TAK285:

1. They propose that TAK285, through an unknown adaptor protein, induces membrane-associated blk to interact with γ -secretase, which then somehow releases BLK into the cytoplasm where it is degraded spontaneously. The main unresolved issue is how TAK285 induces BLK association with γ -secretase - is the myristoyl group directly involved in some way other than as a membrane anchor? To test whether plasma membrane association of BLK per se is needed, BLK could be inducibly anchored to the plasma membrane or possibly a C-terminal CAXX box could be added. In addition, if released soluble BLK is constitutively degraded by the UPS, why is membrane-associated BLK protected?

To address this important point raised by Reviewer 2, we have invested substantial work in deciphering the mechanism further. A key evidence was a recent publication highlighted the ability of γ -secretase to be able to cleave lipid anchors from small molecules fused to membrane tether modifications including palmitic, stearic, oleic and myristic acid (<https://doi.org/10.1021/acsembio.4c00432>). This highlighted that the substrate/target scope of γ -secretase is wider than initially anticipated and covers non-proteinaceous substrates beyond transmembrane helices.

Based on these novel insights, we postulated that TAK285 induces γ -secretase processing of BLK via cleavage of a peptide bond, which includes the N-terminal myristic acid modification (see **Fig 3d**) and thus frees a cryptic degron (see **Reviewer Fig R8**, p.18).

To determine sequence requirements and more detailed biological insights, we have generated multiple genetic fusion, domain swap and individual point mutations including a comprehensive assessment of sequence requirements via a saturated mutagenesis screen.

First, generation of a “neutral” membrane anchor, as suggested, was performed (see **Reviewer Fig R14**). To maintain the N-terminal modification, we opted for a peptide sequence, which is again dependent on myristoylation (adapted from <https://doi.org/10.1038/s44320-025-00109-1>) rather than the suggested CAAX anchor. This “neutral” MYR-BLK variant retains the degradation phenotype via IMP-1088 supporting intact membrane anchorage, but importantly, is resistant to TAK285-mediated degradation. This suggests that a degree of sequence specificity surrounding the myristic acid is required to enable TAK285-mediated, γ -secretase dependent degradation. Identical evidence can be taken from the data reported in the index version of this manuscript, showing that a fusion protein that engrafted the unique domain of SRC on BLK was not degraded by TAK285 despite being membrane anchored (**Fig 3e**, **Extended Data Fig 7d**). Due to space constraints, we have not included this genetic construct in the manuscript.

Fig R14. BLK anchored via a “neutral” myristic acid is not sensitive to TAK285-mediated degradation.

Next, and following the suggestion of comment B.3, we reversed our domain swap experiments and could install TAK-285 and γ -secretase-dependent degradation onto SRC by fusing BLK’s unique domain to SRC (**Fig 3e**). Moreover, the unique domain of BLK itself was sufficient to phenocopy this effect when fusing it to BFP, highlighting the critical importance of these first 57 amino acids and the independence of our mechanism of action to BLK’s kinase domain (**Fig 3g**). To deconvolute sequence requirements in more detail, we thus opted for a deep mutational scanning (DMS) screen. In brief, each amino acid position was mutated to every other possible amino acid. Coupled to our stability reporter, we performed a FACS-based DMS screen akin to our FACS-based CRISPR/Cas9 screen setup. This allowed us to comprehensively assess all amino acid substitutions that would disrupt TAK285-induced BLK degradation (**Fig 3f**). Orthogonally, we assessed IMP-0188 mediated degradation in the identical setup (**Extended Data Fig 7g**).

We noted several positions that either impacted both TAK285 and IMP-1088 treatment (G2) or specifically impaired TAK285 degradation (L3, V4 and S6). Starting with G2, we tested the two top scoring mutations G2I and G2L (**Fig 3f**, **Extended Data Fig 7g**). Both mutations render BLK resistant to TAK285 as well as IMP-1088-mediated degradation (**Extended Data Fig 7h**). Given G2’s critical role for myristoylation (**Reviewer Fig R8**) as expected imaging revealed that G2 mutants are cytosolic (**Extended Data Fig 7k**) and not myristoylated (**Extended Data Fig 7i, j**). Mutating the second glycine residue not only ablates the myristic acid modification (**Reviewer Fig R8**) but also generates a novel, and as shown e.g. for SRC,

more stable N-terminus. Collectively, this supports that cleavage of myristic acid by γ -secretase releases a novel N-terminus, such as the terminal glycine, which is then recognized by yet to be determined E3 ligase(s).

Next focusing on TAK285-specific residues, we identified L3, V4 and S6 as key positions. As already highlighted in the initial manuscript submission, S6 is critical for myristoylation status. Corroborating our initial findings of this variant's cytosolic localization (moved to **Extended Data Fig 7d**) as well as decreased baseline stability (moved to **Extended Data Fig 7f**), we measured the myristoylation status of S6A. Indeed, compared to wt BLK, S6A has strongly impaired myristoylation (added as **Extended Data Fig 7i, j**). Critically, we extended our truncation analysis further and showed that residues #1-7 are sufficient to retain TAK285-mediated BLK degradation (see **Fig 3g**).

Finally, for the two additional key positions identified in our DMS dataset, L3 and V4, we found multiple mutants that are (a) resistant to TAK285-mediated degradation (**Extended Data Fig 7h**), while (b) retaining sensitivity to IMP-1088 treatment. Further, the assessed mutants are (c) myristoylated to at least the same degree as BLK (**Extended Data Fig 7i, j**) and in line with this data (d) localized to the membrane (**Extended Data Fig 7k**). When compared to SRC's sequence we noted that L3 and V4 form an aliphatic patch in BLK (**Reviewer Fig R9**), which we hypothesized may be critical for γ -secretase processing. We hence mutated the corresponding residues in SRC to those of BLK. Indeed, we were thus able to generate a SRC variant that was degraded by TAK285 in a γ -secretase dependent manner (see **Fig 3h**, **Extended Data Fig 7l**).

Our initial BioID data already pointed towards an intrinsic affinity between BLK and γ -secretase (**Extended Data Fig 6i**). We now further substantiate evidence for this interaction *in vitro* with data from an MST experiment with recombinant proteins (see **Extended Data Fig 6j**). This motivated the development of a structural model via Alphafold3 (see **Fig 3i**). After minimal relaxation of the system via MD simulation, we found that within the resulting complex, BLK's unique domain is positioned within the active site of γ -secretase. Specifically, the peptide bond between the myristic acid and G2 is coordinated by two hydrogen bonds formed by the two catalytic residues D385 and D275 of γ -secretase. Furthermore, we noted that L3 is positioned within a neighboring hydrophobic pocket, corroborating our mutational data and this residue's important role to enable γ -secretase processing upon TAK285 treatment. We also attempted structural elucidation but could only observe heterogenous complex formation, which disabled the generation of a high-resolution structure.

Given the structural prediction, the extensive mutagenesis and truncation data as well as new literature evidence highlighting that γ -secretase can cleave myristic acid from small molecules (<https://doi.org/10.1021/acscchembio.4c00432>), our mechanistic hypothesis thus places TAK285 as small molecule that induces the γ -secretase dependent cleavage of myristic acid from BLK, releasing it into the cytosol where the generated neo N-terminus serves as an N-degron (see **Reviewer Fig R8**).

Since the peptide bond of the myristic acid to G2 is specifically coordinated by the two active site aspartic acids, we hypothesized that this is the site for initial cleavage by γ -secretase. As shown in the original submission, myristic acid is lost from BLK (**Fig 3i**) upon TAK285 treatment and in a γ -secretase dependent manner, which supports this notion. Given the positioning of BLK on the cytosolic side of the secretase complex (rather than being a transmembrane protein), as well as the per-residue decomposition of binding free energy (see

Reviewer Fig R10), in which loss of the myristic acid drastically increases the energetic contribution of G2, it is most likely that cleavage results in the release of BLK with a remnant glycine into the cytosol. Despite our best efforts, we were, however, not able to identify the precise cleavage product via MS.

Finally, we set out to identify the kinase mediating BLK degradation. Previous competition experiments (moved to **Extended Data Fig 6k**) already informed that direct BLK binding is not important. To exclude involvement of direct binding at the effector (γ -secretase), we conducted MST experiments, where TAK285 was dose-dependently titrated against purified γ -secretase. However, no binding was detected. Collectively, these data are in line with a network-mediated effect of TAK285. To elucidate this, we performed multiple attempts to map potential targets using chemoproteomics. In addition to the Kinobead profiling data from the index submission, we now have synthesized a tethered analog of TAK285 for immobilization to conduct direct affinity enrichment followed by proteomics analysis. While this analog retained BLK degradation activity (data not shown), in-depth analysis failed to reveal targets beyond MAP2K5 which we had already identified via Kinobeads profiling. We hence set out to validate MAP2K5, but genetic KO did not alter TAK285 degradation (added as **Extended Data Fig 6n**).

Given that TAK285 is a kinase inhibitor, we would suspect phosphorylation to be the most likely PTM to occur and hence enable γ -secretase processing of BLK. The mutational data would suggest S6 as one potential site. We hence re-visited our genome-wide CRISPR/Cas9 screening data, mapping all kinases, all concatenated chemo-proteomic hits, all as well as the top 10 kinases predicted to phosphorylate S6 within BLK (see **Reviewer Fig R11**). However, no noteworthy hits could be identified. MS identification as well as experiments with synthetic peptides likewise could not resolve if S6 phosphorylation indeed acts as potential *steric switch* to tune γ -secretase processing of BLK. Future work will be required to deconvolute the directly acting target of TAK285-induced BLK degradation.

2. As shown using the TAMRA assay, the loss of the myristoyl (Myr) group from BLK is triggered by TAK285 treatment, but it is unclear how the Myr group is lost, or whether it is just the Myr group that is lost. Also, is it possible that γ -secretase hydrolyzes the Myr-Gly bond (the Myr-Gly linkage is an amide linkage like the peptide bond) or cleaves another peptide bond near the N-terminus, which might in turn expose a cryptic N-end rule degron?

We thank this Reviewer for this mechanistic hypothesis. As eluded to above, we have collected evidence that indeed γ -secretase hydrolyses the Myr-Gly bond, and thus releases BLK into the cytosol with a now exposed N-degron.

3. Since other SFKs are myristoylated but are unaffected by TAK285, this suggests that the specificity of BLK degradation is either inherent in the unique N-terminal sequence of BLK or in the interaction of the hypothesized adaptor protein with BLK. The authors replaced the BLK N-terminal sequence with that of SRC and showed that this prevented BLK degradation, but did not do the opposite experiment of swapping the BLK N-terminal sequence into SRC to determine whether this is sufficient to induce SRC degradation (or degradation of another irrelevant protein).

We thank the reviewer for this excellent suggestion. As shown in above answer B.1, we can indeed install degradation capacity onto SRC by inverting the domain swap experiment

(Fig 3e). Likewise, also a fusion of the N-terminal unique domain with BFP renders this protein TAK-285 degradable (Fig 3g). Moreover, it is even sufficient to only mutate residues 3 and 4 of SRC to generate TAK285-dependent degradation mechanism (Fig 3h, Extended Data Fig 7I). This highlights critically the importance of residues #3 and #4 to enable favorable conditions for γ -secretase access.

4. Figure 3J: As a minor point, the total level of BLK protein was increased in lane 4 with a proteasome inhibitor + TAK285 combination but not seen in lane 2 with the same combination – this deserves some comment,

It is unclear why we get slightly increased protein amounts for this treatment condition. We did, however, not see a consistent pattern for this specific increase across replicates (see Reviewer Fig R15). Overall, it, however, favors our hypothesized mechanism since the critical factor to evaluate in this setting is the ratio of TAMRA to FLAG. There we clearly see a marked reduction of myristoylation upon TAK285 treatment and Carfilzomib pre-treatment (corresponding lanes 3&4 in Reviewer Fig R15a and lanes 5&6 in Reviewer Fig R15b).

Fig R15. Replicates 2 and 3 corresponding to original Fig 3j, now 3h.

C. RPIK2/RI-4:

1. It is unclear is how RI-4 induces RIPK2 aggregate formation, or how these aggregates are recognized by the autophagy machinery. Do the authors have any mechanistic insights? Are other proteins involved in aggregate formation?

We appreciate this comment from Reviewer 2, which has also been raised by Reviewer 1. To address this concern and to establish the interactome of RIPK2 after RI-4 drug treatment we performed temporal BioID (1h, 4h after DMSO or RI-4 treatment and 18h after Bafil. A1 or RI-4 and Bafil. A1 treatment). The corresponding results are included on p.13-14 and in Fig 4k, as well as in Extended Data Fig 9a. In sum, we identified many previously established RPOosome components, particularly those associated with an antagonizing function (e.g. TNFAIP3, CYLD, N4BP1 and CCDC50). Furthermore, we found clathrin associated proteins such as EPS15 and proteins associated with ubiquitin dependent vesicle-mediated transport such as GGA3. Our top scoring interactor across all timepoints was SQSTM1/p62, a well-established autophagy receptor. This finding is well aligned with our original data that knockout of the scaffolding protein FIP200 prevents RI-4-induced RIPK2 degradation (Fig 4f, g). FIP200 is a SQSTM1/p62 interactor (Turco *et al.*, 2019) and essential to bridge the recognition of autophagic cargo to the autophagy initiation machinery.

GO enrichment of the drug-induced RIPK2 interactome further highlighted the importance of ubiquitination for this process. The top three (1h, Extended Data Fig 9a) or four (4h, Reviewer Fig R16) most significant GO molecular function terms (2023) were ubiquitin associated terms. For the 1h timepoint (Extended Data Fig 9a), we further identified Kinase

Binding (GO:0019900) and Protein Kinase Binding (GO:0019901) among the top 10 most enriched terms.

Fig R16. GO enrichment for Molecular Function terms (2023) for scored BioID interactors in the 4h RI-4 treatment, matching data in **Extended Data Fig 9a)**

In this regard, the authors have implicated TMUB1 in the process, but it is unclear whether it interacts with RIPK2, perhaps in a ubiquitin-dependent manner.

We performed several experiments to assess RIPK2 and TMUB1's interaction. First, we performed co-immunoprecipitation of RIPK2-Nluc-3xFLAG highlighting significant RI-4 induced TMUB1 interaction (**Fig 4l, Extended Data Fig 9k**). Secondly, we performed IF staining for TMUB1, which showed TMUB1-RIPK2 co-localized assemblies upon RI-4 treatment (**Fig 4m, Extended Data Fig 9l, m**). This crucially supports our genetic evidence that TMUB1 is involved in orchestrating the formation of RIPosome structures to initiate highly coordinated clearance of these RIPK2 assemblies.

To understand if the interaction between RIPK2 and TMUB1 is ubiquitin dependent, we pre-treated cells with the UBA1 inhibitor TAK243 to globally block ubiquitination, followed by treatment with RI-4 and immunoprecipitation. This experiment (**Reviewer Fig R17**) highlights that the ligand-induced interaction between RIPK2 and TMUB1 is independent of ubiquitination. We cannot exclude that the UBL domain of TMUB1 is required in interacting with other components of the RIPosome-like structures that are induced by RI-4. Experimental efforts to reconstitute TMUB1 knockouts with UBL-deficient cDNAs (Δ UBL TMUB1) were inconclusive since the correct localization of TMUB1 was impaired (data now shown).

Fig R17. Co-immunoprecipitation of RIPK2-Nluc-3xFLAG cells pre-treated (pt.) for 1h with DMSO or TAK243 (1 μ M) followed by DMSO or RI-4 (2.5 μ M) treatment (tr.) for 2h.

2. Does autophagy-dependent degradation of RIPK2 require its ubiquitylation? Was the effect of the TAK243 Ub E1 inhibitor on RIPK2 degradation tested in this system?

We tested TAK243 dependency of RI-4 mediated RIPK2 degradation. We could establish ubiquitin dependency of this process (see **Reviewer Fig R18** for representative images; the quantified data has been added to the manuscript in **Extended Data Fig 8l, m**). We further noted that TAK243 did not prevent drug-induced multimer formation. As shown below, at later timepoints, TAK243 alone triggered minor foci formation, which could be explained by the fact that M1-type ubiquitination is required to retain RIPK2 in the cytosol (PMID: 31350258).

Fig R18. TAK243 pre-treatment impairs RI-4 induced Riposome clearance (scale bar 50 μm).

Given the importance of ubiquitination in RIPK2 assembly clearance, we further set out to identify the E3 ligases involved in the process. In parallel to our temporal BioID we set up an additional Ubiquitin-library focused CRISPR screen (see **Reviewer Fig R19**). Integrating both datasets identified multiple IAP containing E3 ligases: BIRC2/6/8 and XIAP as well as RLIM and TRAF2 (CRISPR screen) and UBR4/5 (BioID). Given previously reported data establishing XIAP as one gatekeeper of RIPosomes, we performed single knockout of each hit as well as double knockout (DKO, knockout of each hit on top of XIAP knockout). Only DKO of BIRC2 (cIAP1) and XIAP was able to abrogate degradation (see **Extended Data Fig 9b-d**), while all other combinations showed, at best, only a minor disruption of the degradation phenotype (see **Reviewer Fig R19** and **R20**)

Fig R19. Ubiquitin focused CRISPR/Cas9 screen for RIPK2 stability reporter upon RI-4 mediated degradation. **a** Enriched genes for BFP high (HIGH) and BFP low (LOW) sorted cells compared against the mid fraction (MID). **b-i** Quantified imaging data for indicated KO in RIPK2 stability reporter RKO iCas9 cells (n = 3). RI-4 was treated at 2.5 μ M.

Fig R20. Summarized BioID data and de-validation of UBR4/5 as involved E3 ligases. **a** Significantly changed proximal proteins across the three BioID conditions with scored interactors highlighted per condition. **b-c** Quantified imaging data for indicated KO in RIPK2 stability reporter RKO iCas9 cells (n = 3). RI-4 was treated at 2.5 μ M.

3. Did the authors make an RI-4-resistant RIPK2 mutant to show that the effects of RI-4 are due to inhibition of RIPK2 kinase inhibition. Likewise, is a kinase-dead RIPK2 mutant degraded upon RI-4 treatment? This information could be relevant to how RIPK2 phosphorylation is required for aggregate formation, which might be relevant to recognition by a phosphodependent autophagy receptor protein, such as a phosphodependent LIR domain receptor.

This is another very helpful point raised by Reviewer 2 that we addressed with additional experiments. To avoid confounding factors of endogenous RIPK2^{wt}, we first knocked out RIPK2 in RKO iCas9 cells and reconstituted RIPK2 stability reporters with either RIPK2^{wt}, kinase dead (R74A and D146N) or IAP-domain containing (e.g. XIAP and cIAP) E3 binding impaired (I212D) variants. Unfortunately, both kinase-dead mutants were expressed at much lower levels compared to wt, which limits the conclusion that can be drawn from this data. In general, the kinase-dead mutants still formed RIPK2 foci upon RI-4 treatment albeit at lower numbers per cell and degradation was minorly impaired (**Reviewer Fig R21**). Despite the caveat of lower expression levels, these data are in line with previous evidence that kinase function of RIPK2 is dispensable for both RIPosome formation and clearance (PMID: 30026309).

Interestingly, the E3 binding impaired variant I212D completely abrogated RI-4 induced degradation as well as multimer formation (added as **Extended Data Fig 9e-g**) corroborating the findings for the DKO of XIAP and BIRC2/cIAP1. As noted above I212 has previously been established as a key site for IAP domain engagement by IAP-domain containing E3 ligases such as the identified ligases XIAP and BIRC2/cIAP1 (PMID: 32954645).

Fig R21. Kinase-dead RIPK2 mutant RI-4 mitigated degradation. **a** Representative microscopy images (scale bar 10 μm). **b-c** Quantified microscopy data for R74A and D146N RIPK2 stability reporter cell lines highlighted in (a).

4. If autophagy is required for RIPK2 degradation, as the authors conclude., one would have expected autophagy genes, such as FIP200, to be identified in the CRISPR screen, but this does not appear to have been the case. Some discussion is called for.

Given the complexity of the CRISPR screens, we generally chose 72h post-dox induction for our assay window. This critically enables us to assess also highly essential genes such as the core essential proteasome subunits (e.g. PSMB5). In single KO experiments for FIP200, we noted that efficient KO only occurs at later timepoints, specifically using 96h or 120h of dox induction followed by compound treatment. Hence, it was likely missed at the initial screening timepoint. As shown for individual KOs (from original manuscript, Fig 4f, g), RI-4 mediated degradation is impaired upon genetic FIP200 depletion. As a second line of evidence for macroautophagy, we additionally perturbed degradation using the well-established small molecule Vps34-IN-1 (see Reviewer Fig R22). These data are well aligned with the aforementioned BioID studies that revealed that the autophagy receptor SQSTM1/p62 is a strongly enriched RIPK2 interactor after RI-4 treatment.

Fig R22. VPS34-IN-1 perturbed RI-4 mediated RIPK2 degradation.

5. If CSK degradation is induced by RI-4, as the authors state, then based on the results they obtained in the LYN/SI-3 system, LYN should also have been destabilized – was that the case?

We indeed see CSK degradation, however, inhibition of CSK would lead to a more complete phenotype since remaining CSK levels after RI-4 treatment would likely suffice to sustain LYN inhibition. We hence would not have expected LYN to be downregulated in this setting.

Other points: 1. Line 105: The authors discuss whether the 68 kinases might be correlated with them being clients for the HSP90 chaperone, but do not mention whether they are clients for the kinase-dedicated chaperone CDC37.

As shown in Taipale et al. CDC37 and HSP90 interaction scores are highly correlated, and specific to kinases. We have included a clarifying statement to the manuscript (p.4).

2. Line 182: Do the ABL1 degraders degrade ABL2 and also BCR-ABL? Does the asciminib allosteric ABL inhibitor also induce ABL degradation?

We thank the reviewer for this interesting question. In itself, we did not assess many potent ABL1 inhibitors, due to their effect on BCR-ABL and hence toxicity in our screening cell line K562. Nevertheless, we identified compounds that modulate ABL1 abundance in our screen. Some of these hits, such as the compound DPH, are also annotated as binders of the allosteric myristic acid pocket and lead to kinase stabilization rather than downregulation. Based on these data and the Reviewer's suggestion, we tested the suggested clinically relevant BCR-ABL inhibitor asciminib as well as the tool molecule GNF-2. We identified that, akin to DPH, asciminib and GNF-2 induce ABL1 stability in a dose-dependent manner. Further highlighting on-target specificity, drug-induced stabilization was impaired upon introduction of C464W, a known resistance conferring mutation within ABL1's allosteric, myristoyl binding pocket (PMID: 28819281). We have added this data (**Extended Data Fig 3**) together with the full analysis of kinase stabilizers to the manuscript (**Supplementary Data Table 1**). In total, we identified 204 stabilization events, totaling 64 kinases and 128 different compounds.

3. TAK243 was used as a Ub E1 inhibitor to block protein ubiquitylation, but the target for TAK243 does not seem to have been specified in the text.

We have added the respective information on p.6.

Minor point: 1. Line 91: The actual number of small molecule kinase inhibitors in clinical use worldwide is actually significantly higher than the number of 80 given in Roskowski's list of FDA-approved KIs, since several additional KIs have been approved by other regulatory agencies around the world (see <https://www.guidetopharmacology.org/GRAC/LigandListForward?type=Approved&database=all>.)

We thank the reviewer for this point; indeed many more kinase inhibitors are in clinical use highlighting the importance of kinase inhibitors and the need to understand potential beneficial degradation capacities of these small molecules. In order to have a citable reference, we focused on FDA-approved kinase inhibitors.

Referee #3 (Remarks to the Author):

The manuscript from Winter and colleagues describes a systematic screen for small molecules that cause kinase destabilization. Using K562 cell lines engineered to express 1 of nearly 100 kinase-luciferase fusions, approximately 1500 compounds were screened at multiple time points in each cell line for decreased luminescence, amassing an impressive dataset of compound-induced destabilization events. The power of this dataset is in the systematic approach the authors took and their well-thought controls, which made it possible to identify unique ligand-destabilization relationships with a high probability of reflecting true-positive pharmacological events of biological interest. They uncover a large number of HSP90 chaperone-dependent events, recapitulating this well-known pharmacology at a possibly larger scale. They also identify 3 events for closer study, SI-3-induced LYN destabilization, TAK285 and BLK, and RIP-IN-4 and RIPK2. SI-3 was found to inhibit CSK, which activates LYN, which activates a natural mechanism of LYN degradation that is dependent on CBL. BLK degradation was found to be caused by the intrinsic instability of BLK in the cytoplasm, wherein TAK285 caused decreased myristoylation and decreased membrane localization of BLK through an unknown mechanism involving gamma secretase. RIPK2 degradation was found to be caused by autophagy of compound-induced assemblies resembling RIPosomes that are caused by RIPK2 activation, which are also degraded by autophagy (with contributions from TMUB1 to foci formation). This Reviewer has minimal comments about the data, which are impressive in both scale and rigor. Editorially, the decision on this manuscript will depend on the level of novelty assigned to the three follow-up studies in comparison to the large body of work already documenting various compound-induced destabilization events for kinases (ref. 6 in the manuscript).

We want to thank Reviewer 3 for the positive remarks on the quality and scope of our data. Before going into detail with regards to the improvements we have made to the mechanistic understanding of the individual inhibitor-induced kinase degradation events, we would like to argue that there is also significant novelty contributed by the scale and comprehensive analysis of our data.

Reviewer 3 is of course correct in pointing out that inhibitor-induced kinase degradation events have previously been episodically reported as summarized in Jones, Cell Chemical Biology 2018. Overall, this review cites a total of 22 inhibitor-induced kinase degradation events across 11 kinases. In comparison, our study reports a total of 404 kinase degradation events covering 66 different kinases with 160 selective hits (49 kinases). This huge increase allowed us, for the first time, to establish that inhibitor-induced degradation via direct ligand engagement as well as via network effects is a frequent event. Likewise, our work establishes that there is a rich portfolio of mechanisms underpinning inhibitor-induced kinase degradation beyond chaperone deprivation. In particular, since we took on the challenge of validating several of the newly identified mechanisms, we could develop a common denominator and overarching logic connecting all tested examples. In contrast to conventional targeted protein degradation where the small molecules act as a proximity-inducers between target and ligase, all inhibitors we have tested induce kinase degradation by inducing general, less stable protein “states”. Since endogenous protein stability can be controlled via many different layers, it is hence not surprising that also the assayed inhibitors tune kinase stability via different effects including altering activity, localization and multimerization of the kinase target. Of note, our parallel study on small-molecule degraders of the enzyme IDO1 further expand that mechanistic repertoire. In this case, inhibitors of IDO1 mimic a state of IDO1 that is not engaged with its co-factor Heme and hence more efficiently turned over via its native E3 ligase KLHDC3

(Hennes *et al.*, biorxiv 2024). Collectively, these findings open up a new way of thinking about engineering degrader modalities, enhancing our understanding of endogenously regulated protein stability being one key bottleneck.

Reviewer 3 is also correct that, in validating the findings around RIPK2, BLK and LYN, some questions pointing to the exact molecular mechanism remain open when compared to previous degrader mechanism of action studies published in *Nature*, such as Slabicki *et al.* (BCL6 degradation, 2020), Xie *et al.* and Yeo *et al.* (2025, both on CoREST degradation). We think it would however also be fair to put our manuscript into context with these findings. In all of these published cases, stories dissected the mechanism of a single small molecule. Each of these compounds has also already been identified to act as degrader before, so (while all these papers are beautiful and impactful), they assigned one precise mechanism to a degrader phenotype that has previously been established.

In contrast, the goal of this study is to contribute to a conceptual understanding of an entire class of degrader mechanisms where we have identified >400 new degrader events that have not been reported before, could make systems-level conclusions and finally went into mechanistic detail with three independent (again: never reported) drug-induced degrader mechanisms. Given that, I would hope that this Reviewer agrees that we cannot provide the same mechanistic depth on each of these findings compared with the aforementioned studies.

In this Reviewer's reading of the authors' text, the CSK-dependent degradation of LYN, which is the most developed finding of the manuscript seems to have been a known mechanism, whereas the latter two findings seem more novel but less developed.

Indeed, the novelty of our findings in this section of the manuscript lies primarily in the discovery and mechanistic work up of the compound SRC inhibitor 3 (SI-3), rather than in the biological mechanism itself. A key requirement for mimicking the intrinsic LYN degradation described earlier, mediated through a CSK-regulated activity-stability switch, is the availability of a small molecule that is sufficiently selective for CSK over LYN. While numerous CSK inhibitors exist, SI-3 stands out as a compound that uniquely meets this criterion, demonstrating a clear preference for inhibiting CSK over LYN in cellular settings. This makes SI-3 the first fully chemical solution capable of engaging LYN's intrinsic degradation pathway via the activity-stability switch. SI-3-induced LYN degradation represents the first known example of a monovalent degrader molecule operating via two distinct degradation pathways. We agree with Reviewer 3 that follow-up studies on BLK and RIPK2 where less developed in the index submission of this manuscript. Therefore, we have put significant emphasis and effort on further investigating these mechanisms in the revised version of this manuscript (see also responses to Reviewers 1 and 2).

The BLK findings are intriguing and could potentially be revealing of an unrecognized function of gamma secretase, although the mechanism by which gamma secretase contributes is not settled by these experiments. How is gamma secretase impacting BLK membrane localization? What is the direct target of TAK285 leading to degradation? The authors suggest this for future work.

We thank Reviewer 3 from bringing this up. We were likewise stunned by this mechanism of action. To address this point, we have performed multiple experiments that are outlined below and have been incorporated into the revised version of the manuscript.

Our interpretation of our initial as well as the newly generated data benefitted significantly from a recent publication (<https://doi.org/10.1021/acscchembio.4c00432>) that unraveled that γ -secretase can process lipid modifications specifically shown for a group of compounds that are tethered to the membrane via e.g. an myristic acid. Given that we had already shown cleavage of myristic acid from BLK in a γ -secretase dependent manner (see **Fig 3i**), we thus hypothesized that γ -secretase might acts on BLK directly, cleaving off at least the myristic anchor and thus exposing an N-terminus that is not anchored in the membrane and instable (see **Reviewer Fig R8** (p.16) for proposed scheme).

To determine sequence requirements and more detailed biological insights, we generated multiple genetic fusion, domain swap and individual point mutations including a comprehensive assessment of sequence requirements via a saturated mutagenesis screen.

First, we could show that fusion of BLK's unique domain to SRC enables TAK285-mediated, γ -secretase dependent SRC degradation (added in **Fig 3e**). Moreover, when "engrafted" on BFP, the unique domain itself is sufficient to generate a BFP fusion that is degraded by TAK285 again in a γ -secretase dependent mechanism (see **Fig 3g**). To decode the contributions of each of these first 57 amino acids to TAK285 mediated degradation, we hence performed a saturated mutagenesis screen in which each individual position is mutated to all other possible amino acids. We coupled each variant to our stability reporter system and akin to our CRISPR screen setup sorted for the cellular populations with "high" or "low" BFP levels after vehicle, TAK285 or IMP-1088 treatment (added as **Fig 3f, Extended Data Fig 7g**).

We noted several positions that either impacted both TAK285 and IMP-1088 treatment (G2) or specifically impaired TAK285 degradation (L3, V4 and S6). Starting with G2, we tested the two top scoring mutations G2I and G2L (see **Fig 3f, Extended Data Fig 7g**). Both mutations render BLK resistant to TAK285 as well as IMP-1088-mediated degradation (**Extended Data Fig 7h**). Given G2's critical role for myristoylation (see **Reviewer Fig R8**), as expected, imaging revealed that G2 mutants are cytosolic (**Extended Data Fig 7k**) and not myristoylated (**Extended Data Fig 7i, j**). Mutating the second glycine residue not only ablates the myristic acid modification (see **Reviewer Fig R8**) but also generates a novel, and as shown e.g. for SRC, more stable N-terminus. Collectively, this supports that cleavage of myristic acid by γ -secretase releases a novel N-terminus, such as the terminal glycine, which is then recognized by yet to be determined E3 ligase(s).

Next focusing on TAK285-specific residues, we identified L3, V4 and S6 as key positions. As already highlighted in the initial manuscript submission, S6 is critical for myristoylation status. Corroborating our initial findings of this variant's cytosolic localization (moved to **Extended Data Fig 7d**) as well as decreased baseline stability (moved to **Extended Data Fig 7f**), we measured the myristoylation status of S6A. Indeed, compared to wt BLK, S6A has strongly impaired myristoylation (added as **Extended Data Fig 7i, j**). Critically, we extended our truncation analysis further and showed that residues #1-7 are sufficient to retain TAK285-mediated BLK degradation (**Fig 3g**).

Finally, for the two additional key positions identified in our DMS dataset, L3 and V4, we found multiple mutants that are (a) resistant to TAK285-mediated degradation (**Extended Data Fig 7h**), while (b) retaining sensitivity to IMP-1088 treatment. Further, the assessed mutants are (c) myristoylated to at least the same degree as BLK (**Extended Data Fig 7i, j**) and in line with this data (d) localized to the membrane (examples shown in **Extended Data Fig 7k**).

When compared to SRC's sequence we noted that L3 and V4 form an aliphatic patch in BLK (**Reviewer Fig R9**), which we hypothesized may be critical for γ -secretase processing. We hence mutated the corresponding residues in SRC to those of BLK. Indeed, we were thus able to generate a SRC variant that was degraded by TAK285 in a γ -secretase dependent manner (see **Fig 3h**, **Extended Data Fig 7l**).

Our initial BioID data already pointed towards an intrinsic affinity between BLK and γ -secretase (**Extended Data Fig 6i**). We now further substantiate evidence for this interaction *in vitro* with data from an MST experiment with recombinant proteins (see **Extended Data Fig 6j**). This motivated the development of a structural model via Alphafold3 (see **Fig 3i**). After minimal relaxation of the system via MD simulation, we found that within the resulting complex, BLK's unique domain is positioned within the active site of γ -secretase. Specifically, the peptide bond between the myristic acid and G2 is coordinated by two hydrogen bonds formed by the two catalytic residues D385 and D275 of γ -secretase. Furthermore, we noted that L3 is positioned within a neighboring hydrophobic pocket, corroborating our mutational data and this residue's important role to enable γ -secretase processing upon TAK285 treatment. We also attempted structural elucidation but could only observe heterogenous complex formation, which disabled the generation of a high-resolution structure.

Given the structural prediction, the extensive mutagenesis and truncation data as well as new literature evidence highlighting that γ -secretase can cleave myristic acid from small molecules (<https://doi.org/10.1021/acschembio.4c00432>), our mechanistic hypothesis thus places TAK285 as small molecule that induces the γ -secretase dependent cleavage of myristic acid from BLK, releasing it into the cytosol where the generated neo N-terminus serves as an N-degron (see **Reviewer Fig R8**).

Since the peptide bond of the myristic acid to G2 is specifically coordinated by the two active site aspartic acids, we hypothesized that this is the site for initial cleavage by γ -secretase. As shown in the original submission, myristic acid is lost from BLK (**Fig 3i**) upon TAK285 treatment and in a γ -secretase dependent manner, which supports this notion. Given the positioning of BLK on the cytosolic side of the secretase complex (rather than being a transmembrane protein), as well as the per-residue decomposition of binding free energy (see **Reviewer Fig R10**), in which loss of the myristic acid drastically increases the energetic contribution of G2, it is most likely that cleavage results in the release of BLK with a remnant glycine into the cytosol. Despite our best efforts, we were, however, not able to identify the precise cleavage product via MS.

Finally, we set out to identify the kinase mediating BLK degradation. Previous competition experiments (moved to **Extended Data Fig 6k**) already informed that direct BLK binding is not important. To exclude involvement of direct binding at the effector (γ -secretase), we conducted MST experiments, where TAK285 was dose-dependently titrated against purified γ -secretase. However, no binding was detected. Collectively, these data are in line with a network-mediated effect of TAK285. To elucidate this, we performed multiple attempts to map potential targets using chemoproteomics. In addition to the Kinobead profiling data from the index submission, we now have synthesized a tethered analog of TAK285 for immobilization to conduct direct affinity enrichment followed by proteomics analysis. While this analog retained BLK degradation activity (data not shown), in-depth analysis failed to reveal targets beyond MAP2K5 which we had already identified via Kinobead profiling. We hence set out to validate MAP2K5, but genetic KO did not alter TAK285 degradation (added as **Extended Data Fig 6n**).

Given that TAK285 is a kinase inhibitor, we would suspect phosphorylation to be the most likely PTM to occur and hence enable γ -secretase processing of BLK. The mutational data would suggest S6 as one potential site. We hence re-visited our genome-wide CRISPR/Cas9 screening data, mapping all kinases, all concatenated chemo-proteomic hits, all as well as the top 10 kinases predicted to phosphorylate S6 within BLK (see **Reviewer Fig R11**). However, no noteworthy hits could be identified. MS identification as well as experiments with synthetic peptides likewise could not resolve if S6 phosphorylation indeed acts as potential *steric switch* to tune γ -secretase processing of BLK. Future work will be required to deconvolute the directly acting target of TAK285-induced BLK degradation.

The RIPK2 findings are also interesting in that they revealed a mirroring of pathogen-induced RIBosome formation, but it is not clear how the RIP-IN-4 compound is inducing RIBosomes, which are naturally caused by RIPK2 activation. Are RIPK2 the degraders that were previously assigned as inhibitors actually acting as activators?

RI-4's (RIPK-IN-4) original publication had assessed *in vitro* inhibition, suppression of IL6 secretion and TNF α secretion in rat and human whole blood stimulated by MDP (DOI: 10.1021/acsmchemlett.7b00258). Collectively, these data point to an inhibitory function of RI-4. To exclude any activation in our cell line models, we further tested for CXCL1/5 and IL1a expression via qPCR. While both positive controls (TNF α as general NFKB stimulus and L18-MDP for NOD1/2 dep. stimulation) upregulated the respective transcripts in K562 cells, RI-4 alone did not induce any activation, arguing against an unexpected role as a RIPK2 activator (see **Reviewer Fig R23**).

Fig R23. qPCR analysis of cDNA from K562 cells treated with 10 μ M RI-4 or TNF α or L18-MDP for 6h.

Do all 9 RIPK2 inhibitors/degraders act through this mechanism? Are there any RIPK2 inhibitors that do not cause assembly and autophagy or is this universal?

This question overlaps with a remark from Reviewer 1, who was additionally interested in the involvement of TMUB1 in the process, which is why we refer to our TMUB1 related findings also in this section.

First, to gain a deeper understanding of how other inhibitors may be able to co-opt the degradation mechanism induced by RI-4, we extended our analysis to two additional hit compounds from our screen. We specifically focused on two selective hit compounds GSK2983559 and LY2109761 with prior evidence of binding to RIPK2. The two compounds completely phenocopied RI-4 induced RIPK2 degradation, with foci appearing and being

cleared on a similar scale and timeframe (see **Reviewer Fig R24 a, b**). Again, TMUB1 KO reduced drug-induced foci formation and slowed down RIPosome clearance.

To better understand the role of TMUB1 in this process, we have now performed several experiments to assess RIPK2 and TMUB1's interaction. First, we performed co-immunoprecipitation of RIPK2-Nluc-3xFLAG highlighting significant RI-4 induced TMUB1 interaction (**Fig 4I, Extended Data Fig 9k**). Secondly, we performed IF staining for TMUB1, which showed TMUB1-RIPK2 co-localized assemblies upon RI-4 treatment (**Fig 4m, Extended Data Fig 9l, m**). This crucially supports our genetic evidence that TMUB1 is involved in orchestrating the formation of RIPosome structures to initiate highly coordinated clearance of these RIPK2 multimers. Noteworthy, also GSK2983559 and LY2109761 treatment also led to TMUB1 being immunoprecipitated with RIPK2 (see **Reviewer Fig R24c**).

Fig R24. Analysis of additional RIPK2 degraders **a** Quantification of microscopy images depicting the normalized BFP levels per cell for RKO iCas9 RIPK2 stability reporter CTRL or TMUB1 KO cells (corresponding to KO1 in manuscript) treated with DMSO, RI-4 or the indicated additional hit compounds (n = 3). **b** Quantified foci count per cell (data matched to (a)). **c** FLAG co-immunoprecipitation of RIPK2-Nluc-3xFLAG K562 cells treated for 2h with indicated compounds (n = 3). **d** Quantified microscopy data depicting normalized BFP levels per cell for indicated compounds in RKO iCas9 RIPK2 stability reporter (n =3). **e** Quantified foci count per cell (matched to (d)). **f** Flow cytometric assay of DMSO, RI-4, Gefitinib or Regorafenib treated KBM7 iCas9 RIPK2 stability reporter cells co-treated with DMSO or Bafil. A1 for 18h (n = 2). All RIPK2 targeting compounds were treated at 2.5 μ M.

Moreover, we explored the capacity of additional RI-4 inhibitors to induce and clear RIPK2 assemblies, covering both known Type I and Type II RIPK2 inhibitors. We identified two additional degraders, namely Gefitinib and Regorafenib (Type I and Type II inhibitors, respectively). Contrary to RI-4, however, they induced RIPK2 degradation at a slower rate and with less foci per cell (see **Reviewer Fig R24d, e**). Furthermore, unlike the two additional hit compounds GSK2983559 and LY2109761, TMUB1 was not efficiently engaged by neither Gefitinib nor Regorafenib correlating with their less efficient clearance rate, and again highlighting TMUB1's role as facilitator in this process. Of note, degradation remained

dependent on the lysosomal pathway indicating a partially overlapping mechanism whereby different inhibitors induce RIPK2 assemblies that are subsequently cleared by autophagy (see **Reviewer Fig R24f**).

Finally, we provided additional data in the manuscript (following additional Reviewer comments) surrounding RI-4 mediated RIosome formation and clearance. First, we performed temporal BioID (1h, 4h after DMSO or RI-4 treatment and 18h after Bafil. A1 or RI-4 and Bafil. A1 treatment compared to GFP control). The results are included on p.13-14 and corresponding figures **Fig 4k** and **Extended Data Fig 9a**. In sum, we identified many previously established RIosome components, particularly those associated with an antagonizing function (e.g. TNFAIP3, CYLD, N4BP1 and CCDC50). Furthermore, we found clathrin associated proteins such as EPS15 and proteins associated with ubiquitin dependent vesicle-mediated transport such as GGA3. Our top scoring interactor across all timepoints was SQSTM1/p62, a well-established autophagy receptor. This finding is well aligned with our data that knockout of the scaffolding protein FIP200 prevents RI-4-induced RIPK2 degradation. FIP200 is a SQSTM1/p62 interactor (Turco *et al.*, 2019) and essential to bridge the recognition of autophagic cargo to the autophagy initiation machinery.

GO enrichment of the drug-induced RIPK2 interactome further highlighted the importance of ubiquitination for this process. The top three (1h, **Extended Data Fig 9a**) or four (4h, **Reviewer Fig R16**) most significant GO molecular function terms (2023) were ubiquitin associated terms. For the 1h timepoint (**Extended Data Fig 9a**), we further identified Kinase Binding (GO:0019900) and Protein Kinase Binding (GO:0019901) among the top 10 most enriched terms.

Given the importance of ubiquitination in RIPK2 assembly clearance, we set out to identify the E3 ligases involved in the process. In parallel to our temporal BioID we set up an additional ubiquitin-library focused CRISPR screen (see **Reviewer Fig R19**). Integrating both datasets identified multiple IAP containing E3 ligases: BIRC2/6/8 and XIAP as well as RLIM and TRAF2 (CRISPR screen) and UBR4/5 (BioID). Given previously reported data establishing XIAP as one gatekeeper of RIosomes, we performed single and dual KO of each of the hits and XIAP. Only dual knockout (DKO) of the two IAP containing E3s BIRC2 (cIAP1) and XIAP was able to abrogate degradation (see **Extended Data Fig 9b-d**), while all other combinations showed, at best, only a minor disruption of the degradation phenotype (see **Reviewer Fig R19a-e** and **R20**). Finally, a RIPK2 I212D mutant fully phenocopied the XIAP and cIAP1 (BIRC2) DKO (**Extended Data Fig 9e-g**). This is of particular interest as literature evidence pinpoints I212 as one of the critical residues for the interaction of RIPK2 with IAP domains (PMID [32954645](https://pubmed.ncbi.nlm.nih.gov/32954645/)).

What do the authors think is causing CSK degradation by the RIPK2 compound, which appears to be even more substantial than the RIPK2 degradation (Ext 5c). Is there anything about the CLK1/MAP3K1 stabilization worth mentioning?

We can't offer a clear explanation for the downregulation of CSK. We did not specifically explore the stabilization of CLK1/MAP3K1 but thank the reviewer for this excellent observation. We, however, noted that amongst our drug screening dataset we could also identify multiple stabilization events. Following also comments made by the other reviewers, we thus now significantly extended our analysis and also conducted experimental validation of individual stabilizer examples. The corresponding figures, documents and methods have been

added on p. 4, **Supplementary Data Table 1** and **Extended Data Fig 3**. In total, we identified 204 stabilization events, across 64 kinases and 128 different compounds.

To validate our findings on protein stabilization, we focused our efforts on one kinase, ABL1, that was stabilized by DPH, a small molecule that binds to the allosteric, myristic acid binding pocket of ABL1. We could successfully confirm stabilization in our stability reporter setup in a dose-dependent manner. Further, stabilization was abrogated by mutating C464W, a critical residue within the myristoyl binding pocket (<https://pmc.ncbi.nlm.nih.gov/articles/PMC7566958/>) highlighting a mechanism of action induced by direct drug binding. We next sought out to test other allosteric ABL1 pocket binders/inhibitors with particular focus on clinically relevant small molecules (asciminib and GNF-2) to generalize this phenomenon. These had been in parts excluded in our drug screen (see Materials & Methods) due to their toxicity and global downregulation of protein abundance in the BCR-ABL driven screening cell line K562. Intriguingly, both compounds displayed identical behavior to DPH, stabilizing ABL-1 in a dose dependent manner, which was again abrogated by the C464W mutation (see **Extended Data Fig 3d, e**). For the remainder of stabilization events, we are currently following up multiple leads in a follow-up study. All associated data has, however, been made available in full in above-described documents and datasets.

Referees' comments:

Referee #1 (Remarks to the Author):

The authors have made commendable efforts to address all the reviewers' comments and have added a substantial number of new experiments.

The authors have done an incredible amount of follow-up work to identify the mechanisms of action of the three follow-up compounds. In particular, the BLK degradation story is remarkable, and it is indeed a shame that despite the authors' heroic efforts, they could not identify the target of the small molecule. It would have been so interesting to know how exactly TAK285 activates gamma-secretase to cleave BLK! But it would feel wrong to ask the authors to continue these efforts within this manuscript, which is already bursting at seams. Similarly, the RIPK2 degradation story is significantly improved and provides compelling evidence for the molecular mechanism behind RIPK-IN-4.

The inclusion of stabilizer part in Extended Data Fig. 3 was an interesting addition. The number of stabilization events (64 kinases and 128 different compounds) is quite significant compared to the destabilization events (66 kinases and 232 compounds), suggesting that about one-third of all stability changes involve kinase stabilization. This further argues for indirect "network" effects with many compounds.

My main remaining issues have to do with the larger context of the manuscript. In the end, it presents three distinct stories which are very loosely connected. As previously pointed out, the novelty of LYN degradation via CSK inhibitors is limited. Considering the length of the manuscript, I feel that it would be only strengthened by focusing on the more "developed" stories. I still think that this story would be better served as a separate manuscript. Moving the LYN part in the supplements would likely bury it completely. That said, I will let the editor and the authors discuss whether to include this part in the final manuscript if it proceeds to publication.

We thank the Reviewer for bringing up this suggestion, which we also had considered. Altogether, we however came to the conclusion that we would like to keep the story as part of the manuscript since we believe that the activity-stability switch is an important conceptual mechanism that might underpin a significant fraction of the other inhibitor-induced degradation events we have identified in the global survey.

The second aspect is the practical conclusions of this work. Again, the amount of work put into this is truly impressive, but in the end the take-home message remains unclear. The authors mention that their results highlight "supercharging" endogenous degradation (or stabilization?) circuits. But this notion is quite vague and it is not at all clear to me what supercharging actually means, even after reading the manuscript. The underlying mechanisms are clearly very distinct, and it feels like an overreach to put them all under the umbrella of "supercharging".

Overall, I lean towards recommending accepting this manuscript for publication despite these shortcomings. Future studies by the Winter lab will hopefully solve the mystery of BLK degradation!

In order to clarify the term "supercharging", we have added an additional sentence in the manuscript. Considering that this terminology has thus far been well received when presenting this story at meetings, and considering that we used the same terminology in a parallel manuscript that describes the mechanism of action of inhibitors/degraders of IDO1 (Hennes et al, bioRxiv 2024, accepted in principle in Nature Chemistry), we would strongly prefer to keep this term since we believe it will provide a compelling framework for future studies. The unifying message that we see in our paper is that all of the closely investigated compounds induce kinase degradation by supercharging/accelerating native degradation pathways that are also turning over the kinase in absence of the inhibitor. The three vignettes outline that, from a molecular and mechanistic perspective, supercharging can occur by drug-induced changes in (i) kinase activity, (ii) kinase localization and (iii) kinase multimerisation status. From the aforementioned study on the non-kinase protein IDO1, we know that also (iv) drug-induced changes mimicking co-factor binding can drive target degradation via supercharging. It is hence reasonable to assume that, as the field develops further, additional mechanistic underpinnings will be identified that contribute to the larger logic of inhibitor-induced supercharging.

Below are additional comments related to the revision:

LYN degradation.

In the original manuscript, the authors demonstrated that SI-3 induces LYN degradation by CBL/CBL-B and suggested that Y316A is a new phosphodegron for CBL along with Y32, a known phosphodegron. To address more possible mechanisms underlying LYN degradation by SI-3, the author performed additional experiments including ubiquitination (K48, K62) assay, membrane association (IMP-1088 treatment) and Y508A mutant rendered LYN resistant to SI-3 degradation. As a result, the manuscript included a large volume of experimental data.

The authors wanted to emphasize that SI-3 is the only CSK inhibitor that induces LYN degradation. So, they added comparative Western blot of set of LYN binders. However, the manuscript still lacks a mechanistic explanation as to why SI-3 uniquely exerts this effect, which was requested in the earlier review comment. That said, as discussed above, this is not the most significant story of the manuscript so pursuing this further is not in my opinion essential at this point.

We thank the reviewer for highlighting the added value of our experiments surrounding the mechanism of action of SI-3 induced LYN degradation. As stated in the manuscript, the differential inhibitory capacity of CSK over LYN distinguishes SI-3 from all other CSK binders and thus mitigates LYN degradation. While this is beyond the scope of the current work, future efforts could and should focus on compounds that exert similarly differential inhibitory capacity so as to orthogonally confirm the findings in a bottom-up approach.

BLK degradation.

The authors proposed that TAK285 induces BLK degradation via γ -secretase-mediated cleavage of the N-terminal myristoylation, supported by KO screening and BioID. In the revised version, deep mutational scanning (DMS) was used to further investigate the N-terminal residues of BLK, and potential interactions with γ -secretase were explored through alpha fold prediction. The authors also showed, using MST, that TAK285 does not directly enhance the interaction between BLK and γ -secretase.

However, the key mechanistic question remains still unresolved: how TAK285 promotes γ -secretase-dependent cleavage of BLK in the absence of direct binding to either protein as the authors also mention in the Discussion.

Of note, the findings in Reviewer Fig. R5 are intriguing. The fact that 4-Hydroxyonchocarpin acts as a γ -secretase-dependent degrader, whereas FIN-1 does not, may offer mechanistic insights and deserves further exploration.

We were indeed equally struck by the fact that 4-Hydroxyonchocarpin phenocopies TAK285 behavior given that it lacks structural similarity to TAK285 and also based on the fact that it has a different associated kinase target space. We are fully committed to identifying the key players to further expand on our findings in future work.

RIPK2 degradation.

The RIPK2 section was relatively well addressed in the original submission. The addition of immunoprecipitation data showing a RIPK-IN-4 induced interaction between RIPK2 and TMUB1 strengthens the proposed mechanism.

Testing of additional compounds including GSK583, RIP inhibitor 2, WEHI-345, Regorafenib, Gefitinib, GSK2983559, and LY2109761 was a valuable addition. Interestingly, only RIPK-IN-4 and GSK2983559 promoted foci formation and interaction with TMUB1. LY2109761 —despite not being a classical RIPK inhibitor— also induced foci formation and TMUB1 binding, which could offer new clues into the underlying mechanism.

We thank this reviewer for the positive comments on RIPK2.

Minor points

Extended Data Fig. 5i: Y507 should likely be Y508

Extended Data Fig. 3d,e: The label "GK" should be corrected to "C464W", as the gatekeeper residue in ABL is T315 and the corresponding mutation in the myristoyl pocket is C464W.

We thank this reviewer for the attention to detail and have altered both figures as suggested.

Figure 4k is difficult to understand in the absence of X axis labels. Does the X axis represent something or is the scatter there just to separate points better on the Y axis?

To clarify this figure, we have altered the figure legend and stated that the x-axis was jittered for readability.

Referee #2 (Remarks to the Author):

The authors have done a commendable job of addressing the very large number of points raised in the three reviews, generating an overwhelming 44-page rebuttal (!), which includes a massive amount of new experimental data designed to address many of the issues the reviewers raised. They have added a significant amount of these new data in the revised version (e.g. in Figure 3 and in several of the supplementary figures). Many of the new results satisfactorily address concerns I had raised.

In my view, the most exciting new finding is that in response to TAK285 treatment γ -secretase cleaves the N-terminal myristoyl group from BLK, with the N-terminal 7 residues of BLK being sufficient to confer this response when grafted onto another totally unrelated protein such as BFP. Nevertheless, despite extensive efforts the authors failed to define molecularly how the TAK285 kinase inhibitor promotes γ -secretase cleavage of the BLK myristoyl group - does TAK285 activate cleavage of any other known γ -secretase target proteins? For instance, they were unable to identify the key kinase target for TAK285, determine the novel N-terminal residue of γ -secretase-cleaved demyristoylated BLK, or identify an E3 Ub ligase responsible for soluble BLK degradation. With regard to the new data generated for kinase inhibitor-induced LYN and RIPK2 degradation, the authors' identification of BIRC2/cIAP and XIAP as E3 ligases necessary for RI-4-induced degradation of RIPK2 and the additional evidence that RIPK2 aggregates are cleared by autophagy strengthen the paper.

Points: 1. Although this might be beyond the scope, it would strengthen the authors' conclusions if they provided direct biochemical evidence that purified γ -secretase cleaves the myristoyl group from purified Myr-BLK. Their modeling of the Myr-BLK N-terminal sequence into the active site of γ -secretase and the alignment of the Myr-Gly bond with the catalytic Asp residues is quite persuasive, but modeling alone does not prove that the cleavage occurs at the Myr-Gly amide bond, and it would be important to demonstrate that the N-terminus of the soluble BLK is in fact Gly as is proposed; in the rebuttal the authors say they failed to define the novel N-terminus of released BLK, but perhaps this could be accomplished with the product of a biochemical cleavage assay. This information is relevant to the mechanism through which the soluble BLK is degraded in the cytoplasm. In their rebuttal, the authors mention the negative results they obtained with UBR1/2/4/5 KO intended to determine if released BLK is degraded through an N-end rule Ub ligase-dependent mechanism. Here it is worth noting that, if Gly were the new N-terminus (as posited in Fig R8), the released BLK protein will not be targeted by the N-end rule degradation pathway because N-terminal Gly is a strongly stabilizing N-end rule amino acid residue ($t_{1/2} = 30$ h). Based on the high stability of the G2A BLK, which is not membrane anchored, the Gly2 residue does seem to be important for degradation induced by TAK285. It should be a priority to determine the identity of the E3 ligase for soluble BLK and what it recognizes in the N-terminal 7 residues of BLK. In this regard, based on the known hydrophobic γ -secretase cleavage sites in APP that generate A β peptides, is it possible that γ -secretase cleaves the Leu-Val (or Val-Ser bond) bond in BLK? Consistent with this idea, substitution of Ser3Asn4 in the c-SRC N-terminal region with Leu3Val4 rendered it a target for TAK285/ γ -secretase induced degradation. If Myr-BLK were cleaved before Ser5, then it would be rapidly degraded by the N-end rule pathway, and if c-SRC Leu3Val4 were cleaved before Lys5, it would also

be rapidly degraded by the N-end rule pathway.

We thank this reviewer for these suggestions. In future work, we plan to address cleavage in a biochemical assay; but, as it stands, our inability to identify the direct target of TAK285 has hampered our biochemical efforts. In an attempt to decode the exact position of cleavage and thus answer this question, we instead employed N-terminally fused ubiquitin reporter variants, previously used for studying the N-end rule pathway (10.1126/science.3018930). In brief, the N-terminally installed ubiquitin is cleaved off *in cellulo*, thereby exposing a designated 3' N-terminal residue. We adapted this system to our stability reporter, generating several variants to scan residue positions #1 (Met) to #5 (Ser) as neo-N-termini, measuring the respective baseline stability as well as response to TAK285 treatment via flow cytometry.

As shown below, reporters starting with either residue #1 or #2 fully maintain TAK285 degradation capacity and, as expected, display baseline stability comparable to full-length BLK. Reporters starting at #3 and #4 render BLK unstable at baseline and thus no TAK285-induced degradation is observed any more. We would like to point out that initial cleavage of γ -secretase on the hypothesised Myr-Gly linkage may lead to further processing either by γ -secretase itself or other proteases. Hence, neither of these residues can be excluded as potential neo-N-termini. Finally, cleavage between residue #4 and #5 can be excluded as the reporter starting at residue #5 is insensitive to TAK285-induced degradation. Future work will, however, be required to fully elucidate this intricate mechanism of γ -secretase-induced BLK degradation. We highly appreciate the thoughtful feedback given by this Reviewer, but conclude that these additional experiments will have to be part of a separate, dedicated study.

Reviewer Fig R1. The above designated N-terminal BLK ubiquitin constructs were utilised in flow cytometric assays to determine baseline stability and TAK285 degradation capacity for different neo N-termini in an unbiased manner (n = 3).

2. Although beyond the scope of this paper, it would also be interesting to know if γ -secretase can remove the myristoyl group from other myristoylated SFKs, and possibly other myristoylated proteins, and, if so, what are the functional consequences for turnover of the demyristoylated protein. They showed that demyristoylation of BLK is dependent on the sequence of the first 7 aa of BLK and particularly the Leu3, Val4 and Ser6 residues ruling out most SFKs as targets, but can they predict what other Myr proteins might be targets for γ -secretase? For example, can a bioinformatics exercise be conducted on mammalian proteins with a Gly2 residue to predict additional possible myristoylated protein targets for γ -secretase based on the sequence determinants needed for Myr-BLK cleavage?

We would like to thank the reviewer for this beautiful suggestion. We equally believe that this opens up a novel category of potential γ -secretase substrates. These in turn may be accessible to drug-induced modulation, thus enabling novel routes of e.g. induced re-localisation of membrane tethered proteins for therapeutic purposes. We will explore this avenue in future work; initiating the bioinformatics survey suggested by this reviewer will be a great place to start.

Referee #3 (Remarks to the Author):

The authors have returned an outstanding revision of their original manuscript with a great deal of new data and, importantly, new insights. I agree with the authors' assertion that the scale of their experiments is deeply helpful for understanding the frequency of these events (i.e. inhibitor-induced

destabilization), which has clear implications for seeking out these effects in the process of drug discovery or, alternatively, avoiding them. I would call attention to the increased clarity surrounding gamma secretase mediated destabilization of BLK, which will be of keen interest to many and was thoroughly investigated with multiple orthogonal, unbiased, and data-rich experiments. Though the authors have been unable to identify the proximal target of TAK285 that induces gamma secretase to cleave BLK, the additional data in the revision will provide be a compelling start for future experiments. The increased mechanistic understanding of RIPK2-induced degradation is equally impressive. The authors have clearly put much effort into this revision and responded faithfully to our comments. The manuscript will be of widespread interest, and I would recommend its acceptance so the rest of the community can benefit from this excellent resource and collection of findings.

We would like to thank this reviewer for these positive remarks. We are equally excited about the \$\gamma\$ -secretase findings and will explore this avenue in future work.

Referee #4 (Remarks to the Author):

I co-reviewed this manuscript with one of the reviewers who provided the listed reports.
We would like to thank this reviewer for their co-reviewing comments.